J Physiol 603.22 (2025) pp 7089–7121

7089

# Reduced osmoresponsiveness in magnocellular neuroendocrine neurones during chronic salt-loading

Maja Lozic, Roongrit Klinjampa , Nancy Sabatier, Duncan J. MacGregor, Gareth Leng and Mike Ludwig

*Centre of Discovery Brain Sciences, University of Edinburgh, Edinburgh, UK*

Handling Editors: Vaughan Macefield & Vagner Antunes

The peer review history is available in the Supporting Information section of this article (https://doi.org/10.1113/JP288860#support-information-section).

**Abstract figure legend** We conducted a large systematic survey of the physiological and electrophysiological characteristics of the magnocellular neurosecretory neurones of the supraoptic nucleus in euhydrated and salt-loaded conditions. Our *in vivo* electrophysiological recordings show that, after 7 days of salt-loading, the neuronal responsiveness to acute i.v. osmotic stimuli were markedly impaired with no change in their responses to non-osmotic stimuli. The "plasticity" that we have identified in this system appears not to enhance osmoresponsiveness in response to chronic osmotic challenge, but to abate it, presumably to avoid oversecretion beyond what is physiologically adequate.

**Maja Lozic**, gained her PhD at the University of Belgrade, Serbia, in the laboratory of Professor Nina Japundzic-Zigon. She held a position of Assistant Professor at the Faculty of Medicine in Belgrade, studying the role of the autonomic nervous system in cardiovascular regulation. Maja moved to Edinburgh in 2018 and as a postdoctoral fellow examined the role of hypothalamic neurones in food intake and osmoregulation. Maja currently holds an honorary position at the University of Edinburgh while working as a doctor for the NHS. **Roongrit Klinjampa** is a physiologist specializing in neuroendocrinology. In 2024, Roongrit obtained his PhD in Biomedical Sciences at the Centre for Discovery Brain Sciences of the University of Edinburgh, where he was trained in *in vivo* electrophysiology of hypothalamic neurones under the mentorship of Gareth Leng and Mike Ludwig. He is now Lecturer and early-career researcher at the Chulabhorn Royal Academy in Bangkok, Thailand, interested in the neural control of hormone secretion.

M. Lozic and R. Klinjampa contributed equally to this work.

**Abstract**  Here, we studied the effects of salt-loading in rats on the electrophysiological behaviour of neurones that secrete oxytocin and vasopressin. After 7 days of salt-loading, the basal firing rate of both vasopressin cells and oxytocin cells in urethane-anaesthetized rats was increased by less than 1 spike s$^{-1}$, which is much less than expected from the hyperosmolality induced by saltloading. The neuronal responsiveness to acute osmotic stimuli was also markedly impaired, with no change in their responses to non-osmotic stimuli. We then undertook a systematic search of the literature for studies in salt-loaded rats that had measured oxytocin or vasopressin secretion, plasma osmolality, haematocrit or pituitary hormone content, and reviewed them in light of our electrophysiological findings. The prevailing understanding is that salt loading induces plastic changes in neuronal behaviour to promote exaggerated vasopressin secretion, but the conclusions that we draw from our electrophysiological findings in urethane-anaesthetized rats and the literature review suggest the converse - that vasopressin neurones selectively habituate to osmotic stimuli, presumably to conserve diminished pituitary stores of vasopressin while sustaining enough secretion for maximal renal effects.

(Received 10 March 2025; accepted after revision 27 August 2025; first published online 25 September 2025)

**Corresponding author** M. Ludwig: Centre of Discovery Brain Sciences, Hugh Robson Building, George Square, University of Edinburgh, Edinburgh, EH8 9XD, UK.    Email: mike.ludwig@ed.ac.uk

## Key points

- Seven days of 'salt-loading' produces a large increase in plasma osmolality and depletes the pituitary content of vasopressin and oxytocin, apparently reflecting enhanced secretion.
- Using *in vivo* electrophysiological recordings in urethane-anaesthetized rats, we show that, after 7 days of salt loading, the basal firing rate of both vasopressin and oxytocin cells was increased to a much lesser extent than expected from the hyperosmolality induced by acute osmotic stimuli.
- The neuronal responsiveness to acute osmotic stimuli was also markedly impaired with no change in their responses to non-osmotic stimuli.
- Our results show that these neurones strongly and selectively habituate to chronic osmotic stimuli, presumably to conserve diminished pituitary stores of hormone.
- Our conclusions contradict the prevailing understanding that salt loading promotes an exaggerated hyperexcitability of the hypothalamo-neurohypophysial system.

## Introduction

In some conditions, particularly when kidney function is impaired, elevated salt intake can produce 'salt-sensitive hypertension'. The aetiology of this is unclear (Nishimoto et al., 2024), but, prompted by studies in rats that involved replacing drinking water with 2% NaCl, some have suggested that excessive vasopressin secretion is implicated (Prager-Khoutorsky et al., 2017; Qian, 2018). In an early study, Jones and Pickering (1969) reported that 5 days of such 'salt loading' produced a large increase in plasma osmolality and depleted the neurohypophysial content of vasopressin and oxytocin to 18% and 10% of the control content, and many studies have since confirmed and extended these findings. After 7 days, there is increased expression of many genes in the magnocellular neurones that synthesize the neurohypophysial hormones oxytocin and vasopressin (Greenwood et al., 2015;

Johnson et al., 2015), including mRNA expression of oxytocin, vasopressin and co-existing modulators such as nitric oxide synthase (Kadowaki et al., 1994) and neurotrophins (D'Amato et al., 2012). There are also changes in the expression of genes involved in the actions of neurotransmitters; changes in cell morphology including a remodelling of the actin cytoskeleton (Barad et al., 2020) with potential consequences for dendritic peptide secretion (Tobin et al., 2012); and a doubling of the surface area of magnocellular soma (Zhang et al., 2001) with potential consequences for membrane resistance and voltage-dependent ion channels (Wu & Tasker, 2017).

Then, in 2011, it was reported that, in rats that had been drinking 2% NaCl for 5 weeks, systolic blood pressure was increased by ∼15 mmHg (Cruz et al., 2011). Subsequently, other groups reported a similar rise within 4–7 days (Choe et al., 2015; Gomes et al., 2023; Li et al., 1998; Ribeiro et al., 2015, 2020).

Recently, studies have reported that, in response to salt loading, the responsiveness of magnocellular neurones to excitatory transmitters is increased (Di & Tasker, 2004; Di et al., 2019), their intrinsic osmosensitivity is increased (Levi et al., 2021; Prager-Khoutorsky & Bourque, 2010), and changes in the expression of $Na^+$ channels enhance the excitability of vasopressin neurones (Sharma et al., 2017). Moreover, it has been suggested that the effects of GABA (the major inhibitory transmitter regulating vasopressin cells) are reversed (Choe et al., 2015; Kim et al., 2011) as a result of increased intracellular $[Cl^-]$. These observations suggested that, in salt-loaded rats, vasopressin cell activity might be enhanced beyond that expected from the raised osmolality, with the implication that sustained hypersecretion of vasopressin might contribute to hypertension 'by acting on kidney cells to promote water retention and by promoting constriction of blood vessels' (Marosi & Mattson, 2015).

Here, we studied magnocellular neurones in the supraoptic nucleus of salt-loaded rats to evaluate their discharge behaviour and their responsiveness to physiological stimuli. We expected a higher basal firing rate, an increased responsiveness to acute osmotic stimulation and a change in the responses of vasopressin cells to stimuli that are normally inhibitory. In humans, oxytocin secretion is not increased in response to hypernatraemia (Williams et al., 1986) and oxytocin has no natriuretic effect at physiological concentrations (Rasmussen et al., 2004), but, in the rat, oxytocin cells respond to acute osmotic challenges with similar changes in firing rate to that seen in vasopressin cells (Brimble & Dyball, 1977) and oxytocin is a potent natriuretic factor at physiological concentrations (Leng & Russell, 2019; Verbalis et al., 1991). Accordingly, we expected that oxytocin cell activity would also be elevated in salt-loaded rats.

We found changes in the patterning of activity consistent with enhanced activity-dependent conductances, with consequences for spike pattern consistent with optimizing the efficiency of firing in vasopressin cells. However, the increase in basal firing rate was modest, responses to osmotic challenge were reduced rather than increased and we saw no weakening of inhibitory actions.

## Methods

### Ethical approval

Animal research at the University of Edinburgh is overseen by an Animal Welfare and Ethical Review Body (AWERB). At Edinburgh, the AWERB advises the establishment licence holder on whether to support Project Licence applications involving the use of animals in research, and the review process involves critical evaluation of all ethical, scientific and welfare issues related to the project. The present studies were performed under Project Licence PP2098167, which was issued by the UK Home Office after its approval by the AWERB (https://www.ed.ac.uk/research-innovation/animal-research/animal-welfare-ethics/ethical-review-and-awerb). All animal work complies with the standards stated for *The Journal of Physiology*, and all investigators understand the ethical principles under which the journal operates and confirm that their work complies with this animal ethics checklist.

### Animals

Male Sprague–Dawley rats (body weights 320–380 g) purchased from Charles River Laboratories (Tranent, UK) were randomly allocated to three groups: euhydrated rats with *ad lib* access to water; salt-loaded rats for which drinking water was replaced by 2% NaCl for 7 days; and rehydrated rats that had received 2% NaCl for 7 days but then had drinking water returned to them for 2 days. Rats were maintained in an ambient temperature of 21°C with free access to normal rat chow (RM1; $Na^+$ content 2500 mg $kg^{-1}$; sds-diets.com), with lights on at 07.00 h and off at 19.00 h.

### Anaesthesia

In the morning (at 09.00–10.00 h), rats were anaesthetized with isoflurane inhalation (isoflurane, Part No. 26675-46-7; Piramal Critical Care Ltd, West Drayton, UK) followed by I.P. urethane (Sigma-Aldrich, Merck Life Science UK Ltd, Poole, UK; 25% w/v in distilled water; 1.3 g $kg^{-1}$, I.P.) to achieve long-lasting anaesthesia. Surgical procedures were begun if the rat's mucous membrane colour and respiration appeared normal, but the withdrawal and palpebral reflexes were absent. At this dose, urethane anaesthesia can last for at least 8 h without any need for supplementation (Field & Lang, 1988). At the end of each experiment, rats were killed by an overdose of sodium pentobarbital (200 mg $kg^{-1}$ I.V.).

Once the rat was deeply anaesthetized, a femoral vein and the trachea were cannulated, and the pituitary stalk and supraoptic nucleus were exposed transpharyngeally, as described in detail elsewhere (Leng & Sabatier, 2014).

### Electrophysiological recordings

A stimulating electrode (SNEX-200X; Clarke Electromedical Instruments, Reading, UK) was placed on the pituitary stalk to deliver single matched biphasic pulses (1 ms and 1 mA) for antidromic identification of supraoptic neurones (Leng & Sabatier, 2014). A glass micropipette (tip diameter ~1 μm, filled with 0.9% NaCl) mounted on an SM7-remote control system (Luigs &

Neumann, Ratingen, Germany) was introduced into the exposed supraoptic nucleus for extracellular recording of single neurones and an Ag/AgCl wire for ground reference was inserted into the tissue adjacent to the exposure. Recordings were made with an Axopatch 200B amplifier connected to a CV203BU head stage (Molecular Devices Inc., Sunnyvale, CA, USA). The Axopatch was used in current clamp ('track') mode with a four-pole low-pass Bessel filter (bandpass 5 kHz). Signals were filtered using a Hum Bug 50 Hz noise eliminator (Quest Scientific Instruments Inc., North Vancouver, BC, Canada) to exclude residual mains interference. The signal was interfaced at 10 kHz by a Micro 1401 analogue-to-digital converter (CED1401; Cambridge Electronic Design Ltd, Cambridge, UK) to a personal computer using Spike2 software, version 10.07 (Cambridge Electronic Design Ltd).

Recordings were made between 11.00 h and 17.00 h by one of three researchers (RK, NS and MaL) who were randomly allocated rats of the three groups. To generate a sample of supraoptic neurones without selection bias, cells were found by using electrical stimuli applied to the neural stalk at 0.3 Hz to ensure that slow-firing or silent cells would be recognized by the evoked antidromic spike. Each cell encountered was tested for antidromic identification (constant latency and collision tests) and then recorded without stimulation until a 10-min period of stable spontaneous activity had been recorded. Cells were identified as phasic if they displayed abrupt transitions between periods of activity at >4 spikes $s^{-1}$ and periods of near silence, and generally phasic cells were recorded over a longer basal period to establish baseline activity. In some cases, two (or very rarely three) cells could be distinguished in the recorded signal by their distinct spike heights, different waveforms and different latencies to antidromic activation. These signals were separated by spike height using analysis features of Spike2.

## Spike waveform analyses

Spike waveforms were constructed as spike-triggered averages of >>1000 spikes recorded over a period when spike height was stable. Although extracellular recording is generally used only to detect spike occurrences, the shape of the spikes reflects features of the intracellular action potential. Spikes recorded close to the axon or dendrite are expected to have an initial positive phase (reflecting outward return currents), whereas spikes recorded close to the soma are expected to be larger, negative spikes that reflect inward currents in the soma and most proximal dendrites (Gold et al., 2006). The cells presented diverse spike waveforms, and commonly both the shape and size altered during the recordings. We analysed only cells with large negative-going spikes,

generally with a 'hump' on the repolarizing phase that is considered to correspond to an inward $Ca^{2+}$ current linked to invasion of the proximal dendrite (Mason & Leng, 1984). We were interested in whether the waveform would reveal the sequence of hyperpolarization and depolarization expected of the superposition of a hyperpolarizing afterpotential (HAP) and a depolarizing afterpotential (DAP). To average waveforms from different cells, we aligned the average waveforms to the position of the maximum negative deflection ($t_0$), then to a basal potential measured between 10 and 15 ms before each spike (the zero level) and finally normalized them to the maximum spike height (i.e. cells had a normalized spike height of 1 at $t_0$). We considered whether spike-triggered waveforms would be compromised by the occurrences of subsequent spontaneous spikes, and so, for a sample of cells, we compared the average waveforms of all spikes with those of spikes that were not followed by another spike in the subsequent 100 ms. These were, in all cases, indistinguishable (by eye).

## Spike patterning

For all cells that we examined, the basal electrophysiological characteristics from ∼10 min of stable basal activity (i.e. a period with no evident drift or other change in firing pattern). The variability of activity was assessed by the coefficient of variation (CV = SD mean$^{-1}$) of interspike intervals (ISIs). For events governed by a purely random process, the CV is 1 and is independent of the rate; for events occurring more regularly, the CV is <1; and when events occur in clusters the CV is >1.

For each cell, we constructed an ISI distribution in 1-ms bins. The skew, kurtosis and the mode of each distribution was calculated using built-in functions in Excel (Microsoft Corp., Redmond, WA, USA). Modes were not recorded for a few oxytocin cells that fired too slowly to register a clear mode. To average data across cells, the distributions were normalized to the total number of intervals.

The hazard function expresses ISI data in a way that reflects how neuronal excitability changes over time since the last spike, and were constructed (in 10-ms bins) for each cell using the formula (Sabatier et al., 2004):

$$hazard\,in\,bin\,[t, t + 10] = 100 * (\#ISIs\,in\,bin\,[t, t + 10]) /$$
$$(\#ISIs > t + 10 \text{ ms})$$

The maximum value of the hazard and its position, (corresponding to the peak post-spike depolarization) were determined from ISIs <90 ms. The asymptote was estimated from the average hazard between 140 and 290 ms, and the difference between the maximum and asymptote ('peak hazard') was taken as an indication of the amplitude of the DAP.

In phasic cells, bursts were identified as a period of ≥5 s containing ≥20 spikes separated by ISIs ≤1 s, with intervals of >5 s between bursts. We analysed phasic cells where we recorded at least two complete bursts and complete interburst intervals in a control period of >15 min, measuring mean burst duration, interburst duration, intraburst firing rate and total spikes per burst. The software is freely available from us on request.

### Responses to I.V. injection of cholecystokinin (CCK) and phenylephrine (PE)

After at least 10 min of stable basal activity, cells were tested for their response to I.V. CCK (25 µg kg$^{-1}$; CCK-(26-33)-sulphated; Tocris Bioscience, Bristol, UK; dissolved in 0.9% NaCl at 25 µg mL$^{-1}$ and pre-warmed to ~37°C). Such injections increase plasma [oxytocin] but not [vasopressin] (Verbalis et al., 1986); they excite neurones in lactating rats that are identified as oxytocin cells by their response to suckling (Leng et al., 1991) and inhibit most phasic cells (Renaud et al., 1987). Repeated injections of CCK give consistent electrophysiological responses within cells (Leng et al., 1992), but the response magnitude varies between cells.

Some cells were tested with I.V. injection of PE (10 µg in 0.1 mL of isotonic saline, pre-warmed to ~37°C), which transiently raises arterial blood pressure by ~60 mmHg, and generally inhibits phasic cells with no effect on oxytocin cells (Harris, 1979; Leng & Sabatier, 2014). Its effect on phasic cells depends on the phase of their activity cycle at which it is given: close to the onset of a burst, there can be no effect or even a prolongation of the burst; if given late in a burst, the burst is generally abruptly terminated, but bursts are easily stopped by perturbations given when the burst is close to spontaneously ending (Sabatier & Leng, 2007). In this study, injections were given when phasic cells were in the middle of bursts. We quantified the response as the percentage change in firing rate in the 5 min after injection compared to the 5 min before injection.

In two urethane-anaesthetized rats, we cannulated a femoral vein and a femoral artery to record mean blood pressure as described previously (Leng et al., 1991). These experiments were conducted to confirm the effect of PE injections on blood pressure and the lack of effect of CCK.

### Microdialysis

In some rats, a microdialysis probe was placed onto the ventral surface of the nucleus (Ludwig & Leng, 1997). The neuronal data from these studies were not included in the population study because of the differences in surgical procedures that might have affected neuronal excitability. In these experiments, the recording micropipette was placed through the centre of the probe loop. Artificial cerebrospinal fluid (aCSF; pH 7.2, composition (in mM): 138 NaCl, 3.36 KCl, 9.52 NaHCO$_3$, 0.49 Na$_2$HPO$_4$, 2.16 urea, 0.49 NaH$_2$PO$_4$, 1.26 CaCl$_2$ and 1.18 MgCl$_2$) was dialysed at 3 µL min$^{-1}$, and, after recording basal activity, was changed to aCSF containing 2 mM bicuculline (Sigma-Aldrich) (Leng et al., 2001; Ludwig & Leng, 2000). Drugs administered in this way penetrate only a short distance into the brain: the concentrations achieved 0.5–1 mm below the surface are about four orders of magnitude below the dialysate concentration over this duration of infusion (Ludwig & Leng, 1997). Randle et al. (1986) give 1.4 µM as the concentration of bicuculline necessary for 50% inhibition of GABA-mediated IPSPs in magnocellular neurones, and so the dose administered by dialysis was expected to produce concentrations within the supraoptic nucleus at the low end of the effective range.

In some of these experiments, the transpharyngeal exposure of the supraoptic nucleus was extended rostrally and medially to uncover the rostral extent of the optic chiasm, and a concentric bipolar stimulating electrode (SNEX-100; Clarke Electromedical Instruments) was placed on the region of the organum vasculosum of the lamina terminalis (OVLT). Supraoptic neurones respond to electrical stimuli applied to the OVLT (0.1–1 mA peak-to peak matched biphasic 1 ms pulses) with a mixture of excitation and inhibition that varies between neurones and with stimulus intensity (Sabatier & Leng, 2006).

### Acute hyperosmotic stimulation

In the late afternoon of each experiment, one cell was selected for testing with I.V. infusion of pre-warmed 2 M NaCl at 26 µL min$^{-1}$ (Leng et al., 2001); over 30 min, which delivers 1.56 mmol NaCl. In urethane-anaesthetized rats, there is minimal urine production, and the infused salt remains in the body (Severs et al., 1981). The total body water of a 350g rat is ~245 mL (Culebras et al., 1977) and, after excess Na$^+$ intake, osmolality reaches equilibrium through redistribution of body water and electrolytes between compartments. An excess intake of 1.56 mmol NaCl in the fluid compartment thus implies an elevation of ~6.5 mM [Na$^+$] in plasma in the steady state (i.e. an increase in osmolality of ~13 mOsm kg$^{-1}$). Leng et al. (2001) measured plasma [Na$^+$] when infusing 2 M NaCl at various rates and reported that plasma [Na$^+$] increased by 6.1 mM per mmol infused, close to the level expected from these theoretical considerations.

Each cell was recorded for 10–15 min, then the rat was given an I.V. injection of CCK, and the response observed over 15–30 min to allow activity to return to a stable level. A new basal rate was then determined over

5 min of stationary activity for continuously firing neurones or 10 min for phasic cells. The i.v. infusion pump was switched on, and the onset of the stimulus was calculated as beginning 5 min later, to allow for the (measured) lag time in the infusion line. The infusion was stopped after 35 min (i.e. after 30 min of hypertonic infusion). Recordings were continued for up to 60 min after the onset of infusion. The firing rate for each cell was calculated in 30 s bins, and expressed as the mean change from basal by subtracting the basal rate from all bin values.

### Hypovolemia

Polyethylene glycol (PEG) (molecular weight 3350; Sigma-Aldrich) dissolved in 0.9% NaCl at room temperature (30% w/v) was pre-warmed to ~37°C, and injected i.p. at 20 mL kg$^{-1}$ to develop an acute hypovolemia (Kondo et al., 2004). Such injections produce a slowly progressing hypovolemia as indicated by measurements of haematocrit (Dunn et al., 1973) with no significant change in plasma osmolality, and vasopressin secretion is stimulated exponentially, with no significant increase until plasma volume has fallen by at least 7%. As reported by Dunn et al. (1973), at 30 min after i.p. injection of this dose of PEG, plasma volume was reduced by 14%. This, by their observed relationship between volume depletion and vasopressin secretion, corresponds to an expected increase in vasopressin secretion from 2.3 to ~10 pg mL$^{-1}$.

We recorded from vasopressin cells for ≥10 min of stable basal activity before PEG injection, and subsequently for up to 70 min. We quantified the response of each cell as the difference between the basal rate and the rate 30–45 min after PEG. We tested only one cell in any animal.

### Plasma [Na$^+$], drinking behaviour, plasma and pituitary vasopressin content

Eight euhydrated rats and eight rats salt-loaded for 7 days were used to measure plasma [Na$^+$], and pituitary and plasma vasopressin. Rats were anaesthetized with urethane and a femoral vein was cannulated for removal of ~1 mL of blood. Whole blood was collected into EDTA-coated tubes that were gently shaken to mix the blood and the anticoagulant. The tubes were kept on ice until processing. Then, tubes were placed in a 4°C centrifuge (BRK5424 rotor; Centurion Scientific Ltd, Chichester, UK) and centrifuged at 906 × *g* for 10 min at 4°C. Plasma was removed from the pellet using a micropipette, aliquoted into 100 μL tubes (Part No. 667201; CELLSTAR® tubes; Grainer Bio-One, Kremsmünster, Austria) and stored at –20°C. Then rats were decapitated and the neurointermediate lobes extracted. The lobes

were sonicated in 0.5 mL of distilled water for 1 min intervals using a probe sonicator (MSE Soniprep 150 Ultrasonic Disintegrator; Sanyo, Osaka, Japan). The homogenates were frozen and thawed three times before being centrifuged at 906 × *g* for 5 min at 4°C. The supernatant was aliquoted at 100 μL into capped tubes and kept at –20°C until assayed for vasopressin content.

Plasma vasopressin was measured in extracted samples by a sensitive and selective radioimmunoassay (detection limit: 0.1 pg sample$^{-1}$; RIAgnosis, Regensburg, Germany) (Landgraf et al., 1995). Plasma [Na$^+$] was measured in triplicate by flame photometry (Jenway Flame Photometer, FF-200 series; Cole-Parmer, Stone, UK).

Another 12 rats were used to study the effects of salt loading on body weight and fluid intake. Each day for 8 days, the rats were weighed and their fluid intake over the preceding 24 h was measured. Five of the rats had water available throughout; for the other seven rats, water was replaced with 2% NaCl.

### Literature search and analysis

We searched the published literature in the Web of Science™ Core Collection (https://clarivate.com/products/web-of-science), using search terms that identified studies in rats involving drinking 2% NaCl for at least 2 days, supplemented by searching the reference lists of retrieved papers. Most of the retrieved studies presented the relevant data in graphs, from which we extracted numbers using an online tool (graphreader.com). We aggregated data on plasma osmolality and [Na$^+$], plasma vasopressin and oxytocin, haematocrit, and pituitary hormone content, and we report study means (i.e. the mean of the means reported in each study) with the SD and 95% confidence limits.

### Quantification and statistical analysis

We collected data on multiple variables from three cell types in 37 salt-loaded rats, 41 rehydrated rats and 68 euhydrated rats, recording from one to 10 cells in each rat. In euhydrated rats, the main source of variability was between cells rather than between animals (see below), and so we used the cell as the unit of assessment. Our approach to significance testing was conservative, because of the many potential comparisons, because of doubt about the appropriate unit of assessment, and because the use of hypothesis-testing statistics is questionable where hypotheses cannot be specified fully in advance. We quote the mean ± SD and 95% confidence limits (CI) for all data in the Results, and draw explicit conclusions sparsely, and only when the CI limits do not overlap between compared groups.

We are making all primary data analysed in this study openly available; specifically, electrophysiological data for ~500 identified supraoptic neurones comprising records of the arrival times of spikes over 15–120 min, with metadata specifying the timing and nature of interventions. We also make available our analyses of these data: the ISI histograms and hazard functions and responses to CCK and hypertonic saline, etc., and our summaries of the associated secondary data (https://github.com/HypoModel/MagnoNeuroDat).

## Results

### Electrophysiological studies

**CCK-induced inhibition of vasopressin cells involves GABA.** The most robust way of distinguishing between oxytocin and vasopressin cells in male rats *in vivo* is by their responses to i.v. CCK. Oxytocin cells are excited via a direct projection from A2 noradrenergic neurones; most vasopressin cells are inhibited, but this pathway is undetermined. If GABA is involved, and if the response of vasopressin cells to GABA is reversed after salt loading, then the CCK test will be unreliable. Accordingly, we tested whether the inhibition of vasopressin cells involves actions of GABA in the supraoptic nucleus.

We used a dialysis probe to apply the GABA antagonist bicuculline directly to the supraoptic nucleus; this increases the firing rate of all supraoptic neurones, and blocks their inhibitory response to electrical stimulation of the arcuate nucleus (Ludwig & Leng, 2000) and the OVLT (Leng et al., 2001) (Fig. 1*A*–*D*). In five experiments, we tested one neurone inhibited by CCK; in each case, bicuculline attenuated the inhibition, and, in four, the attenuation was reversed on washout (Fig. 1*E* and *F*), with the fifth neurone being lost before washout. Thus, the inhibition involves GABA actions in the supraoptic nucleus. Accordingly, using CCK to identify cells in salt-loaded rats required caution, and so we begin by describing the whole population of recorded supraoptic neurones.

**Unchanged responses to CCK.** In euhydrated rats, cells excited by CCK are mainly oxytocin cells, whereas most vasopressin cells are inhibited (Leng et al., 1991); we characterized cells as unresponsive if their firing rate changed by $\leq 0.3$ spikes s$^{-1}$. Response magnitudes varied between cells, but the temporal profile was relatively consistent; responses began within 30 s after injection, peaked at 2–3 min, and returned to the basal level by ~15 min (Fig. 2*A*).

In 182 recordings from 68 euhydrated rats, 28 out of 44 VP$_{phasic}$ cells were tested with CCK: 20 were inhibited, four were excited and four were unresponsive. Of continuously firing cells, 46 were excited by a mean $\pm$ SD of $+1.1 \pm 0.72$ spikes s$^{-1}$ and thereby identified as oxytocin cells; 79 were inhibited by $3.3 \pm 1.9$ spikes s$^{-1}$; and thereby identified as vasopressin cells (VP$_{cont}$ cells); 13 cells were unresponsive and their identity was considered to be undetermined. Thus, of the 169 identified neurones, 27% were oxytocin cells and 73% were vasopressin cells.

In 160 recordings from 37 salt-loaded rats, 53 out of 71 VP$_{phasic}$ cells were tested with CCK; 45 were inhibited, four were excited and four were unresponsive. Of continuously firing cells, 45 were excited ($+0.9 \pm 0.5$ spikes s$^{-1}$) and 28 were inhibited ($-4.0 \pm 3.4$ spikes s$^{-1}$); 16 cells were unresponsive. If the excited cells are oxytocin cells and the inhibited cells are vasopressin cells, 31% of 'identified' neurones were oxytocin cells and 69% were vasopressin cells.

In 134 recordings from 41 rehydrated rats, 31 out of 38 VP$_{phasic}$ cells were tested with CCK; 21 were inhibited, four were excited and six were unresponsive. Of continuously firing cells, 28 were excited ($+0.9 \pm 0.7$ spikes s$^{-1}$) and 52 inhibited ($-3.1 \pm 2.2$ spikes s$^{-1}$); 16 cells were unresponsive. Ift the excited cells are oxytocin cells and the inhibited cells are vasopressin cells, 24% of 'identified' neurones were oxytocin cells and 76% were vasopressin cells.

Thus, in all groups, the proportions of 'identified' oxytocin cells were similar and close to the ratio expected from immunocytochemical studies (Rhodes et al., 1981). Their response to CCK was independent of the basal firing rate (not shown) and was similar in the three groups, as was the temporal profile of the response (Fig. 2*B*). The inhibitory responses of putative vasopressin cells were also similar across groups (Fig. 2*B*). Thus, the responses to CCK appear to be unaffected by salt loading, and hence can be used to identify cells with the same confidence as in euhydrated rats.

**Unchanged responses to phenylephrine (PE).** In three euhydrated rats, five VP$_{cont}$ cells were challenged with i.v. PE, and we measured their change in firing rate in the 5 min after injection. As expected, all five were inhibited to $44 \pm 14\%$ of the basal rate (CI = 31–56). In eight salt-loaded rats, we tested four VP$_{cont}$ cells and six VP$_{phasic}$ cells. One VP$_{phasic}$ cell increased its firing rate by 11%; the other nine cells were all inhibited. The ten vasopressin cells combined were inhibited to $46 \pm 11\%$ of their basal rate (CI = 25–66). Thus, we found no indication of lower responsiveness to PE in salt-loaded rats (Fig. 2*C*–*E*).

The responses of vasopressin cells to CCK were similar to their responses to PE, which inhibits cells as a consequence of the large increase in blood pressure that it evokes (Harris, 1979). Accordingly, in two experiments, we checked that, although PE evoked the expected large, increase in arterial blood pressure, CCK injections had no such effect (Fig. 2*F*).

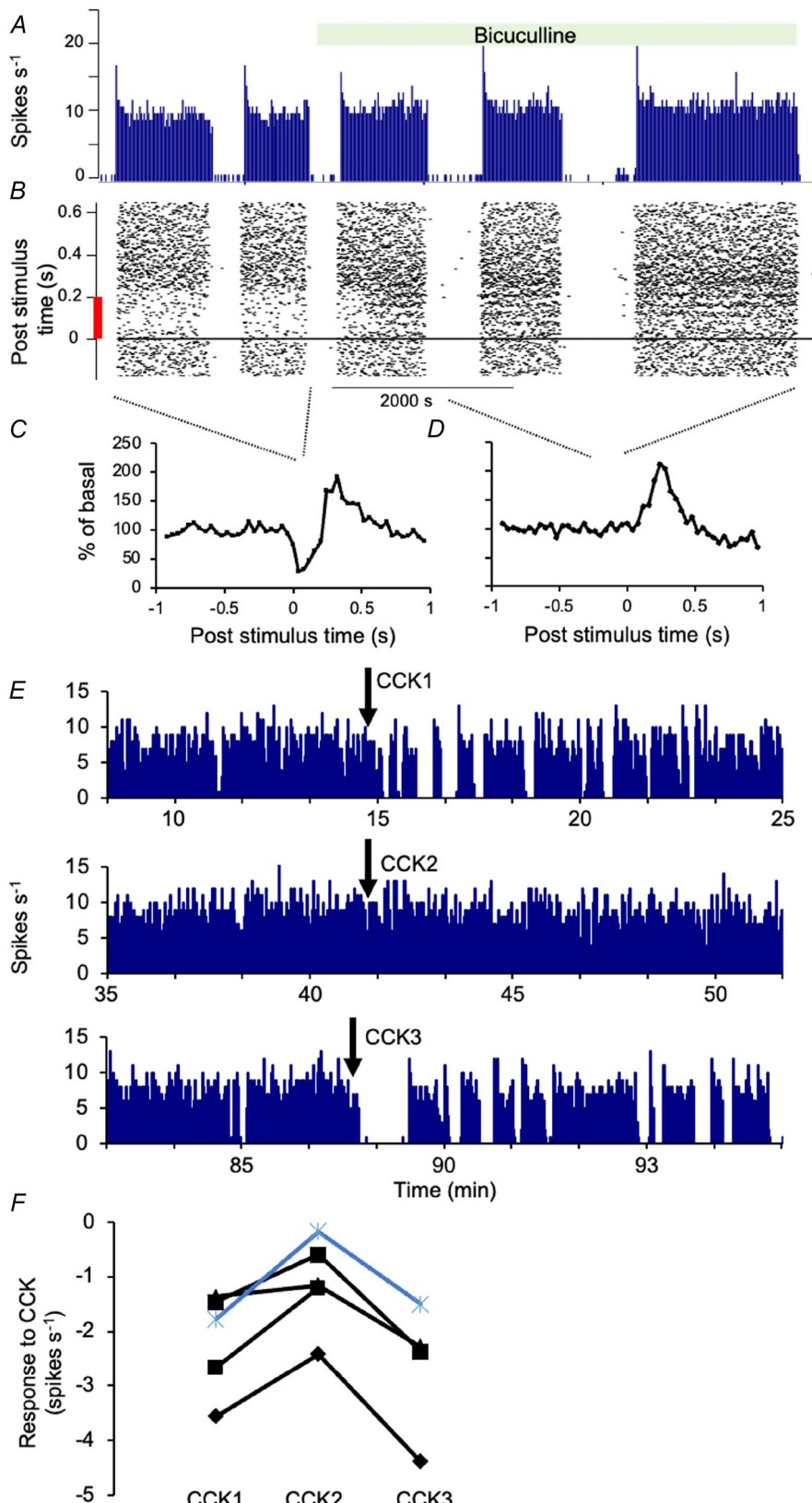

**Figure 1. Effects of bicuculline on CCK-induced inhibition of vasopressin cells**
*A*, the firing rate of a VP$_{phasic}$ cell in 1 s bins. The OVLT was stimulated every 5 s throughout the recording, evoking a response comprising an early inhibition and later excitation. Bicuculline (2 mM) was applied to the SON

by retrodialysis as indicated by the green bar. *B*, corresponding raster of the response to OVLT stimulation: the inhibitory effect is apparent as a sparsity of spikes following each stimulus, and is abolished by bicuculline. *C* and *D*, peristimulus-time histograms (spike counts in 40 ms bins as a percentage of the average pre-stimulation counts) corresponding to responses during the first two phasic bursts in (*A*) (orange dotted lines) and two bursts during bicuculline (blue dotted lines). The inhibition apparent in (*C*) is completely blocked in (*D*). *E*, a VP$_{phasic}$ cell exposed to bicuculline (spikes in 1 s bins, extracts from a 6200 s recording). CCK was administered before (CCK1), during (CCK2) and after washout of bicuculline (CCK3). *F*, responses of four vasopressin cells recorded through the protocol in (*E*), and a fifth cell lost before washout. Responses were calculated as the difference between the basal rate and that in the 5 min after CCK. Each line connects data from a single cell; the cell in (*E*) is the blue line. For each cell, the inhibitory response was lowest in the presence of bicuculline.

**Table 1. Mean ± SD firing rates of supraoptic neurones with 95% CIs**

|  | Euhydrated | Salt-loaded | Rehydrated |
|---|---|---|---|
| All cells (spikes s$^{-1}$) | 5.1 (2.4); CI 4.8–5.5 $n = 182$ | 6.0 (3.0); CI **5.5–6.4**[a] $n = 160$ | 5.3 (3.0); CI 4.8–5.8 $n = 134$ |
| Oxytocin (spikes s$^{-1}$) | 3.6 (1.8); CI 3.1–4.1 $n = 46$ | 4.3 (2.6); CI 3.6–5.1 $n = 45$ | 3.1 (2.3); CI 2.3–4.0 $n = 28$ |
| VP$_{cont}$ (spikes s$^{-1}$) | 6.4 (2.2); CI 5.9–6.9 $n = 81$ | 8.4 (3.2); CI **7.1–9.6**[a] $n = 28$ | 7.2 (2.7); CI 6.5–8.0 $n = 52$ |
| VP$_{phasic}$ (spikes s$^{-1}$) | 4.3 (1.7); CI 3.8–4.8 $n = 42$ | 5.9 (2.4); CI **5.4–6.5**[a] $n = 71$ | 4.9 (2.0); CI 4.2–5.5 $n = 38$ |
| All VP cells (spikes s$^{-1}$) | 5.7 (2.3); CI 5.3–6.0 $n = 123$ | 6.6 (2.9); CI **6.0–7.2**[a] $n = 99$ | 6.2 (2.7); CI 5.7–6.8 $n = 90$ |
| VP$_{phasic}$ Intraburst (spikes s$^{-1}$) | 5.5 (1.6) CI 5–6.1 $n = 38$ | 8.1 (2.1); CI **7.6–8.6**[ab] $n = 60$ | 6.1 (1.9); CI 5.4–6.7 $n = 35$ |
| Burst duration (s) | 66 (50), CI 51–82 $n = 38$ | 79 (53); CI 66–93 $n = 60$ | 76 (53); CI 59–94 $n = 35$ |
| Silence duration (s) | 22 (25), CI 14–30 $n = 38$ | 24 (15); CI 20–28 $n = 60$ | 18 (13); CI 14–22 $n = 35$ |

Mean ± SD firing rates of supraoptic neurones with 95% CIs. The CIs for data in salt-loaded rats in bold with suffix 'a' do not overlap with the CIs from euhydrated rats, those with suffix 'b' do not overlap with CIs from rehydrated rats.

## Changes in firing rates and patterning

Supraoptic neurones were recorded from 68 euhydrated rats, 37 salt-loaded rats and 41 rehydrated rats. In euhydrated rats, 182 neurones fired at a mean ± SD of 5.1 ± 2.4 spikes s$^{-1}$ (median 4.8 spikes s$^{-1}$); the animal mean, calculated from the first neuron recorded in each animal, was 5.1 ± 2.5 spikes s$^{-1}$. From the similarity of the variances, we inferred that the main source of variability was between neurones rather than between animals.

In salt-loaded rats, 160 neurones fired at 6.0 ± 3.0 spikes s$^{-1}$ (median 5.6 spikes s$^{-1}$) with an animal mean of 6.1 ± 3.8 spikes s$^{-1}$). In rehydrated rats, 134 neurones fired at 5.3 ± 3.0 spikes s$^{-1}$ (median 5.2 spikes s$^{-1}$), and the animal mean was 5.6 ± 2.8 spikes s$^{-1}$ (Fig. 3*A* and *B*).

More neurones fired phasically in salt-loaded rats (VP$_{phasic}$ cells; 44%; 71/160) than in either euhydrated rats (23%; 42/182) or rehydrated rats (28%; 38/134) (Fig. 3*C*).

## Elevated firing rates of all cell types (Table 1)

Oxytocin cells, VP$_{phasic}$ cells and VP$_{cont}$ cells all fired faster in salt-loaded rats than in euhydrated rats (Fig. 3*B*). In VP$_{phasic}$ cells, the mean intraburst firing rate was higher in salt-loaded rats (8.1 ± 2.1 spikes s$^{-1}$) than in euhydrated rats (5.5 ± 1.6 spikes s$^{-1}$), with no marked difference in mean burst length or interburst durations. Combining VP$_{phasic}$ and VP$_{cont}$ cells, vasopressin cells fired at 5.7 ± 2.3 spikes s$^{-1}$ in euhydrated rats and at 6.6 ± 2.9 spikes s$^{-1}$ in salt-loaded rats. Oxytocin cells fired at 3.6 ± 1.8 spikes s$^{-1}$ in euhydrated rats, and 4.3 ± 2.6 spikes s$^{-1}$ in salt-loaded rats. Rates in rehydrated rats were consistently lower than in salt-loaded rats and generally close to those in euhydrated rats.

## Oxytocin cells fire more regularly

The CV measures the dispersion of ISIs: random distributions of intervals have a CV of 1; more regular distributions have a CV <1; and burst-patterned distributions (as in VP$_{phasic}$ cells) have a CV >>1.

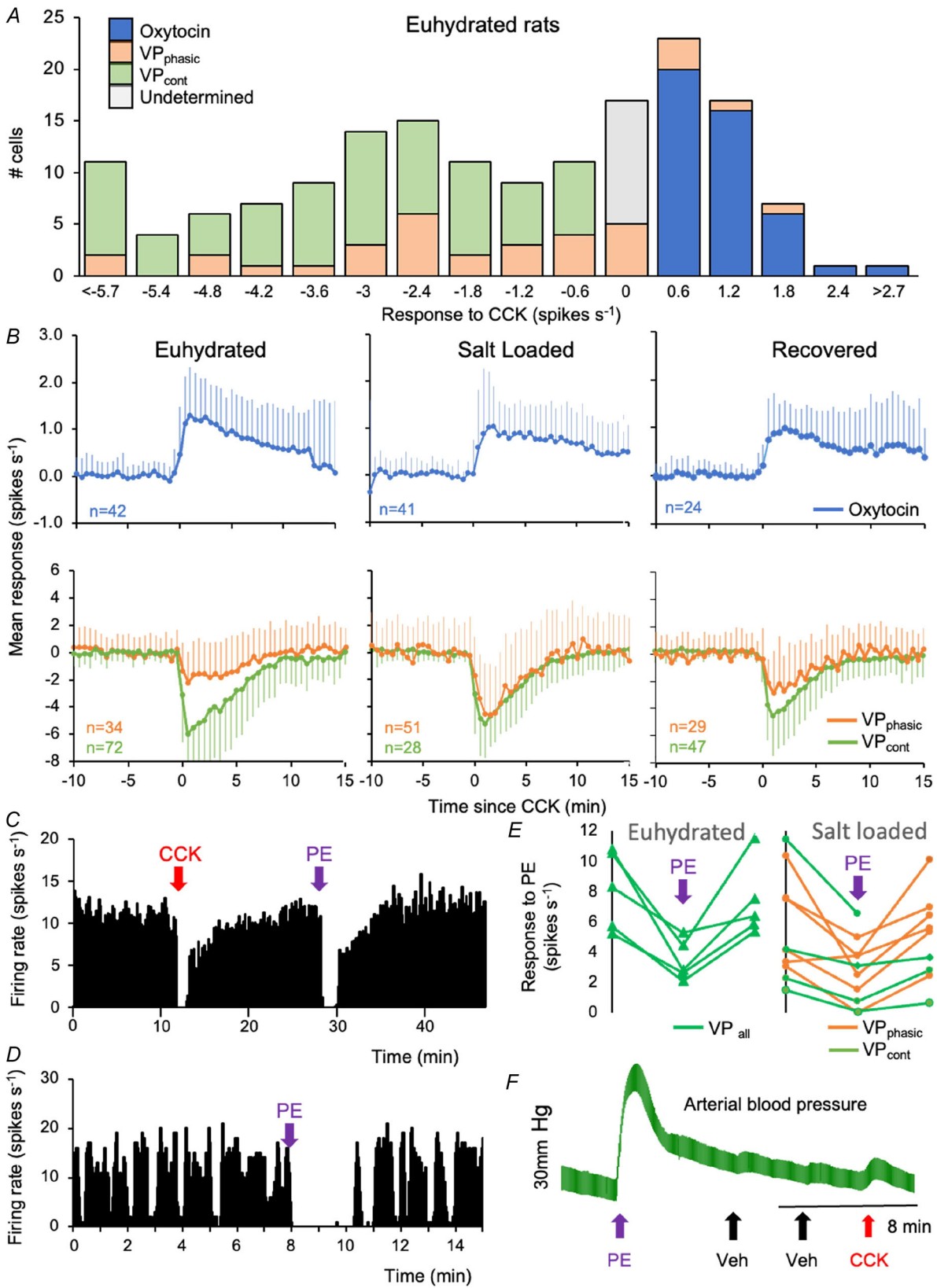

**Figure 2. Responses of supraoptic neurones to CCK and phenylephrine (PE)**
*A*, distribution of response magnitudes to CCK in euhydrated rats (mean change in firing rate over 5 min, bin width 0.6 spikes s$^{-1}$). Most VP$_{phasic}$ cells (orange), were inhibited. Continuously firing cells excited by >0.3 spikes

$s^{-1}$ were classed as oxytocin cells (blue), whereas those inhibited by >0.3 spikes $s^{-1}$ were classed as $VP_{cont}$ cells (green). Cells that changed by <0.3 spikes $s^{-1}$ (grey) were of undetermined identity. B, mean ± SD responses to I.V. CCK. The responses of both oxytocin cells and vasopressin cells are unaffected by salt loading. Spike activity of (C) a $VP_{cont}$ cell and (D) a $VP_{phasic}$ cell from a euhydrated rat, in 1 s bins. $VP_{cont}$ cells, identified as inhibited by CCK, were also inhibited by PE. E, responses of individual vasopressin cells to PE, showing, for each cell tested, the basal rate, the rate in the 5 min after PE, and the rate after recovery. Each line connects data from a single cell; one cell in salt-loaded rats was lost without recovery. F, arterial blood pressure in a euhydrated rat during I.V. injection of PE, isotonic saline (Veh) and CCK; PE produces a large, transient increase.

**Table 2. Characteristics of ISI distributions in the different cell types (Mean ± SD; 95% CI)**

| | | Euhydrated | Salt-loaded | Rehydrated |
|---|---|---|---|---|
| CV | Oxytocin | 0.88 ± 0.14; 0.84–0.92 | 0.83 ± 0.21; 0.77-0.90 | 0.91 ± 0.20; 0.83–0.98 |
| | $VP_{cont}$ | 1.27 ± 0.42; 1.18–1.37 | **1.00 ± 0.32; 0.89–1.12[a]** | 1.08 ± 0.32; 1.00–1.17 |
| Mode (ms) | Oxytocin | 56.8 ± 19.4; 50.9–62.7 | **71.4 ± 13.3; 67.5–75.3[a]** | 58.3 ± 23.9; 56.2–69.2 |
| | $VP_{cont}$ | 36.9 ± 16.1; 33.4–40.5 | 46.5 ± 20.6; 38.9–54.1 | 47.1 ± 20.3; 41.6–52.6 |
| | $VP_{phasic}$ | 50.4 ± 17.4; 45.1–55.6 | 45.2(14.3; 41.9–48.5 | 49.3(13.9; 44.9–53.7 |
| Skewness | Oxytocin | 1.7 ± 0.7; 1.5–2.0 | 1.7 ± 0.7; 1.5–1.9 | 1.7 ± 0.6; 1.4–1.9 |
| | $VP_{cont}$ | 3.9 ± 1.3; 3.2–3.8 | 3.4 ± 1.3; 3.0–3.9 | 3.1 ± 1.1; 2.8–3.4 |
| | $VP_{phasic}$ | 2.9 ± 0.8; 2.7–3.2 | **3.5 ± 0.9; 3.3–3.7[a,b]** | 2.8 ± 0.9; 2.5–3.1 |
| Kurtosis | Oxytocin | 3.9 ± 4.3; 2.6–5.1 | 3.1 ± 3.3; 2.1–4.0 | 2.8 ± 2.3; 1.9–3.7 |
| | $VP_{cont}$ | 15.1 ± 11.8; 12.6–17.7 | 14.0 ± 10.7;10.9–17.9 | 11.5 ± 8.3; 9.2–13.7 |
| | $VP_{phasic}$ | 10.1 ± 7.4; 7.8–12.3 | **13.5 ± 8.9; 11.4–15.6[b]** | 8.8 ± 6.7; 6.6–10.9 |
| Hazard | Oxytocin | 3.6 ± 3.0; 2.7–4.5 | **1.8 ± 2.2; 1.2–2.5[a]** | 3.2 ± 2.4; 2.3–4.1 |
| | $VP_{cont}$ | 12.1 ± 8.9; 10.1–14.0 | 12.3 ± 10.4; 8.4–16.2 | 9.2 ± 6.6; 7.4–11.0 |
| | $VP_{phasic}$ | 10.7 ± 5.4; 9.1–12.4 | **13.2 ± 8.3; 11.2–15.1[b]** | 8.1 ± 5.7; 6.3–10.0 |

Oxytocin cells in salt-loaded rats have a longer mode than in euhydrated rats, and a lower peak hazard, consistent with a larger HAP. $VP_{phasic}$ cells in salt-loaded rats have a more skewed ISI distribution and a larger peak hazard than in rehydrated rats, both consistent with a larger DAP. The bold data in the salt-loaded column have CIs that do not overlap with the CIs in euhydrated rats (a) and/or rehydrated rats (b).

The mean CV in oxytocin cells was <1 in all groups, whereas $VP_{cont}$ cells had a CV >1 (Fig. 3D). In salt-loaded rats, the CV of $VP_{cont}$ cells was lower than in euhydrated rats (1.0 ± 0.3 vs. 1.27 ± 0.42), reflecting more regular firing (Table 2).

### ISI distributions differ between cell types; changes after salt loading

A unimodal distribution can be described by its skewness and kurtosis. Skewness measures its symmetry; positive values reflect a right skew and negative values a left skew; kurtosis measures 'tailedness' – a distribution with high kurtosis has many outlying values. All ISI distributions were strongly right skewed, more so for vasopressin cells than oxytocin cells (Fig. 3E). In all groups, kurtosis was positively correlated with skewness. Although there were differences between cell types, the only conspicuous difference between animal groups was that ISI distributions of $VP_{phasic}$ cells were more skewed in salt-loaded rats (Table 2).

However, there were differences in another signature statistic of the ISI distribution: the mode. In all groups, the mean mode was larger in oxytocin cells than in $VP_{cont}$

cells, and it was larger still in salt-loaded rats (71 ± 19 ms vs. 56 ± 13 ms in euhydrated rats) (Fig. 3E and Table 2). This suggests that the HAP is enhanced in salt-loaded rats, which would also account for the smaller CV. In oxytocin cells, the ISI distribution can be modelled by assuming that synaptic inputs arrive randomly, while the chance of triggering a spike is subject to a HAP that decays exponentially (Leng et al., 2017). If one of two cells firing at the same rate has a larger HAP, then its spikes will be compressed within a narrower range, giving a smaller CV. Consistent with this, the CV was inversely correlated with the mode in both oxytocin cells and $VP_{cont}$ cells.

### Attenuated responses to acute hyperosmotic stimulation

In 19 euhydrated rats, 21 neurones were tested for their response to I.V. hypertonic saline (six oxytocin; eight $VP_{phasic}$; seven $VP_{cont}$; including two double recordings, one of two oxytocin cells and one of a $VP_{phasic}$ and a $VP_{cont}$ cell). In 20 salt-loaded rats, 23 neurones were tested (four oxytocin; 12 $VP_{phasic}$; four $VP_{cont}$; three undetermined; including three double recordings of $VP_{phasic}$ cells). In 18

rehydrated rats, 19 neurones were tested (three oxytocin; four $VP_{phasic}$; nine $VP_{cont}$; two undetermined; includes one double recording of $VP_{cont}$ cells).

In each experimental group, the firing rate increased progressively during the infusion (Fig. 4). The response was weakest in salt-loaded rats (Fig. 4*F*), and this was not an effect of the higher basal rate *per se*: in all groups, the response was independent of basal firing rate, and was apparently linear throughout the infusion but with a lower slope in salt-loaded rats (0.77 spikes $s^{-1}$ per mmol infused vs. 1.92 spikes $s^{-1}$ per mmol infused in euhydrated rats; linear trend lines fitted to mean changes from basal)

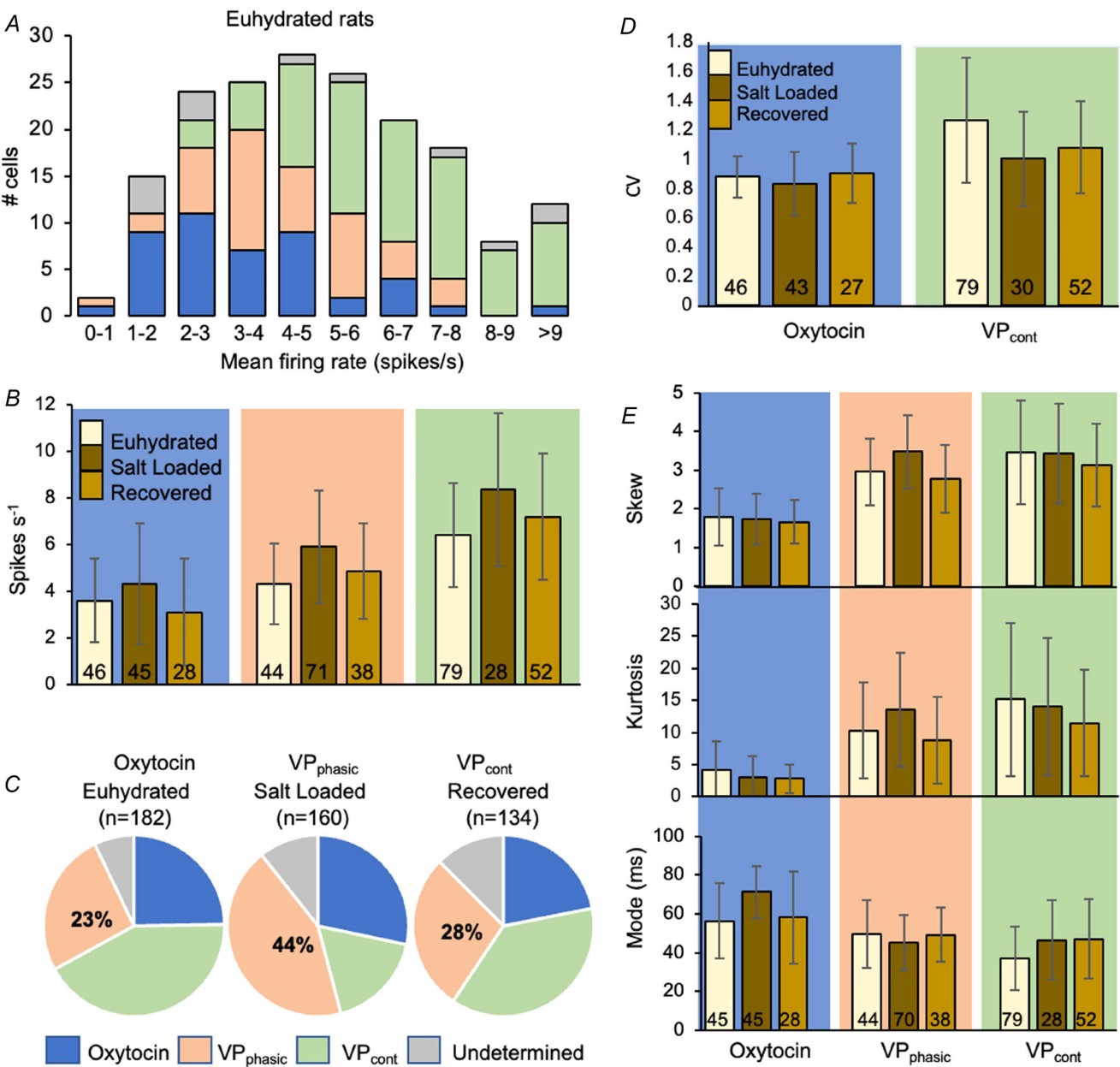

**Figure 3. Analysis of firing rate of supraoptic neurones; comparison between euhydrated, salt-loaded and rehydrated rats**

*A*, distribution of basal firing rates (spikes $s^{-1}$) of supraoptic neurones in euhydrated rats. *B*, mean ± SD firing rates of supraoptic neurones in euhydrated, salt-loaded and rehydrated rats. Oxytocin cells, $VP_{cont}$ cells and $VP_{phasic}$ cells all fire ~1 spikes $s^{-1}$ faster after 7 days of salt loading and recover after rehydration. *C*, more neurones fired phasically in salt-loaded rats than in euhydrated or rehydrated rats. *D*, the CV of oxytocin cells is lower than that of $VP_{cont}$ cells in all experimental groups. *E*, skew (top), kurtosis (middle) and mode (bottom) of ISI histograms for each cell type in each group. The columns in (*B*), (*D*) and (*E*) show the mean ± SD; for full statistical details, see Tables 1 and 2).

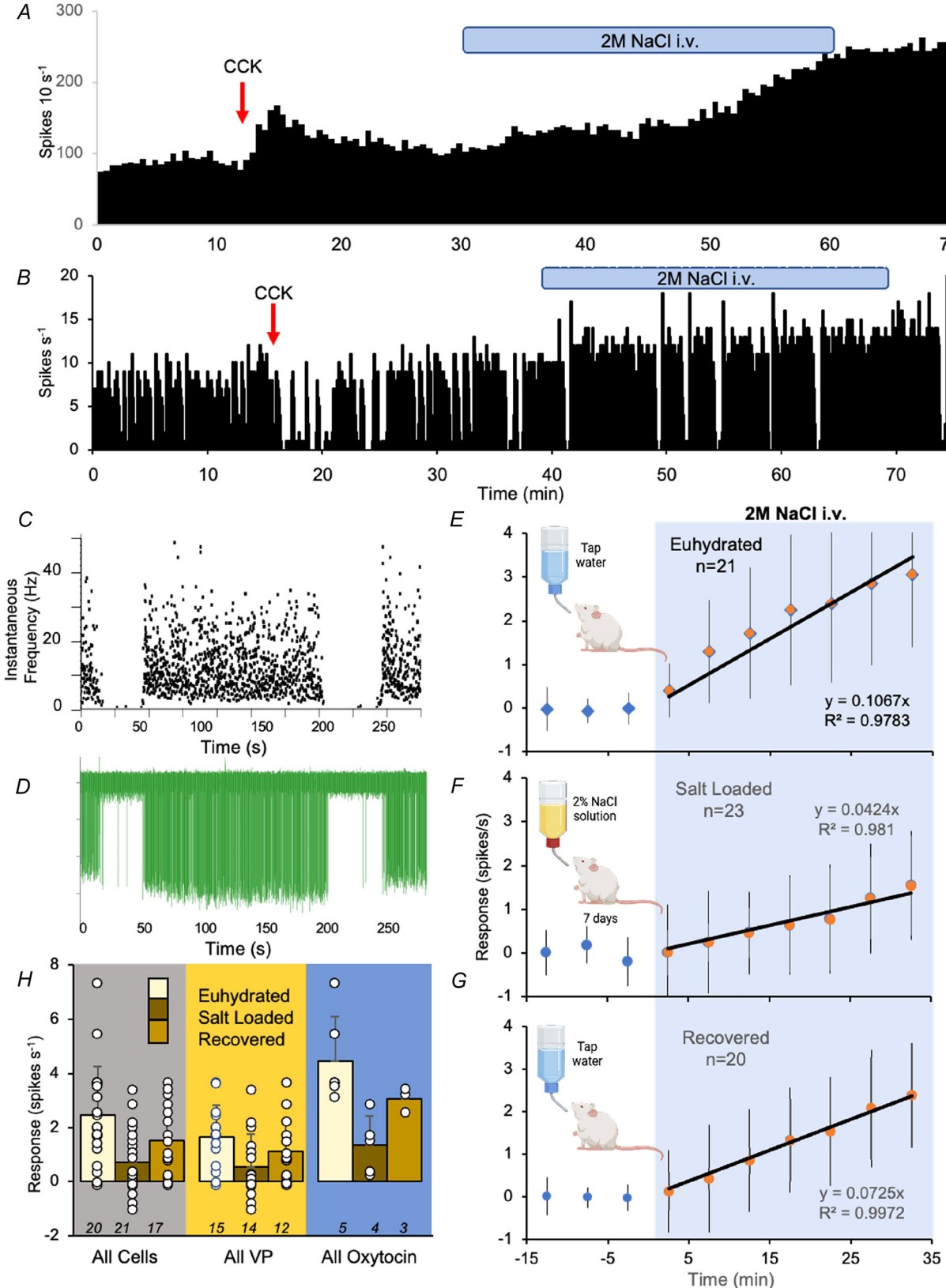

**Figure 4. Responses to acute osmotic stimulation**
Spike activity of (*A*) an oxytocin and (*B*) a VP$_{phasic}$ cell in euhydrated rats, in 1 s bins; CCK produced a transient excitation (*A*) and inhibition (*B*) respectively. After basal level activity had returned, 2 M NaCl was infused I.V. for 30

min (blue bar); both cells were progressively activated. *C*, instantaneous frequency plot (*D*) and extract of voltage trace showing individual spikes from a burst of activity from the cell shown in (*B*). Data from euhydrated (*E*), salt-loaded (*F*) and rehydrated (*G*) rats showing mean ± SD responses to the infusion (blue shaded region). Data are differences in firing rate (in 5 min bins) from the basal rate. The trendline is the best linear fit to the mean changes given a fixed intercept at 0. The equations and the $r^2$ value of the fits are indicated. *H*, mean ± SD responses of oxytocin cells, VP$_{phasic}$ cells, and VP$_{cont}$ cells in each experimental group. For full statistical details, see Table 3.

**Table 3. Mean ± SD responses (spikes s$^{-1}$) of supraoptic neurones after 20–30 min infusion of hypertonic saline**

| | | Euhydrated | | Salt-loaded | | Rehydrated | |
|---|---|---|---|---|---|---|---|
| | | | *n* | | *n* | | *n* |
| Oxytocin | Basal, mean ± SD | 3.7 ± 1.7 | 6 | 5.1 ± 4.0 | 5 | 2.9 ± 1.8 | 3 |
| | Response, mean ± SD | 4.5 ± 1.6 | | 1.3 ± 1.2 | | 3.1 ± 0.4 | |
| | CI (95% confidence) | 3.2–5.8 | | **0.4–2.3[a]** | | 2.6–3.6 | |
| Vasopressin | Basal | 4.2 ± 2.3 | 15 | 5.1 ± 2.7 | 16 | 5.1 ± 2.4 | 14 |
| | Response | 1.7 ± 1.3 | | 0.6 ± 1.2 | | 1.1 ± 1.2 | |
| | CI | 1.1–2.3 | | **0–1.1[a]** | | **0.5–1.7** | |
| All | Basal | 4.1 ± 2.1 | 21 | 5.3 ± 2.9 | 23 | 5.0 ± 2.4 | 20 |
| | Response | 2.5 ± 1.8 | | 0.7 ± 1.2 | | 1.5 ± 1.3 | |
| | CI | 1.7–3.2 | | **0.2–1.2[a]** | | 0.9–2.1 | |

The 95% CIs for the responses of vasopressin cells, oxytocin cells and for all cells in salt-loaded rats do not overlap with CIs from euhydrated rats (bold data with suffix a); data for rehydrated rats are intermediate.

(Fig. 4*G*). The attenuated responsiveness was partially reversed in rehydrated rats (Fig. 4*G*).

We quantified the responses by comparing the basal rate to that 20–30 min after the onset of infusion (Table 3). The mean ± SD response was +2.5 ± 1.8 spikes s$^{-1}$ in euhydrated rats and +0.7 ± 1.2 spikes s$^{-1}$ in salt-loaded rats. For VP$_{phasic}$ cells and VP$_{cont}$ cells, their combined response was +1.7 ± 1.3 spikes s$^{-1}$ in euhydrated rats and +0.6 ± 1.2 spikes s$^{-1}$ in salt-loaded rats. Responses in rehydrated rats were intermediate.

### Unchanged responses to hypovolemia

We tested whether a reduced plasma volume would stimulate greater vasopressin cell activity in salt-loaded rats, as expected from the synergy between osmolality and hypovolemia (Kondo et al., 2004; Stricker & Verbalis, 1986). In seven euhydrated rats, two VP$_{phasic}$ cells and five VP$_{cont}$ cells were recorded for ≥45 min after I.P. PEG; six cells were excited and one inhibited for a mean ± SD response at 30–45 min of +0.4 ± 0.5 spikes s$^{-1}$ (CI = 0–0.8). In five salt-loaded rats, two VP$_{phasic}$ cells were excited (by 1.4 and 2.2 spikes s$^{-1}$) and three were inhibited (by 0.6, 1.1 and 1.2 spikes s$^{-1}$) for a mean response of +0.1 ± 0.8 spikes s$^{-1}$ (CI = 1.2–1.5). In six rehydrated rats, four VP$_{phasic}$ cells and two VP$_{cont}$ cells were excited by between 0.6 and 1.2 spikes s$^{-1}$ for a mean of 1.2 ± 0.8 spikes s$^{-1}$ (CI = 0.6–1.8). Thus, responses to PEG were varied, but while the two largest responses

were in salt-loaded rats, we saw no enhancement of overall responsiveness.

### Enhanced post-spike hyperexcitability in vasopressin cells

Comparing vasopressin cells between experimental groups is problematic because it appears from the above that many VP$_{cont}$ cells adopt a phasic firing pattern after salt loading. In doing so, they seem to adopt the normal characteristics of VP$_{phasic}$ cells, as VP$_{phasic}$ cells in all three experimental groups had similar ISI characteristics, but this suggests that the HAP may be enhanced after salt-loading in VP$_{cont}$ cells that become VP$_{phasic}$ cells.

In each group, the modal ISI was largely independent of the mean firing rate (Fig. 5). However, to test whether increased extracellular osmolality produced an increased HAP, we looked at how acute hyperosmotic stimulation affected hazard functions. In oxytocin cells, hazard functions begin with ≥15 ms of zero hazard reflecting a post-spike hyperpolarization (the HAP), rising to a maximum at 30–80 ms as the HAP decays (Leng et al., 2017). The mode of the ISI distribution is close to the time of maximum hazard, and hence is determined by the magnitude and time constant of the HAP. We found no evidence that acute osmotic stimulation changed the position of the maximum hazard in any cell that would indicate a change in the HAP (Fig. 5).

In vasopressin cells, hazard functions again begin with ≥15 ms of zero hazard, rising (more steeply than in

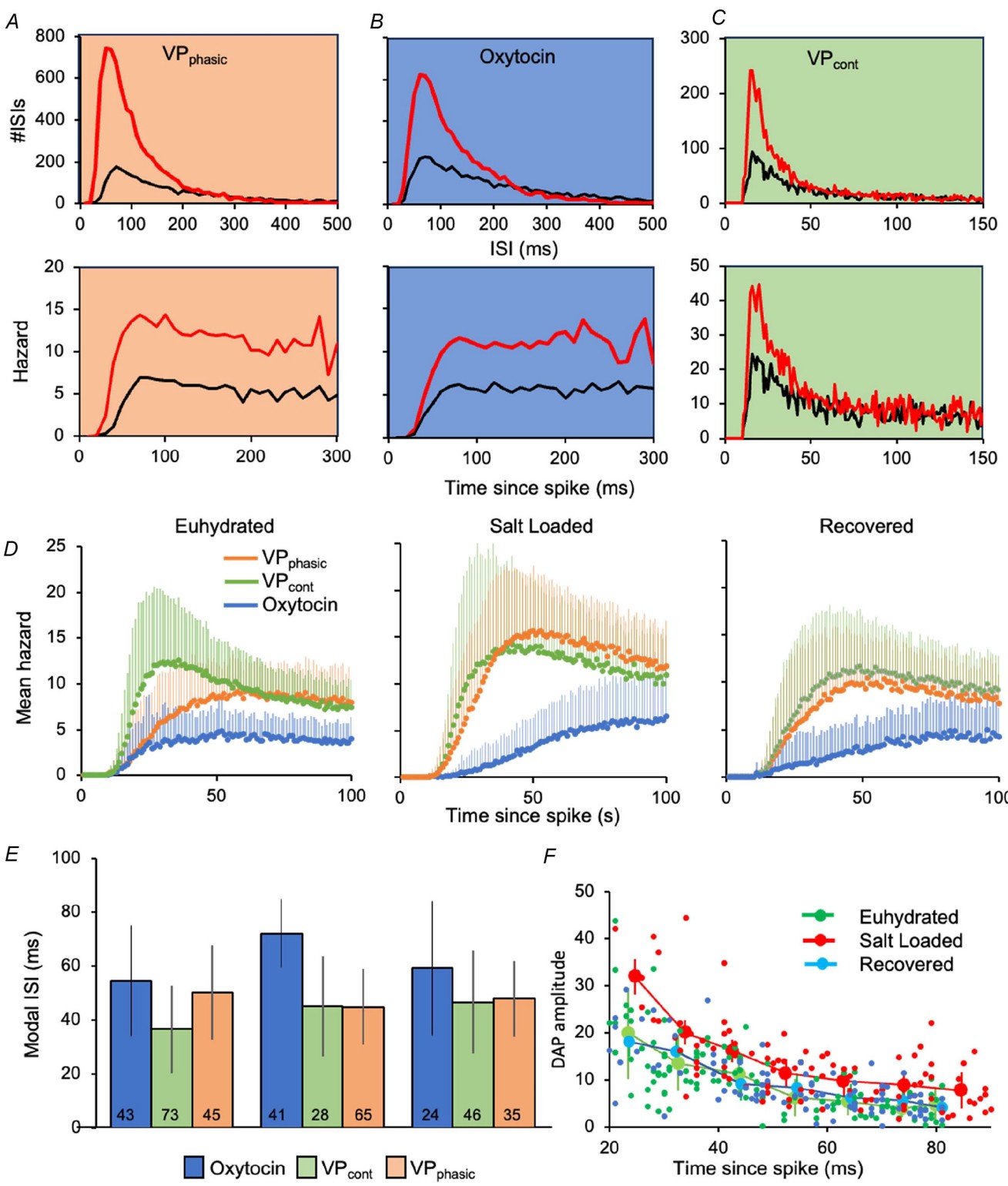

**Figure 5. ISI distributions**

*A*, ISI histograms (in 10 ms bins) of a VP$_{phasic}$ cell (shown in Fig. 4*B*). ISIs are plotted of 800 s of basal activity (blue line) and 800 s of activity after I.V. infusion of hypertonic saline (red line). Below, the corresponding hazard functions show a small DAP in both cases, and the mode of the ISI histogram is unchanged. *B*, equivalent histograms from an oxytocin cell in a euhydrated rat. The hazard functions, as is usual in oxytocin cells, show little sign of a DAP, and, again, there is no change in the mode after an increase in firing rate induced by I.V. infusion of hypertonic saline. *C*, similar histograms from a VP$_{cont}$ cell in a euhydrated rat, constructed in 1 ms bins to resolve the early

mode clearly; note the difference in time scales between (*C*) and (*A*)/(*B*). The hazard functions display a very large DAP, and again, the mode is unchanged after infusion. *D*, mean ± SD hazard functions shown as overlays of the three populations in each group on an expanded time scale. *E*, mean ± SD modal ISI in each group; the number of cells in each group is indicated in each column. *F*, inferred DAP from vasopressin cells. In each experimental group, data from $VP_{phasic}$ cells and $VP_{cont\ cells}$ were combined and gathered in bins by the position of the peak hazard. The mean ± SD peak hazard is shown for each bin, revealing an inverse relationship between these parameters. The data from euhydrated rats and rehydrated rats are indistinguishable, but salt-loaded rats show an enhanced peak hazard at all time points, indicating an enhanced DAP. Full data are available at: https://github.com/HypoModel/MagnoNeuroDat.

oxytocin cells) to a maximum at 30–50 ms. The hazard then falls, usually to a relatively steady 'asymptotic' state by ~150 ms (Fig. 5). This sequence can be attributed to the superposition of a large, short-lasting HAP and a smaller, longer-lasting DAP (Leng et al., 2017; MacGregor & Leng, 2012; Maicas Royo et al., 2016). The temporal profile is unique to each neuron and is relatively unaffected by changes in background firing rate.

A large hazard 'peak' was taken as evidence of a DAP, but its magnitude depends on the magnitudes and half-lives of both the DAP and the HAP. We saw little evidence of a DAP in most oxytocin cells; for them, the mean peak (in hazard units) was 3.6 ± 3.0 in euhydrated rats vs. 1.8 ± 2.2 in salt-loaded rats. The reduction in salt-loaded rats is consistent with the larger HAP inferred from changes in the mode. The peak was much larger in vasopressin cells, and similar between $VP_{phasic}$ cells and $VP_{cont}$ cells (Table 2). In euhydrated rats, it was 10.8 ± 8.0 in vasopressin cells combined (CI = 9.3–12.3) vs. 12.8 ± 8.9 in salt-loaded rats (CI = 11.1–14.6). Thus, the DAP might be enhanced in salt-loaded rats, but the CIs overlap. However, the HAP occludes the DAP in the early part of the hazard function; hence an enhanced HAP will delay the peak. Accordingly, we plotted the peak hazard against the time of peak, combining data from $VP_{phasic}$ cells and $VP_{cont}$ cells (Fig. 5*F*). The data for the euhydrated rats and rehydrated rats are closely superimposed, but those from salt-loaded rats are elevated at all time points, consistent with an elevation in DAP magnitude (with no change in half-life).

### Spike waveforms mirror hazard functions

Although extracellular unit recording is generally used only to detect spike occurrences, the spike waveform reflects features of the intracellular action potential (Gold et al., 2006). We constructed spike-triggered averages of >1000 spikes over periods with large negative-going spikes (Fig. 6), expecting that such waveforms would mainly reflect somatic currents (Gold et al., 2006). We normalized the averages to a spike height of 1 and inverted them to be consistent with the convention that depolarizing changes are positive-going.

The spikes were followed by a slowly decaying hyperpolarization that lasted for 30–80 ms, consistent with the

expected HAP, and these profiles showed a close alignment with the ascending phase of the corresponding hazard function (Fig. 7*A*–*C*). In cell averages, oxytocin cells in both euhydrated rats and salt-loaded rats showed a longer post-spike hyperpolarization than $VP_{cont}$ cells (Fig. 7*Da*) and an extended ascending phase consistent with the longer mode in salt-loaded rats. Mean spike waveforms from $VP_{cont}$ [Correction made on 18 November 2025 after first online publication: "$VP_{phasic}$" has been updated to "$VP_{cont}$".] cells in salt-loaded rats were similar to those from euhydrated rats (Fig. 7*Db*) and those from $VP_{phasic}$ [Correction made on 18 November 2025 after first online publication: "$VP_{cont}$" has been updated to "$VP_{phasic}$".] cells showed indications of an enhanced DAP, but the 'overshoot' of post-spike excitability in vasopressin cells was apparent in some voltage traces (Fig. 7*C*, right panel) but not all.

### Survey of published literature

We found 39 studies in which plasma osmolality was measured in euhydrated and salt-loaded rats (Fig. 8*A* and Table 4). Osmolality was higher in salt-loaded rats at all time points in all studies, and increased with duration (up to 7 days) from a mean ± SD of 295.2 ± 9.4 mOsm $kg^{-1}$ (CI = 292–298). Across 22 studies, osmolality at 7 days was 335 ± 27 mOsm $kg^{-1}$ (CI = 323–346), which is 39 ± 24 mOsm $kg^{-1}$ (CI = 29–49) above euhydrated levels.

In response to volume depletion, vasopressin secretion can rise to high levels that exert pressor effects (Cowley & Liard, 1988; Dunn et al., 1973). Across 11 studies in salt-loaded rats (Fig. 8*B* and Table 5), haematocrit was increased by a mean of 3.3 (CI = 1.3–5.2), corresponding to a reduction in plasma volume of ~7% (if haematocrit is a valid surrogate for plasma volume in these circumstances). This is not enough to increase vasopressin secretion in euhydrated rats, but hypovolemia interacts synergistically with hyperosmolality in stimulating vasopressin and oxytocin secretion (Stricker & Verbalis, 1986).

Across 16 studies that had measured pituitary content of vasopressin and/or oxytocin (Fig. 8*C* and Table 6); both fell to an apparent nadir by ~5 days. For each study we averaged data for days 5–8 of salt loading; at this stage,

**Table 4. Reported mean levels of plasma osmolality (mOsm kg$^{-1}$) during 7 days of salt loading**

| Reference | EU | day1 | day 2 | day 3 | day 4 | day 5 | day 6 | day 7 | delta |
|---|---|---|---|---|---|---|---|---|---|
| Jones & Pickering (1969) | 314 | 321 | 314 | 346 | | 336 | 307 | | |
| Zingg et al. (1986) | 290 | 296 | 295 | | | | | | |
| Carter & Lightman (1987) | 291 | 297 | 301 | 305 | 314 | 307 | 315 | 322 | 31 |
| Zingg et al. (1988) | 290 | | | | | | 312 | | |
| Hyodo & Urano (1991) | 289 | | 305 | | 309 | | | 322 | 33 |
| Carter & Murphy (1989) | 288 | | | 308 | | | | | |
| Hooi et al. (1989) | 308 | | 331 | 319 | | | | 362 | 54 |
| Lafarga et al. (1992) | 294 | 319 | | | | | | | |
| Lepetit et al. (1992) | 297 | | | 319 | | | | | |
| Miyata et al. (1994) | 295 | 303 | | 313 | | 335 | | 375 | 80 |
| Yagita et al. (1994) | 308 | | | | | | | 343 | 35 |
| Dai & Yao (1995) | 290 | | | | 304 | | | | |
| | 282 | | | | 352 | | | | |
| Li et al. (1998) | 310 | | | | 320 | | | | |
| Curras-Collazo & Dao (1999) | 296 | | | 325 | 330 | | | 330 | 34 |
| Glasgow et al. (2000) | 302 | | | | | | | 317 | 15 |
| Zemo & McCabe (2001) | 263 | | | | | | | 292 | 28.6 |
| Morita et al. (2001) | 280 | | | | | | | 344 | 64 |
| Zhang et al. (2001) | 291 | | | | | | | 312 | 21 |
| Saito et al. (2003) | 293 | | | | | | | 378 | 85 |
| Somponpun & Sladek (2003) | 296 | | | 313 | | | | | |
| Bojanowska & Stempniak (2003) | 279 | | | | | | 314 | | |
| Ventura et al. (2005) | 299 | 308 | | | | | | | |
| Summy-Long et al. (2006) | 295 | | 297 | | | 299 | | | |
| Yue et al. (2008) | 291 | 301 | 308 | 313 | | 326 | | | |
| Kim et al. (2011) | 306 | 331 | | | | | | 359 | 53 |
| Doherty & Sladek (2011) | 301 | | | 308 | | | | | |
| Choe et al. (2015) | 301 | | | | | | | 350 | 49 |
| Greenwood et al. (2015) | 298 | | | | | | | 326 | 28 |
| Ribeiro et al. (2015) | 295 | | | | 333 | | | | |
| Mucio-Ramirez et al. (2017) | 297 | | | | | 321 | | | |
| Balapattabi et al. (2018) | 301 | | | | | | | 313 | 12 |
| Balapattabi et al. (2019) | 299 | | | | | | | 309 | 9.8 |
| Di et al. (2019) | 295 | | | | | | 353 | 329 | 34 |
| Ribeiro et al. (2015) | 294 | | | | | | | 308 | 14 |
| Barad et al. (2020) | 299 | | | | | | | 382 | 83 |
| Neupane et al. (2021) | 308 | 330 | | | | | | 361 | 53 |
| Levi et al. (2021) | 299 | | | | | | | 339 | 40 |
| Gomes et al. (2023) | 288 | | | | | | | 290 | 2 |
| Mean | 295.2 | | | | | | | 334.7 | 38.8 |
| *n* | 38 | | | | | | | 22 | 22 |
| SD | 9.4 | | | | | | | 26.8 | 23.9 |
| SEM | 1.5 | | | | | | | 5.7 | 5.1 |
| CI$_{high}$ | 298.1 | | | | | | | 345.9 | 48.8 |
| CI$_{low}$ | 292.2 | | | | | | | 323.4 | 28.8 |

The column headed delta shows the difference between osmolality at day 7 and osmolality in euhydrated rats (EU)

vasopressin content was $17 \pm 11\%$ that of euhydrated rats (11 studies, CI = 11–24) and oxytocin content was $24 \pm 18\%$ that of euhydrated rats (four studies, CI = 9–40).

Across 15 studies that had measured plasma [vasopressin] and/or [oxytocin] by radioimmunoassays (Table 7), the concentrations of both were raised in salt-loaded rats, but with no obvious continuing increase by day (Fig. 8*D*). In euhydrated rats, the mean [vasopressin] was $2.0 \pm 1.4$ pg mL$^{-1}$ (15 studies) and [oxytocin] was $5.4 \pm 4.4$ pg mL$^{-1}$ (10 studies). In salt-loaded rats, we averaged data for the different durations of salt loading beyond 2 days to give a single estimate

**Table 5. Reported levels of plasma haematocrit (Hcrt)**

| Reference | EU Hcrt | SL days | SL Hcrt | SL Av | Delta |
|---|---|---|---|---|---|
| Jones & Pickering (1969) | 43 | 1 | 47 | | |
| | | 2 | 46 | 47.1 | 4.1 |
| | | 3 | 48.5 | | |
| | | 5 | 47 | | |
| Lepetit et al. (1992) | 43 | 3 | 47 | 47 | 4 |
| Yagita et al. (1994) | 41.2 | 7 | 49.2 | 49.2 | 8 |
| Callahan et al. (1996) | 42 | 4 | 39 | 39 | −3 |
| Li et al. (1998) | 42 | 4 | 41 | 41 | −1 |
| Morita et al. (2001) | 40.5 | 7 | 48.3 | 48.3 | 7.8 |
| Bojanowska & Stempniak (2003) | 42 | 6 | 46 | 46 | 4 |
| Somponpun & Sladek (2003) | 41.2 | 3 | 45.1 | 45.1 | 3.9 |
| Summy-Long et al. (2006) | 44.7 | 2 | 45.1 | 43.75 | −0.95 |
| | | 5 | 43 | | |
| | | 8 | 44.5 | | |
| Balapattabi et al. (2018) | 42.6 | 7 | 47.8 | 47.8 | 5.2 |
| Balapattabi et al. (2019) | 42.5 | 7 | 47 | 47 | 4.5 |
| Ribeiro et al. (2020) | 43.8 | 7 | 46.4 | 46.4 | 2.6 |
| Mean | 42.3 | | | 45.5 | 3.2 |
| *N* | 12 | | | 12 | 12 |
| SD | 1.2 | | | 3.0 | 3.4 |
| SEM | 0.4 | | | 0.9 | 1.0 |
| $CI_{high}$ | 43.0 | | | 47.2 | 5.2 |
| $CI_{low}$ | 41.6 | | | 43.8 | 1.3 |

The columns headed EU show mean reported values (% packed cell volume in plasma samples) in euhydrated rats. Columns headed 'day …' show data from salt-loaded rats at the days indicated. The column 'SL Av' shows averages of group data from SL rats, and the column 'delta' shows the difference between this average and EU levels.

of concentrations in salt-loaded rats for each study. This gave means of $7.0 \pm 5.6$ pg mL$^{-1}$ vasopressin ($n = 13$) and $11.3 \pm 7.8$ pg mL$^{-1}$ oxytocin ($n = 8$), corresponding to increases of $5.3 \pm 1.4$ pg mL$^{-1}$ vasopressin (CI = 2.5–8.1) and $6.1 \pm 1.9$ pg mL$^{-1}$ oxytocin (CI = 2.5–9.8).

In collating data for Table 7, we included only studies that had measured plasma [vasopressin] by radioimmunoassays in blood samples taken without confounding anaesthetic effects (i.e. from trunk blood or from chronically implanted catheters). We thus excluded the study of Ludwig et al. (1996), who had reported that, under urethane anaesthesia, plasma [vasopressin] was 11.5 pg mL$^{-1}$ euhydrated rats, and 20.7 pg mL$^{-1}$ in rats that had drunk 2% NaCl for 2 days, by which time plasma [Na$^+$] had increased by 14 mM (28 mOsm kg$^{-1}$). We also excluded three studies that used enzyme-linked immunoassays (ELISAs) for vasopressin without extracting plasma samples because this can produce unrealistic measurements (Bie, 2024). Leng and Sabatier (2016) noted that plasma matrix interference can lead to an elevated baseline and/or exaggerate measured levels of oxytocin or vasopressin by a constant factor. Two studies on salt-loaded rats (Balapattabi et al., 2018, 2019) measured [vasopressin] in unextracted plasma samples

using an ELISA, reporting means of 15.7 and 14.8 pg mL$^{-1}$ in euhydrated rats and, 59.9 and 57.6 pg mL$^{-1}$ after 7 days of salt loading. Thus, in both studies, the values in both euhydrated rats and salt-loaded rats were ~8-fold greater than levels in extracted samples (Table 7). Kim et al. (2021) used a different ELISA, reporting a mean of 8.1 pg mL$^{-1}$ in euhydrated rats and 46.8 pg mL$^{-1}$ after 7 days of salt loading, which is four-fold and six-fold greater than levels in extracted samples.

### Comparison with measurements in the present study

We measured plasma [Na$^+$] and [vasopressin] and pituitary vasopressin content in eight salt-loaded rats and eight euhydrated rats (for full data, see https://github.com/HypoModel/MagnoNeuroDat). Plasma [Na$^+$] was higher in salt-loaded rats ($174 \pm 37$ mM; CI = 147–201 vs. $136 \pm 7$ mM; CI = 130–152), pituitary vasopressin content was lower ($125 \pm 49$ ng; CI = 92–159) vs. $875 \pm 201$ ng; CI = 740–1014) and plasma [vasopressin] was higher ($2.5 \pm 1.5$ pg mL$^{-1}$; CI = 1.5–3.6) vs. $1.1 \pm 0.3$ pg mL$^{-1}$; CI = 0.9–1.3). Thus, the status of salt-loaded rats in the present study was broadly as expected from the literature.

**Table 6. Reported levels of vasopressin (VP) and oxytocin (OT) measured in neurointermediate lobes**

| Reference | EU VP | EU OT | Duration (days) | SL VP | %control | SL OT | %control | VP av | OT av |
|---|---|---|---|---|---|---|---|---|---|
| Jones & Pickering (1969) | 259 | 177.5 | 1 | 201 | 78 | 126 | 71 | | |
| | | | 2 | 131 | 51 | 52 | 29 | | |
| | | | 3 | 82 | 32 | 36 | 20 | | |
| | | | 5 | 46 | 18 | 17 | 10 | 18 | 10 |
| Dyball & Pountney (1973) | 158 | 201.7 | 3 | 52.2 | 33 | 57.5 | 29 | | |
| Bakker & Dyball (1975) | 225 | 223 | 1 | 214 | 95 | 189 | 85 | | |
| | | | 3 | 107 | 48 | 81 | 36 | | |
| George (1976) | 18321 | | 3 | 8406 | 46 | | | | |
| Hollt et al. (1980) | 184.5 | | 5 | 9.1 | 5 | | | 5 | |
| Zingg et al. (1986) | 1239 | | 1 | 948 | 77 | | | | |
| | | | 2 | 723 | 58 | | | | |
| | | | 6 | 216 | 17 | | | 17 | |
| Hooi et al. (1989) | 2286 | | 2 | 301 | 13 | | | | |
| | | | 7 | 167 | 7 | | | 7 | |
| Higuchi et al. (1991) | 1.59 | 0.45 | 2 | 0.97 | 61 | 0.28 | 62 | | |
| | | | 4 | 0.51 | 32 | 0.23 | 51 | | |
| Crowley & Amico (1993) | 1081 | 1376 | 5 | 158 | 15 | 294 | 21 | 26 | 19 |
| | 1389 | 1795 | 5 | 514 | 37 | 307 | 17 | | |
| Kadowaki et al. (1994) | 4.7 | 6.1 | 3 | 2 | 43 | 2.3 | 38 | | |
| | | | 4 | 1.97 | 42 | 1.51 | 25 | | |
| Sheikh et al. (1998) | 284 | | 2 | 182 | 64 | | | | |
| | | | 7 | 51 | 18 | | | 18 | |
| Morita et al. (2001) | 975 | | 7 | 48 | 5 | | | 5 | |
| Bojanowska & Stempniak (2003) | 3343 | 1609 | 6 | 1226 | 37 | 827 | 51 | 37 | 51 |
| Kondo et al. (2004) | 1.84 | | 12 | 0.33 | 18 | | | | |
| Summy-Long et al. (2006) | 672 | 465 | 2 | 264 | 39 | 444 | 95 | | |
| | | | 5 | 209 | 31 | 190 | 41 | 23 | 31 |
| | | | 8 | 104 | 15 | 97 | 21 | | |
| Yue et al. (2008) | 100 | 100 | 1 | | 73 | | 44 | | |
| | | | 2 | | 65 | | 42 | | |
| | | | 3 | | 44 | | 22 | | |
| | | | 5 | | 20 | | 10 | 20 | 10 |
| | | | | | | Mean | | 17.6 | 24.2 |
| | | | | | | *n* | | 10 | 5 |
| | | | | | | SD | | 10.6 | 17.7 |
| | | | | | | SEM | | 3.4 | 7.9 |
| | | | | | | CI$_{high}$ | | 24.2 | 39.7 |
| | | | | | | Ci$_{low}$ | | 11.0 | 8.7 |

The columns headed EU show mean reported values in euhydrated rats, columns headed SL show data from salt-loaded rats at the days indicated. The data are expressed in different ways by different authors (per lobe, per mg tissue or protein, etc.) and are given here as the values reported; % changes from EU levels are calculated and combined as means. Columns headed 'av' show averages of % changes from EU levels in group data from rats salt-loaded for 5–8 days.

The effects of salt loading on body weight and fluid intake were monitored in five euhydrated rats and seven salt-loaded rats. Body weight declined progressively during salt loading, with a mean weight loss of $16 \pm 5\%$ of initial body weight after 7 days. Fluid intake increased progressively from $32 \pm 15$ mL day$^{-1}$ to $117 \pm 34$ mL day$^{-1}$, similar to changes reported in other studies (e.g. Balapattabi et al., 2018; Greenwood et al., 2014).

## Discussion

As explained in the Introduction, we had expected that, in salt-loaded rats, both oxytocin cells and vasopressin cells would be more active and more responsive to acute osmotic stimulation, and we expected to see a change in the responses of vasopressin cells to stimuli that are normally inhibitory. The basal firing rate of both vasopressin cells and oxytocin cells was elevated

**Table 7. Reported concentrations (pg mL$^{-1}$) of vasopressin (VP) and oxytocin (OT) measured in extracted plasma by radio-immunoassay**

| Reference | EU VP | EU OT | day SL | SL VP | av | SL OT | av |
|---|---|---|---|---|---|---|---|
| Mens et al. (1980) | 1.4 | | 1 | 15 | 14,7 | | |
| | | | 2 | 17.7 | | | |
| | | | 3 | 14.7 | | | |
| Carter & Lightman (1987) | 3.4 | 6.1 | 1 | 7.1 | 12.8 | 8 | 12.7 |
| | | | 2 | 15.3 | | 13.3 | |
| | | | 3 | 12.3 | | 11.1 | |
| | | | 4 | 10 | | 12.9 | |
| | | | 5 | 10.4 | | 12.2 | |
| | | | 6 | 20.4 | | 14.7 | |
| | | | 7 | 13.7 | | 11.4 | |
| | | | 9 | 10 | | 13.6 | |
| Van Tol et al. (1987) | 0.6 | 2 | 14 | 5.2 | 5.2 | 3.5 | 3.5 |
| Hooi et al. (1989) | 2 | | 2 | 5.2 | 4.1 | | |
| | | | 7 | 4.1 | | | |
| Blackburn et al. (1990) | 4.3 | 9.8 | 2 | 10.4 | | 17.9 | |
| Callahan et al. (1996) | 1.8 | | 4 | 2.4 | 2.4 | | |
| Li et al. (1998) | 0.3 | | 4 | 1.7 | 1.7 | | |
| Morita et al. (2001) | 0.9 | | 7 | 6 | 6 | | |
| Bojanowska & Stempniak (2003) | 4.1 | 1.7 | 6 | 21.7 | 21.7 | 18.8 | 18.8 |
| Somponpun & Sladek (2003) | 1 | 15.7 | 3 | 4.2 | 4.2 | 25.5 | 25.5 |
| Cisowska-Maciejewska & Ciosek (2005) | 4.4 | 3.2 | 2 | 6.7 | | 4.3 | |
| Ventura et al. (2005) | 2.8 | 4.7 | 1 | 4.7 | 5.6 | 17.3 | 8.5 |
| | | | 7 | 5.6 | | 8.5 | |
| (Summy-Long et al., 2006) | 1.6 | 5.3 | 2 | 2.9 | 2.4 | 5.5 | 7.9 |
| | | | 5 | 1.9 | | 9.2 | |
| | | | 8 | 2.9 | | 6.5 | |
| Yue et al. (2008) | 1 | 4.7 | 1 | 3.2 | 3.8 | 14.3 | 11.2 |
| | | | 2 | 3.8 | | 16 | |
| | | | 3 | 3.4 | | 6.5 | |
| | | | 5 | 4.1 | | 8 | |
| Greenwood et al. (2015) | 1.1 | 0.8 | 7 | 6.4 | 6.4 | 2.1 | 2.1 |
| Mean | 2.0 | 5.4 | | | 7.0 | | 11.3 |
| *n* | 15 | 10 | | | 13 | | 8 |
| SD | 1.4 | 4.4 | | | 5.9 | | 7.8 |
| SEM | 0.4 | 1.4 | | | 1.6 | | 2.8 |
| CI$_{high}$ | 2.8 | 8.2 | | | 10.2 | | 16.7 |
| CI$_{low}$ | 1.3 | 2.6 | | | 3.8 | | 5.9 |

The columns headed EU show mean reported values in euhydrated rats. Columns headed SL show data from salt-loaded rats at the days indicated. Columns headed 'av' show averages of reported group data from rats salt-loaded for more than two days (data in italics).

in salt-loaded rats, and we found a marked increase in the incidence of phasic firing in vasopressin cells. Because phasic firing is efficient at stimulating vasopressin secretion, minimizing energy expenditure on action potential generation (Bicknell, 1988), this seems to be an appropriate adaptation to chronic stimulation. However, the elevation in firing rate was much more modest than expected, responses to osmotic challenge were reduced rather than increased, and we saw no weakening of inhibitory actions.

## Attenuated osmoresponsiveness of supraoptic neurones

In the present study, supraoptic neurones increased their firing rate linearly during i.v. infusions of 2 M NaCl, which raise plasma [Na$^+$] by ~4.9 mM Na$^+$ (9.8 mOsm kg$^{-1}$) per mmol infused (Leng et al., 2001). In euhydrated rats, the response slope was ~1.9 spikes s$^{-1}$ per mmol infused, similar to previous studies (Leng et al., 2001; Roy et al., 2021); hence, the mean firing rate increases

by $\sim$0.2 spikes $s^{-1}$ for every mOsm $kg^{-1}$ increase. After 7 days of salt loading, plasma osmolality is raised by an average of $\sim$39 mOsm $kg^{-1}$ (Table 4), and so, if supraoptic neurones in salt-loaded rats retained their absolute osmoresponsiveness, their firing rate of would be elevated by an average of 7.8 spikes $s^{-1}$. However, we found an elevation of just 0.9 spikes $s^{-1}$ (CI = 0.7–1.2),

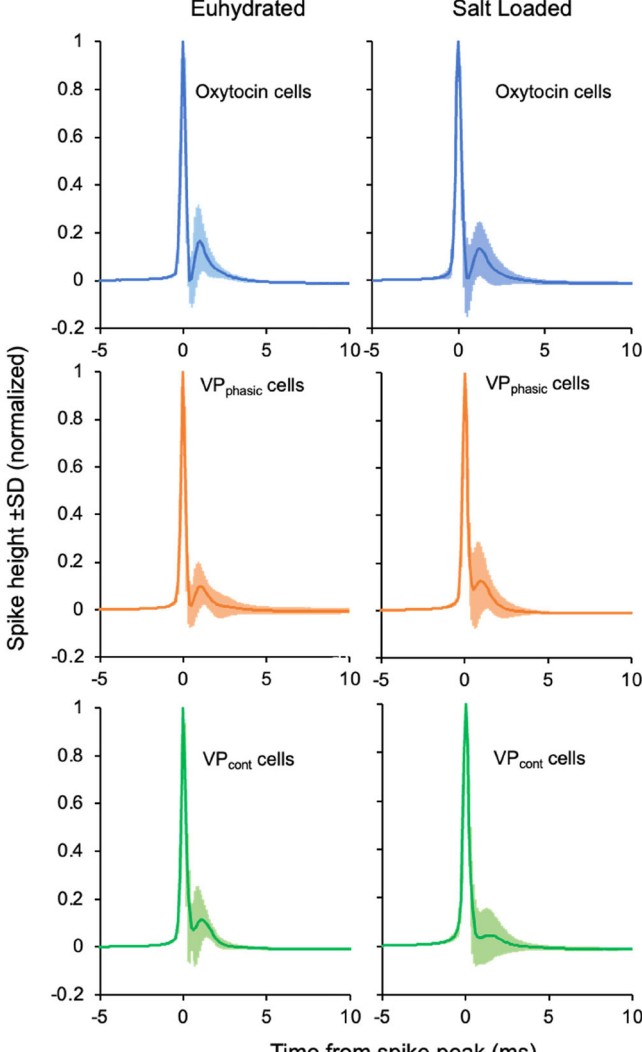

**Figure 6. Spike waveforms in euhydrated and salt-loaded rats**
Mean ± SD spike waveforms from cells in euhydrated and salt-loaded rats. The average waveforms from individual cells were normalized to a spike height of 1. The normalized waveforms show little variability except at the tail of the descending phase where there is a 'hump' that varies with electrode position. The waveforms are shown as positive-going by convention; the recorded waveforms were negative-going. Oxytocin cell data are from 21 cells from 17 euhydrated rats and 22 cells from 18 salt-loaded rats; $VP_{cont}$ data are from 27 cells from 20 euhydrated rats and 19 cells from 16 salt-loaded rats; $VP_{phasic}$ data are from 20 cells from 16 euhydrated rats and 30 cells from 20 salt-loaded rats. Full data are available at: https://github.com/HypoModel/MagnoNeuroDat.

and their responsiveness to hypertonic infusions was also attenuated.

There have been only two previous studies of magnocellular neurones in salt-loaded rats *in vivo*. Dyball and Pountney (1973) in urethane-anaesthetized rats reported that the median firing rate of supraoptic neurones increased from between 2 and 4 spikes $s^{-1}$ in euhydrated rats ($n$ = 69 cells) to >4 spikes $s^{-1}$ in 3 day salt-loaded rats (again 69 cells), whereas the intraburst firing rate of phasic cells increased from $\sim$5 to $\sim$8 spikes $s^{-1}$. In the present study, the corresponding levels were similar: the median firing rate was 4.8 spikes $s^{-1}$ in euhydrated rats and 5.6 spikes $s^{-1}$ in salt-loaded rats, whereas the mean intraburst firing rate in phasic cells increased from 5.5 to 8.6 spikes $s^{-1}$. Choe et al. (2015), in a relatively small study, also in urethane-anaesthetized rats, gave no data for oxytocin cells but reported an increase in mean firing rate of 3.9 spikes $s^{-1}$ in vasopressin cells after 7 days of salt loading, a much larger increase than seen in the present study, but still less than expected from the rise in plasma osmolality.

Thus there appears to be an inconsistency between the results obtained *in vivo* under urethane anaesthesia and data from *in vitro* studies that have indicated that vasopressin cells are hyperactive in salt-loaded rats as a result of enhanced osmosensitivity.

## Is there hypersecretion of vasopressin and oxytocin secretion in salt-loaded rats?

As detailed in Table 7, previous studies have reported that, in rats salt-loaded for 7 days or more, plasma [vasopressin] is increased by just 4.4 ± 3.2 pg $mL^{-1}$ (seven studies, CI = 2.0–6.8) and [oxytocin] by just 3.0 ± 2.0 pg $mL^{-1}$ (five studies, CI = 1.2–4.7). By contrast, in euhydrated rats, infusions of hypertonic saline that raise plasma osmolality produce much higher hormone levels. In conscious rats, Huang et al. (2000) raised plasma osmolality by 43 mOsm $kg^{-1}$ by infusing 1 M NaCl at 2 mL $h^{-1}$ for 6 h, and reported that plasma [vasopressin] increased from 3 to 75 pg $mL^{-1}$ and plasma [oxytocin] from 11 to 175 pg $mL^{-1}$. Also in conscious rats, Terwel and Jolles (1994) infused 9% NaCl at 30 μL $min^{-1}$, and reported that plasma [vasopressin] increased from 3 pg $mL^{-1}$ to >100 pg $mL^{-1}$ beyond a plasma osmolality of 320 mOsmol $kg^{-1}$. In urethane-anaesthetized rats, Leng et al. (2001) raised plasma osmolality from 296 to 334 mOsm $kg^{-1}$ by infusing 1 M NaCl at 26 μL $min^{-1}$ for 60 min and reported that plasma [oxytocin] increased from 10.5 to 148 pg $mL^{-1}$.

However, although the hormone levels in salt-loaded rats are much lower than might be expected from plasma osmolality, they may nevertheless be adequate for maximal concentration of the urine. In humans, plasma

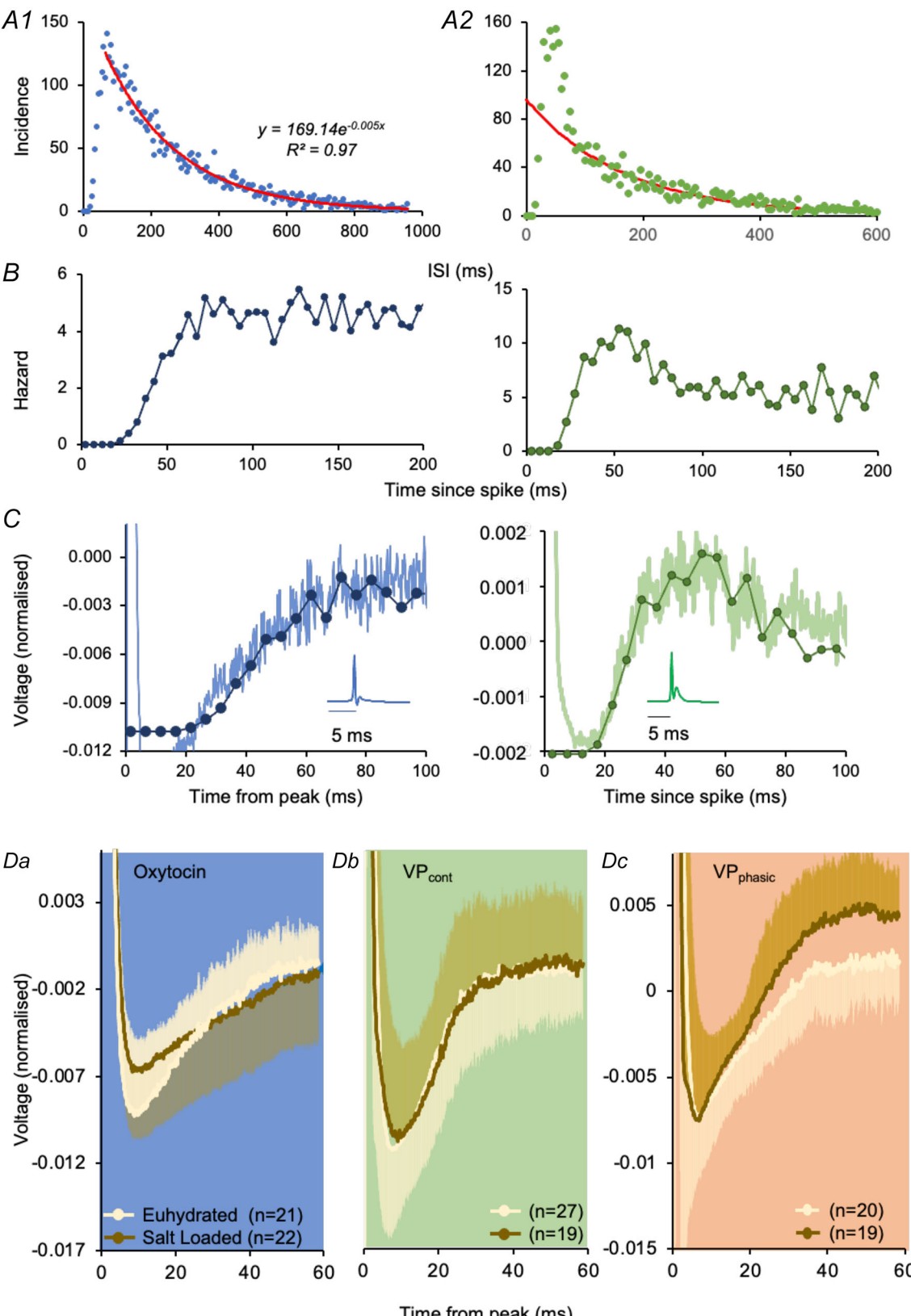

**Figure 7. Waveform analysis**

ISI distributions of an oxytocin cell (blue) (*A1*) and a VP$_{cont}$ cell (green) (*A2*) from a euhydrated rat. The tail of the distribution in A1 is well fitted by a negative exponential, whereas an exponential fitted to the tail of the VP$_{cont}$

cell distribution undershoots the peak. *B*, hazard functions corresponding to the distributions in (*A*). *C*, hazard functions from (*B*) overlaid on mean spike-triggered voltage records from the same cells. *D*, spike triggered voltage records (mean ± SD, shown unilaterally for clarity) from (*Da*) oxytocin, (*Db*) VP$_{cont}$ cells and (*Dc*) VP$_{phasic}$ cells from euhydrated and salt-loaded rats. Full data are available at: https://github.com/HypoModel/MagnoNeuroDat. [Correction made on 12 November 2025 after first online publication: The figure has been updated to reflect the correct background colour for fig 7*Db* and *Dc*.]

osmolality is normally between 280 and 295 mOsm kg$^{-1}$; at ~295 mOsm kg$^{-1}$, the urine is maximally concentrated, and 'the entire physiologic range of urine osmolality is accomplished by relatively small changes in plasma vasopressin of 0 to 5 pg ml$^{-1}$' (Verbalis, 2018). In these respects, rats do not appear to be very different. For example, Gellai et al. (1984) studied the renal effects of infusing vasopressin to Brattleboro rats deficient in endogenous vasopressin, finding that a plasma concentration of 8 pg mL$^{-1}$ 'may be approaching the level where vasopressin induces maximal water permeability of the late distal tubules and collecting ducts'. In their classic study, Dunn et al. (1973) reported that plasma [vasopressin] in conscious rats reaches ~10 pg mL$^{-1}$ when plasma osmolality is raised by ~10 mOsm kg$^{-1}$.

### Is an increased firing rate of 1 spike s$^{-1}$ consistent with the observed increase in vasopressin secretion?

MacGregor and Leng (2013) fitted computational models of vasopressin cell activity and stimulus-secretion coupling to a range of published data. The combined model (in fig. 8*E* and *F* of their paper) allows the average spiking frequency of a heterogeneous population of model phasic cells to be linked to predicted plasma concentration; in the range 1–6 spikes s$^{-1}$, the relationship is linear, and secretion increases by ~6.4 pg mL$^{-1}$ for every increase of 1 spike s$^{-1}$. In the present study, vasopressin cells discharged 0.9 spikes s$^{-1}$ faster in salt-loaded rats than in euhydrated rats, and, across the published literature, plasma [vasopressin] is 4.4 pg mL$^{-1}$ higher (in rats salt-loaded for >7 days). This is below model predictions, but the model is based on data from euhydrated rats; the depleted pituitary content in salt-loaded rats might be expected to reduce activity-dependent secretion (Higuchi et al., 1991).

### Are the reported levels of vasopressin and oxytocin consistent with the observed depletion of pituitary content?

In euhydrated rats, Roberts et al. (1991) reported a linear relationship between the rate of vasopressin infusion and plasma [vasopressin]; from their fig. 5, a 6.6 pg mL$^{-1}$ increase in [vasopressin] will arise from an infusion of vasopressin at 14 ng h$^{-1}$, equivalent to an increase in secretion of 336 ng day$^{-1}$. The mean vasopressin content

of the pituitary in euhydrated rats was 875 ng in the present study; 336 ng is 38% of this, which is more than the mean depletion in the first day of salt loading (19% from the four studies in Fig. 7). Thus, an increase of ~1 spike s$^{-1}$ in vasopressin cell activity is, as far as can be evaluated, consistent with both measured levels of plasma [vasopressin] and the depletion of pituitary vasopressin content, assuming that replenishment of stores through increased secretion lags behind depletion.

Vasopressin mRNA levels begin to rise on day 1 (Hyodo & Urano, 1991; Lightman & Young, 1987; Yue et al., 2008), but peptide translation, vesicle packaging and axonal transport all take time, and so replenishment of pituitary content will lag behind depletion. mRNA levels increase further over the following days, apparently reaching equilibrium by ~5 days (van Tol et al., 1987), when pituitary content reaches a nadir at ~20% of the initial content. There has been no report of any recovery of pituitary content during salt loading, so it appears that daily synthesis comes into balance with daily secretion but never exceeds it.

### No evidence of change in GABA actions

We found no indication of a reversal of the effects of GABA in salt-loaded rats, and so there is a discrepancy between our results and findings in the literature. However, the evidence on GABA is inconsistent. Widmer et al. (2003) reported that, in 3–4-week-old rats, the GABA agonist muscimol increased intracellular [Ca$^{2+}$] in isolated oxytocin cells and increased oxytocin release from supraoptic nucleus explants but not in older rats. Haam et al. (2012) reported that, in hypothalamic slices from euhydrated rats, GABA agonists inhibited oxytocin cells but excited vasopressin cells, but others reported that, in supraoptic neurones from euhydrated rats in hypothalamic slices (Kim et al., 2011) or a hypothalamic explant (Choe et al., 2015), GABA is exclusively inhibitory, consistent with *in vivo* electrophysiological studies (Leng et al., 2001; Ludwig & Leng, 2000; Sabatier & Leng, 2006). Studies in both *in vitro* preparations reported a change from GABA inhibition to excitation in both oxytocin and vasopressin cells in tissue taken from salt-loaded rats. Choe et al. (2015) reported that this switch was accompanied by a downregulation of NKCC2 expression in vasopressin cells with no significant change in NKCC1 expression, and that the NKCC1 blocker bumetanide was without effect, whereas Kim et al. (2011) reported that the

switch was accompanied by an increase in NKCC1 protein expression in the supraoptic nucleus and a relatively small increase in NKCC2 expression, and that bumetanide reversed the switch.

Thus, the responsiveness to GABA varies in different *in vitro* preparations. One possibility is that the trauma involved in *in vitro* preparations alters intracellular [Cl⁻], as described for hippocampal slices (Dzhala et al., 2012).

The shear stresses involved in slice preparations are different to those in explant preparations, whereas the latter may better preserve the integrity of the ventral glial lamina. It seems possible that, as suggested by Choe et al. (2015), the hypertrophied magnocellular neurones of salt-loaded rats may be more vulnerable to this trauma.

Changes in the effects of GABA may reflect increases in intracellular [Cl⁻], as can follow prolonged opening

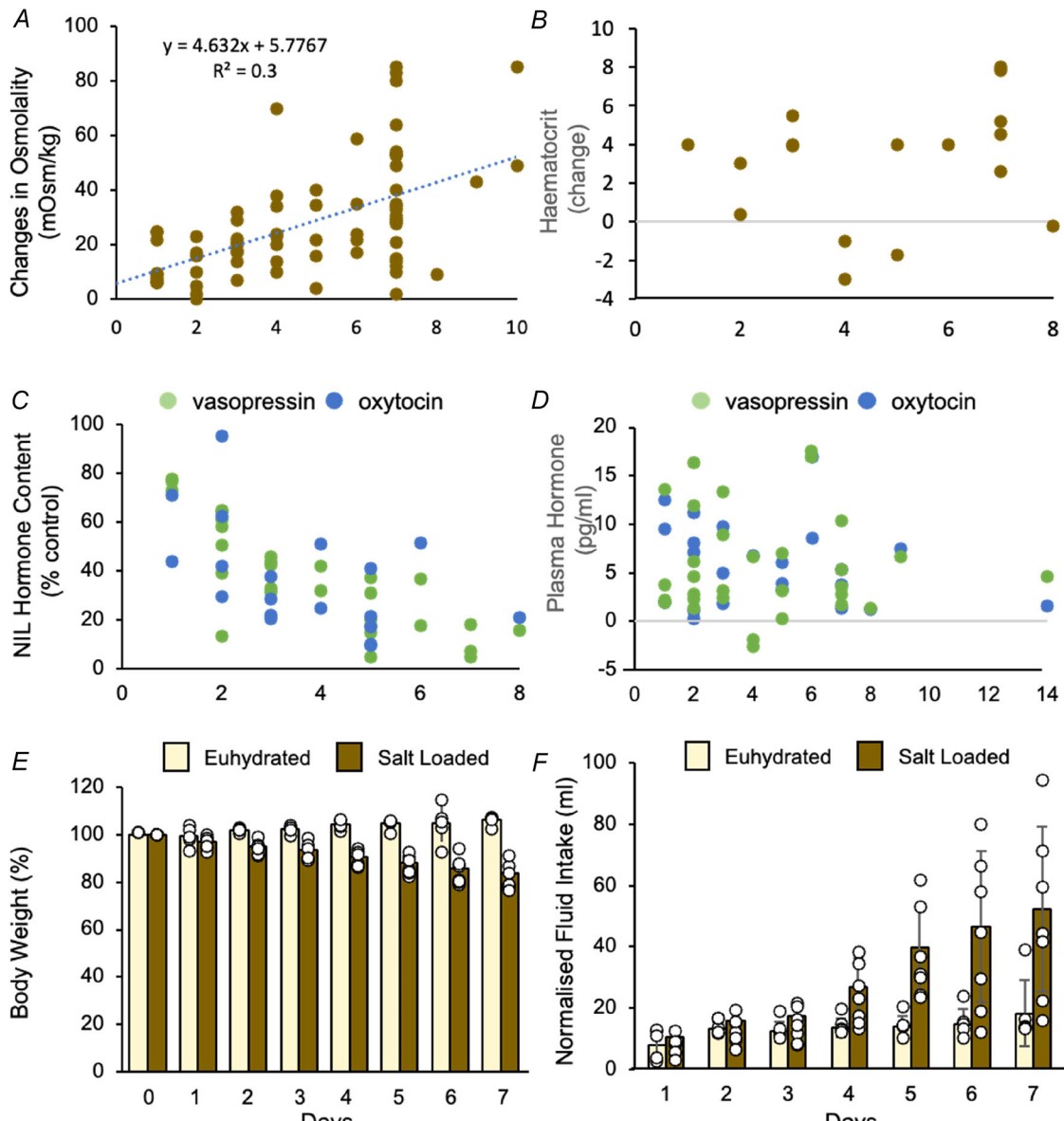

**Figure 8. Effects of drinking 2%NaCl for up to 10 days on fluid balance**
*A–D*, each point is the mean level reported in a published study corresponding to the relevant time point: full details of the extracted data are given as in Tables 4–7. *A*, effects on plasma osmolality, shown as changes from the reported levels in euhydrated rats. The trendline is a linear fit to the study means through 0, with equation and $r^2$ value indicated. *B*, reported levels of plasma haematocrit. *C*, neurointermediate lobe (NIL) content of vasopressin (green) and oxytocin (blue dots). *D*, plasma measurements of vasopressin (green) and oxytocin (blue dots) measured by radioimmunoassay. *E–F*, measurements of body weight and fluid intake in salt-loaded rats in the present study. *E*, change in body weight (% of initial weight) during 7 days of drinking 2% NaCl (*n* = 7) or tap water (*n* = 5). *F*, daily fluid intake in the same rats, expressed as mL kg⁻¹ body weight.

of membrane $Cl^-$ channels. One candidate for causing this is taurine, an intracellular osmolyte that is exported when cells are exposed to a fall in extracellular osmolality. The astrocytes that surround magnocellular neurones are a rich source of taurine and release large amounts in response to hypotonic challenge (Deleuze et al., 1998). This activates glycine receptors on the magnocellular neurones, inhibiting them by inducing $Cl^-$ entry (Bres et al., 2000; Choe et al., 2012; Hussy et al., 2000). Because *in vitro* studies of the effects of salt loading have studied neuronal behaviour in an extracellular environment characteristic of euhydrated conditions, the hypotonic shock involved in moving from a strongly hypertonic environment *in vivo* may trigger taurine release that alters the neuronal $Cl^-$ equilibrium.

### Osmoreceptive mechanisms

In physiological circumstances plasma osmolality is mainly determined by [NaCl], but, as shown by Verney (1947), intracarotid infusions of sucrose in dogs produce similar reductions in urine flow and increases in urinary $[Cl^-]$ to those evoked by equimolar infusions of NaCl. Verney (1947) deduced that the response was 'due not specifically to sodium chloride, but to the rise in osmotic pressure'. However, when urea was infused to produce a similar increase in plasma osmolality, no antidiuresis was observed. From this, Verney (1947) inferred that the osmoreceptors were permeable to urea but impermeable to NaCl and sucrose; he speculated that they were located in the highly-vascularized supraoptic nucleus, and that osmotically-induced cell shrinkage was transduced into electrical signals by mechanical stretch receptors. This was ultimately confirmed by Oliet and Bourque (1993), who showed that supraoptic neurones are indeed intrinsically osmosensitive and that this mainly derives from osmotically-induced cell shrinkage, transduced by mechanosensitive ion channels.

Bourque and co-workers subsequently identified the channel involved as a N-terminal variant of the transient receptor potential vanilloid 1 (Trpv1) channel, which appears to interact with a layer of actin filaments (F-actin) beneath the plasma membrane and with a network of microtubules occupying the cytoplasm (Prager-Khoutorsky & Bourque, 2015; Prager-Khoutorsky et al., 2014). The same mechanism appears to be present in osmoreceptive neurones of the OVLT, a midline circumventricular organ located anterior to the supraoptic nuclei that is the source of an afferent input to the magnocellular neurones that contributes to their osmoresponsiveness (Ciura & Bourque, 2006).

**Previous measures of osmoresponsiveness.** Although the initial response to hypernatraemia involves cell shrinkage, after 7 days of salt loading the supraoptic neurones are hypertrophied, with a doubling of soma area (Zhang et al., 2001) and an increase in the density of subcortical F-actin (Levi et al., 2021). One explanation for a change in osmoresponsiveness might be that the excitability of supraoptic neurones adapts during salt loading as a result of adaptation of intrinsic osmoreceptor mechanisms.

Ludwig et al. (1996) reported that (under urethane anaesthesia) an I.P. injection of 3.5 M NaCl (0.6 mL per 100 g body weight) raised plasma [vasopressin] by 64 pg $mL^{-1}$ in euhydrated rats and 93 pg $mL^{-1}$ in 2 day salt-loaded rats, which is a mean difference of 29 pg $mL^{-1}$, with a standard error (calculated from the data provided) of 25 pg $mL^{-1}$. Thus, this study provides no evidence of a loss of osmoresponsiveness at this early stage of salt loading, but no strong evidence of an enhancement either.

Levi et al. (2021) injected conscious rats s.c. with 2 M NaCl at 1.43 mL $kg^{-1}$ and measured Fos expression in vasopressin cells 1 h later as a marker of neuronal activation. This stimulus is not specific because s.c. injections of even small volumes of hypertonic saline (20 μL of 10% NaCl) are painful to rats (Ami & Satou, 2015), but it induced Fos expression in ∼90% of vasopressin cells in 7 day salt-loaded rats compared to ∼65% in euhydrated rats consistent with enhanced osmoresponsiveness. Levi et al. (2021) found further evidence for this from electrophysiological studies in brain slices. When the osmolality of the bathing medium was raised from 297.5 to 312.5 mOsmol $kg^{-1}$, vasopressin cells from salt-loaded rats responded more strongly than cells from euhydrated rats. Isolated vasopressin cells from salt-loaded rats also responded more strongly to cell shrinkage induced by applying negative pressure to a patch pipette. In both cases, osmoresponsiveness was measured not from the high basal levels of extracellular osmolality typically found in salt-loaded rats (339 mOsmol $kg^{-1}$) (Table 7) but from levels typical of euhydrated rats.

It thus appears from studies conducted *in vitro* that, when the hypertrophied supraoptic neurones of salt-loaded rats are removed from their strongly hypernatraemic extracellular environment and studied in normonatraemic medium, they show enhanced activation in response to acute osmotic stimulation. From the present data, however, it seems that in their hypernatraemic extracellular environment they are less osmoresponsive than normal. However, because urethane itself raises plasma osmolality, we must consider whether this might affect the interpretation of *in vivo* studies.

### Urethane anaesthesia

Clearly, urethane anaesthesia has limitations, and is obviously unsuitable for studying renal mechanisms,

but urethane has been the anaesthetic of choice for *in vivo* studies of magnocellular neurones because it does not appear to impair neuroendocrine reflexes, and as studies in decerebrate rats indicated that urethane, unlike other anaesthetics, has no sustained effect on the activity of hypothalamic neurones (Cross & Dyer, 1971). A direct comparison between the electrical activity of neurones in conscious and anaesthetized rats was made by Summerlee and Lincoln (1981), who described the electrophysiological characteristics of oxytocin cells in lactating rats and their responses to suckling as 'identical' to observations reported on urethane-anaesthetized rats. Other studies have compared urethane with other anaesthetics. Pentobarbitone depresses the activity and responsiveness of magnocellular neurones, but Catheline et al. (2006) used isoflurane, an inhalational anaesthetic that they could closely control to maintain a stable level of anaesthesia. They showed that the pattern of reflex milk ejections in lactating rats under isoflurane was the same as in conscious rats, and that the electrophysiological characteristics of oxytocin neurones were similar to those recorded under urethane anaesthesia. They also detailed the characteristics of 14 phasic cells, reporting a mean intraburst frequency of 7.4 spikes $s^{-1}$, which is slightly higher than observed in urethane-anaesthetized male rats in the present study.

Plasma [vasopressin] and [oxytocin] are elevated in urethane-anaesthetized rats, and because anaesthetic doses of urethane raise plasma osmolality by 10–15 mOsmol $kg^{-1}$, it is commonly assumed that this explains the increased hormone levels. However, osmoreceptors do not respond to osmolality *per se*: transduction of osmotic pressure changes depends on stretch-sensitive membrane receptors that signal volume changes (Prager-Khoutorsky & Bourque, 2015). Because urethane is readily soluble in water and lipids (Field & Lang, 1988), it crosses cell membranes freely, and therefore does not produce a change in cell volume. Nevertheless, urethane may contribute to activating magnocellular neurones in other ways; anaesthetics inevitably affect many body systems in diverse ways. In particular, the increase in plasma [vasopressin] may reflect reduced clearance because I.P. urethane reduces blood flow through the kidneys (Severs et al., 1981), and between one-third and one-half of the metabolic clearance of vasopressin occurs via the kidneys (Rabkin et al., 1979).

The reduced renal blood flow appears to be the consequence of an accumulation of fluid within the peritoneum. Although urethane does not produce a sustained rise in plasma [$Na^+$], the early response to I.P. injection is an osmotic movement of water into the peritoneal space from the vasculature and adjacent tissues that causes what Severs et al. (1981) described as a 'toxic effect on the mesenteric vasculature'. This transient hypo-volaemia and hypernatraemia, together with the pain of i.p injection, may also account for the strong expression of Fos in magnocellular neurones that is consistently seen in urethane-anaesthetized rats Indeed, Meyer et al. (1987) reported that, in lactating rats, at 1 h after urethane injection mean plasma [vasopressin] was ~140 pg $mL^{-1}$, falling to ~22 pg $mL^{-1}$ 2 h later, a level that remained stable for the following 3 h. Plasma [oxytocin] was also transiently increased in the first hour after urethane injection, but much less markedly (to ~30 pg $mL^{-1}$).

## The salt-loading model

Salt-loading rats by substituting 2% NaCl for drinking water does not mimic any plausible physiological challenge for humans, but Norway rats (*Rattus norvegicus*, from which laboratory rats are derived) have colonized a wide range of environments (Modlinska & Pisula, 2020), including estuaries and islands with no standing sources of fresh water. They are considered to have spread globally as stowaways on ships, with bilge water – a mix of fresh water and seawater – the only available source of fluid. In 1943, Adolph (1943) asked 'Do rats thrive when drinking seawater?', and replaced their drinking water with either seawater (whole or diluted) or various concentrations of NaCl. The maximal urinary [$Cl^-$] that he observed was 0.6 M, compared to 0.37 M in man, but he noted that 'the rat's unique ability to spare some water by producing highly concentrated urine is overpowered by its natural inability to spare water from evaporation'. On seawater alone (0.52 M $Cl^-$), rats lost 25% of their body weight over 6 days.

However, rats seem to accommodate to drinking 2% NaCl, and Pena et al. (1988) maintained rats for 90 days on this to study its effects on the morphology of the neural lobe. In rats, chronic stress increases the expression of corticotropin releasing factor mRNA in parvocellular neurones of the paraventricular nucleus, and promotes the secretion of ACTH and glucocorticoids (Herman & Tasker, 2016). However, after 7 days of drinking 2% NaCl, plasma [corticosterone] and [ACTH] and corticotropin releasing factor mRNA expression in the parvocellular neurones are all *reduced* (Amaya et al., 2001).

Rats drinking 2% NaCl initially reduce their food intake and lose body weight (by ~10% after 5 days in the study of Fujio et al. (2006) and 16% after 7 days in the present study). They drink relatively little on the first day (Greenwood et al., 2014) and this results in a lower plasma volume and a higher plasma [$Na^+$]. The subsequent thirst induces more intake of the 2% NaCl, which ameliorates the hypovolemia at the same time as exacerbating the hypertonicity – but, with the increased fluid intake, urine production can be correspondingly increased. For example, Balapattabi et al. (2018) reported

that, over 7 days of salt loading, daily fluid intake increased from ~35 to ~85 mL and urine excretion increased from ~20 to ~70 mL, and Fujio et al. (2006) reported that, after 5 days, urinary $[Na^+]$ had increased from ~100 to ~420 mm.

In summary, salt-loaded rats drink large amounts of 2% NaCl to ameliorate the rise in plasma osmolality. The excess salt intake (~29 mmol $day^{-1}$ for a rat drinking 85 mL) must be eliminated in the urine or plasma osmolality will continue to rise; 31 mmol $Na^+$ (29 from fluid and 2 from food) can be excreted in 70 mL of urine at 440 mm $Na^+$, comprising a concentration close to the maximum sustainable urinary $[Na^+]$ for a rat (Stricker et al., 2001). After 7 days, plasma [vasopressin] is ~8 pg $mL^{-1}$ (Table 7), which is 'close to the level where vasopressin induces maximal water permeability at the kidney' (Gellai et al., 1984), whereas plasma [oxytocin] is ~11 pg $mL^{-1}$, which is above the threshold concentration required to stimulate natriuresis (5–6 pg $mL^{-1}$; Verbalis et al., 1991). By this time, mRNA levels have stabilized, and synthesis appears to balance secretion as neural lobe content also appears to have stabilized.

In 7 day salt-loaded rats, we find that magnocellular neurones have a markedly attenuated osmoresponsiveness; despite a high plasma osmolality, they fire only ~1 spike $s^{-1}$ faster than in euhydrated conditions, although this is enough to sustain the required hormone concentrations in plasma. To fire any faster would exacerbate the already extreme depletion of pituitary content to no adaptive benefit for water and electrolyte homeostasis.

The following supplemental information is openly available online at: https://github.com/HypoModel/MagnoNeuroDat.

(1) '*Electrophysiological studies*'. Primary electrophysiology data: spike times of cells at 0.1 ms resolution with metadata on interventions; all recorded supraoptic neurones, in all three groups of rats ((Excel files 'Primary data EU'. 'Primary data SL' and 'Primary data RE')

(2) Analysis of $VP_{phasic}$ cells contributing to Table 1 (Excel file 'Burst analyses open')

(3) '*Changes in firing rates and patterning*'. Secondary data: all CCK responses, ISI distributions and hazard functions, and summaries of data from these contributing to Tables 1 and 2 and Figs 2 and 3 (Excel files 'EU Open,' 'SL Open' and 'RE Open').

(4) '*Attenuated responses to acute hyperosmotic stimulation*'. All analysed data contributing to Table 3 and Fig. 4 (Excel File 'Hyperosmotic Open').

(5) '*CCK induced inhibition of vasopressin cells involves GABA*'. All cell data contributing to Fig. 1 (Excel file 'cckGABA Open')

(6) '*Unchanged responses to phenylephrine*'. All cell data contributing to Fig. 2 (Excel file 'Phenylephrine Open').

(7) '*Unchanged responses to hypovolemia*'. All cell data contributing to this section of the Results (Excel file 'PEG analysis Open').

(8) '*Enhanced post-spike hyperexcitability in vasopressin cells*'. All cell data contributing to Fig. 6 (Excel file 'Waveforms Open').

(9) '*Spike waveforms mirror hazard functions*'. All data and analyses contributing to Fig. 7 (Excel file 'Waveform Hazard Open').

(10) '*Comparison with measurements in the present study*'. Plasma $[Na^+]$ and [vasopressin] and pituitary vasopressin content in eight salt-loaded rats and eight euhydrated rats (Word file 'Comparison data').

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

## Additional information

### Data availability statement

Data from the electrophysiological recordings: the Excel workbooks containing data used to create graphs and tables in the paper have been deposited at https://github.com/HypoModel/MagnoNeuroDat. Any additional information required to reanalyse the data reported in this paper is available from the lead contact upon request. Further information and requests for recordings and software should be directed to and will be fulfilled by the lead contact, Mike Ludwig (mike.ludwig@ed.ac.uk).

### Competing interests

The authors declare that they have no competing interests.

### Authors contributions

G.L. and M.L. were responsible for conceptualization. G.L. and M.L. were responsible for methodology. R.K., D.M. and G.L. were responsible for formal analysis. M.a.L., R.K. and N.S. were responsible for investigations and data curation. G.L. and M.L. were responsible for writing the original draft. G.L. and M.L. were responsible for reviewing and editing. G.L. and M.L. were responsible for resources and funding acquisition. G.L., M.L. and N.S. were responsible for supervision. All authors have approved the final version of the manuscript submitted for publication and agree to be accountable for all aspects of the work. All persons designated as authors qualify for authorship, and all those who qualify for authorship are listed.

### Funding

This work was supported by a grant from the Biotechnology and Biological Research Council (BB/S021035/1) (GL, DM and ML) and The PhD Scholarship in Commemoration of His Majesty King Bhumibol Adulyadej's 90th Birthday Anniversary from Chulabhorn Royal Academy, Thailand, awarded to RK.

### Acknowledgements

We thank Dr Chris Coyle for collecting the data on body weight and fluid intake (Fig. 8*E* and *F*).

## Author's present address

Roongrit Klinjampa: Princess Srisavangavadhana Faculty of Medicine, Chulabhorn Royal Academy, Bangkok, Thailand.

## Keywords

electrophysiology, GABA, hypothalamus, microdialysis, osmosensitivity, oxytocin, salt-loading, supraoptic nucleus, vasopressin

## Supporting information

Additional supporting information can be found online in the Supporting Information section at the end of the HTML view of the article. Supporting information files available:

**Peer Review History**

