## [Peer Review History · The Journal of Physiology]

Reduced osmoresponsiveness in magnocellular neuroendocrine neurones during chronic salt-loading

Maja Lozic, Roongrit Klinjampa, Nancy Sabatier, Duncan J MacGregor, Gareth Leng, and Mike Ludwig
DOI: 10.1113/JP288860

Corresponding author(s): Mike Ludwig (mike.ludwig@ed.ac.uk)

The following individual(s) involved in review of this submission have agreed to reveal their identity: Michael Joseph McKinley (Referee #1); Peter Bie (Referee #2)

Review Timeline:

Submission Date:	10-Mar-2025
Editorial Decision:	11-Apr-2025
Revision Received:	24-Jun-2025
Editorial Decision:	28-Jul-2025
Revision Received:	07-Aug-2025
Editorial Decision:	20-Aug-2025
Revision Received:	26-Aug-2025
Accepted:	27-Aug-2025

Senior Editor: Vaughan Macefield

Reviewing Editor: Vagner Antunes

Transaction Report:

Dear Dr Ludwig,

Re: JP-RP-2025-288860 "**Functional plasticity in magnocellular neuroendocrine neurones during chronic salt-loading: reduced osmosensitiveness to avoid futile hypersecretion**" by Maja Lozic, Roongrit Klinjampa, Nancy Sabatier, Duncan J MacGregor, Gareth Leng, and Mike Ludwig

Thank you for submitting your manuscript to The Journal of Physiology. It has been assessed by a Reviewing Editor and by 3 expert referees and we are pleased to tell you that it is potentially acceptable for publication following satisfactory major revision.

REVISION CHECKLIST:

We look forward to receiving your revised submission.

Yours sincerely,

Vaughan Macefield
Senior Editor
The Journal of Physiology

REQUIRED ITEMS

- Author photo and profile. First or joint first authors are asked to provide a short biography (no more than 100 words for one author or 150 words in total for joint first authors) and a portrait photograph. These should be uploaded and clearly labelled together in a Word document with the revised version of the manuscript. See Information for Authors for further details.

- You must start the Methods section with a paragraph headed Ethical approval (https://jp.msubmit.net/cgi-bin/main.plex?form_type=display_requirements#methods).

Research must comply with The Journal's policies regarding animal experiments (<https://physoc.onlinelibrary.wiley.com/hub/animal-experiments>) and adherence to these policies must be stated in the manuscript.

Authors should confirm in their Methods section that their experiments were carried out according to the guidelines laid down by their institution's animal welfare committee, including an ethics approval reference number. The Methods section must contain a statement about access to food, water and housing, details of the anaesthetic regime: anaesthetic used, dose and route of administration, and method of killing the experimental animals.

- Your manuscript must include a complete Additional Information section, including competing interests; funding; author contributions and acknowledgements.

- Please upload separate high-quality figure files via the submission form.

- You must upload original, uncropped western blot/gel images (including controls) if they are not included in the manuscript. This is to confirm that no inappropriate, unethical or misleading image manipulation has occurred. These should be uploaded as 'Supporting information for review process only'. Please label/highlight the original gels so that we can clearly see which sections/lanes have been used in the manuscript figures. For more information, see: <https://physoc.onlinelibrary.wiley.com/hub/journal-policies#imagmanip>.

- Please ensure that any tables are editable and in Word format, and wherever possible, embedded in the article file itself.

- Please ensure that the Article File you upload is a Word file.

- Papers must comply with the Statistics Policy: https://jp.msubmit.net/cgi-bin/main.plex?form_type=display_requirements#statistics.

In summary:

- If n {less than or equal to} 30, all data points must be plotted in the figure in a way that reveals their range and distribution. A bar graph with data points overlaid, a box and whisker plot or a violin plot (preferably with data points included) are acceptable formats.
- If $n > 30$, then the entire raw dataset must be made available either as supporting information, or hosted on a not-for-profit repository, e.g. FigShare, with access details provided in the manuscript.
- 'n' clearly defined (e.g. x cells from y slices in z animals) in the Methods. Authors should be mindful of pseudoreplication.
- All relevant 'n' values must be clearly stated in the main text, figures and tables.
- The most appropriate summary statistic (e.g. mean or median and standard deviation) must be used. Standard Error of the Mean (SEM) alone is not permitted.
- Exact p values must be stated. Authors must not use 'greater than' or 'less than'. Exact p values must be stated to three significant figures even when 'no statistical significance' is claimed.

- Please include an Abstract Figure file, as well as the Figure Legend text within the main article file. The Abstract Figure is a piece of artwork designed to give readers an immediate understanding of the research and should summarise the main conclusions. If possible, the image should be easily 'readable' from left to right or top to bottom. It should show the physiological relevance of the manuscript so readers can assess the importance and content of its findings. Abstract Figures should not merely recapitulate other figures in the manuscript. Please try to keep the diagram as simple as possible and without superfluous information that may distract from the main conclusion(s). Abstract Figures must be provided by authors no later than the revised manuscript stage and should be uploaded as a separate file during online submission labelled as File Type 'Abstract Figure'. Please also ensure that you include the figure legend in the main article file. All Abstract Figures should be created using BioRender. Authors should use The Journal's premium BioRender account to export high-resolution images. Details on how to use and access the premium account are included as part of this email.

EDITOR COMMENTS

Reviewing Editor:

The manuscript was reviewed by three experts in the field, all of whom recognized the relevance and timeliness of your study, which addresses the physiology of magnocellular neurons in a salt-loading model. The reviewers agree that your data challenge current paradigms and have the potential to significantly impact the field. However, they also raised serious concerns as summarized below:

1. The introduction section was noted as overly lengthy and in need of substantial editing. The reviewer recommends condensing the background to improve clarity and focusing more directly on content that is essential to your hypotheses and findings.
2. A major concern was raised regarding the interpretation of the 7-day salt-loading protocol. The reviewer questions whether the observed effects reflect true salt loading or a dehydration state. This distinction should be clarified in the discussion, supported by additional data or literature-based justification.
3. Concerns were expressed about the strength of the conclusions. The reviewer calls for a more detailed and transparent methodology, a clear acknowledgment of the study's limitations, and a thorough comparison with prior studies-particularly those with conflicting results.

For more details, please see the reviewers' comments.

Senior Editor:

Thank you for submitting your manuscript to The Journal of Physiology. I have now received comments from three independent reviewers and the Reviewing Editor, all experts in the field. As you will see from their comments, there are some important methodological issues you will need to address before we can consider the manuscript further. In addition, you will need to reduce the length of the Introduction. I invite you to revise the manuscript accordingly and submit point-by-

point responses to the reviewers' comments. I look forward to receiving your revised manuscript in due course.

Additionally, please note the requirements of the statistics policy.

REFeree COMMENTS

Referee #1:

The manuscript describes experiments that investigated the effect of 7 days of ingestion of 20% NaCl on the electrophysiological characteristics of supraoptic magnocellular neurons. This work continues investigations of these neurons by this team of investigators. The methods used have been honed by them over the years and the results show that vasopressin secreting neurons are less responsive to hypertonicity than would be expected from the rise in osmolality resulting from hyperosmotic treatments.

While the experimental work is competently performed and the conclusions drawn acceptable, there are aspects of the manuscript that require attention. My main concern is the experimental model (salt loading with 20% NaCl) used and what it means from a physiological viewpoint. A rat drinking a pure solution of 0.34M NaCl is unlikely to happen outside a laboratory. While the rats are probably salt loaded and are extremely hypernatremic (plasma), it is not clear whether they are dehydrated and extracellular volume reduced. An important piece of information that is missing is a 7 day history of the rats prior to experimentation. The daily intakes of 2% NaCl, food, body weight, and excretion of Na in urine and faeces would have been very helpful in knowing how much Na had been ingested and retained during those 7 days. I suspect that food intake would have been reduced in these rats, thus if they maintained or increased body weight it would indicate they were not dehydrated. The measurements mentioned above could still be done in a group of rats. As well, the maximum Na concentration that rats achieve in urine under this condition would also be helpful to know. Stricker et al. *Am J Physiol* 280, R831-R842, 2001 showed evidence that rats drinking 0.3M NaCl could maintain normal plasma NaCl over 24 h, but did not study a longer period. It should be noted that dehydration from water deprivation results in a natriuresis causing an overall bodily Na deficit (Schloorlemmer & Evered, *Can J Physiol Pharmacol* 71, 379-386, 1993; McKinley et al. *Am J Physiol* 245, R287-R292, 1983). I suspect that the rats drinking 2% NaCl had reduced food intake which will also affect body weight. Would a better model of salt loading be providing rats with only isotonic saline to drink together with a high sodium diet?

In regard to the manuscript, some aspects need revising. The meta-analyses of literature on plasma osmolality, haematocrit and plasma vasopressin and oxytocin is "overkill" and better suited to a review article. They should be removed from methods, and results and the appropriate ranges of values given and referenced in the Discussion when comparing the current results, by citing only the most appropriate of the references.

In Results, the number of animals used in some experimental groups is not given. In Methods, it states that "there was a large number of animals in each experimental group". This is an open ended statement and does not provide sufficient information on numbers which should be stated in results in each section where animal numbers are missing.

In the legend Figure 1, it is stated that there was stimulation of OVLT, but the technique used is not given in Methods. In Fig. 5G, the fawn and brown colours are rather similar could be better differentiated.

On the third page of Discussion, line 25, I think "hypertonicity" rather than "hypotonicity" is meant.

Minor comments.

1. Throughout, 2% NaCl should have a space (not 2%NaCL).
2. Where first used, abbreviations should be specified e.g. interspike interval (ISI), hyperpolarizing afterpotential (HAP) and DAP.
3. Page 6, line 20. A word is missing between "made --- an Axopatch"
4. Page 7, 3rd line from bottom, "all".
5. Page 10, 2 nd line. A reference should be cited for total body water.

Referee #2:

The manuscript is big, covering 49 pages including 97 literature references. Unfortunately, neither lines nor pages are

numbered, making reviewer references to specific sections, sentences, and expressions unnecessarily unprecise.

The title conveys two messages, (i) chronic salt loading is associated with reduced responsiveness of magnocellular neurons, and (ii) this reduced responsiveness can be rationalized to constitute avoidance of futile hypersecretion. The former is a clear statement open to assessment. The latter is less comprehensible. Notably, the title does not reveal that the manuscript includes a (pointed, but detailed) literature review of 36 studies in rats drinking 2% NaCl for 2 days or more.

The protocols were designed to (i) survey relevant literature data, and (ii) to test the hypothesis that 'salt loading, induced by replacing drinking water with 2%NaCl, hyperactivates these neurones leading to hypersecretion and salt-dependent hypertension.' The descriptions of the aim of the study and the essence of the results ("These findings contradict the prevailing understanding that salt-loading promotes a pathological hyperexcitability and exaggerated vasopressin secretion") would benefit from distinctions between (i) hyperexcitability and hyperactivity, and (ii) acute and chronic hyperexcitability/hyperactivity. The summary informs the reader that - prior to this work - the prevailing understanding is that chronic neuronal hyperactivity is exaggerated compared to the acute response to salt loading (subsequently described as an increase in 'intrinsic osmosensitivity'), and that the present data show the opposite, i.e., that the chronic neuronal activity is markedly lower than that expected from the magnitude of the acute response. In this perspective the manuscript seems to convey valuable information.

The introduction is extensive (1000+ words) and begins with the description of the elevation in blood pressure occurring in rats exposed with 2% saline as the only drinking water. The modest elevation in systolic blood pressure seen under these circumstances (+15 mmHg is provided as a frequent observation) probably has little to do with the normal concept of salt-sensitive hypertension. The authors describe that (i) 2% saline is aversive to normal rats, (ii), prolonged exposure to it generates 'excessive thirst', and (iii) literature data indicate that plasma osmolality in this situation increases by some 35 mOsm/kg (which is a lot). This means that the rats for days were kept under conditions which were very unpleasant for them. In this context, a 15 mmHg increase in systolic blood pressure seems unexpectedly small and fully explainable by the stressful situation. Remarkably there is no indication in the manuscript acknowledging that the rats were kept under continuously stressful conditions for a week. The salt loading increased plasma sodium concentrations from 136 to 174 mmol/L indicating that the increase in plasma osmolality was higher than 35 mOsm/kg. Potentially, there is imbalance between the unquestionable distress of the animals and the potential advancement of science provided by the results.

In the presentation and discussion of the experimental and literature parts of the present study, the experiments should be given priority. The literature survey should be appreciated in light of the new data.

Numerically, the key results are remarkable. In salt loaded rats, plasma vasopressin levels were 2.3-fold higher than in euhydrated controls (2.5 vs.1.1 pg/ml). The corresponding firing rates for supraoptic neurons averaged 6.0 and 5.1 spikes/s (cf. legend to Fig. 3: "Supraoptic cells of all types fire ~1 spikes s⁻¹ faster after 7 days of salt loading and recover after rehydration."). Under the (reasonable) assumption that the total body clearance of plasma vasopressin is not markedly changed by salt loading, the discrepancy between the increases in firing frequency and plasma vasopressin is puzzling, particularly in light of the expected finding that pituitary stores of vasopressin were markedly reduced during salt loading.

Discussion: Several comments are warranted: (i) ' However, the present results suggest that either the excitability of supraoptic neurones adapts during sustained hyperosmolality, or their osmosensitiveness is reduced.': If it is assumed, the supraoptic neurons themselves are not the osmosensitive units, what is the differences between excitability and osmosensitiveness? (ii) Vasopressin numerics: 'It seems impossible to obtain interpretable measures of plasma vasopressin in the same experimental conditions as the present study, because urethane anaesthesia impairs vasopressin and oxytocin clearance.' Is this correct? It is reasonable to assume a priori that metabolic clearance of vasopressin is much greater than the renal clearance, and numbers could be discussed. (iii) The authors handle oxytocin neurons and oxytocin secretion as if it is self-evident that oxytocin should be secreted in response to salt loading. Not so to the present reviewer. (iv) several new paragraphs are not immediately appearing as self-contained trains of thoughts/arguments (do not use 'this' in first sentence of new paragraph). (v) 'hypotonicity': rather 'hypertonicity'. (vi) is it possible that neurohypophysial tissue normally provides both a passing way and a parking lot for vasopressin, and that under chronic salt loading just the parking lot is empty?

The attitude of the editor to the inclusion of a review type section in an original, hypothesis driven, project-based manuscript could be either 'this does not belong here' or 'this is a model manuscript'. Considering the present, highly questionable biomedical publishing environment, this reviewer favors the second option. Addition of the review section adds valuable perspectives to the study.

Minor issues

Introduction: 'futile waste': a pleonasm, waste is futile

Electrophysiological characteristics: 'ISI distribution': undefined acronym. Applies to other acronyms as well.

Acute hyperosmotic stimulation with 2M NaCl: '0.91 mg of NaCl': please use mmol

Discussion: 2 x 'per 100 mg infused': please use mmol

Referee #3:

This is an interesting study which shows the responses of supraoptic vasopressin (AVP) and oxytocin (OXT) neurons to osmotic stimuli after a salt-loading and hypovolemia. The study makes an overview of previous works, summarizing the values of plasma osmolality, hematocrit, and AVP and OXT in plasma and posterior pituitary, measured in control and salt-loaded rats. In addition, the study uses *in vivo* electrophysiology to analyse firing patterns of supposed AVP and OXT neurons.

While it is a potentially interesting work, including many data on the firing patterns of different neuronal populations (for example, phasic vs non-phasic AVP neurons), there are numerous methodological issues which make problematic the interpretation. In particular, the conclusions of the authors are in contradiction with several already well-known concepts. The study does not make a detailed examination or give possible explanations of why the results are contradicting the established knowledge. In addition, the discussion could benefit to mention some limitations and to interpret the data in a more careful way, because actually the conclusions and statements are too much strong and sometimes depend of a problematic methodology.

Major points:

1. In summary, the authors write: "We tested the hypothesis that salt loading, induced by replacing drinking water with 2% NaCl, hyperactivates these neurons leading to hypersecretion and salt-dependent hypertension." However, this statement should be reconsidered, since the authors did not measure neither hormonal secretion nor hypertension.

2. In the introduction, it is mentioned: "In humans, as summarised by Verbalis, plasma osmolarity is normally between 280 and 295 mOsm/kg; at ~295 mOsm/kg the urine is maximally concentrated, and 'the entire physiologic range of urine osmolarity is accomplished by relatively small changes in plasma vasopressin of 0 to 5 pg/ml' (Verbalis, 2018)." This affirmation is supposed to apply in the same way for the rats. But on what the authors did based themselves for to say this? Why this should apply also to the rat? Because rats dispose of a more large capacity for concentrate the urine compared to the human (2-3 folds), it seems very probably that the physiological range of osmolarity which can be regulated by vasopressin is much more large in rat. So, this affirmation, and also the calculation of the "putative" maximal efficient concentration of AVP, are not really supported and should be modified. Notably, the estimation of what is the maximal "efficient" concentration of AVP is presented like the main rationale of the study and of the conclusions, and it is actually misleading.

3- Further, the authors indicate that "plasma vasopressin by well-validated radioimmunoassays in rats after salt-loading for {greater than or equal to}3 days; the mean vasopressin concentration in euhydrated rats was 1.8 pg/ml, rising to 7.3 pg/ml after salt-loading." These values appear not coherent with the previous publications of the same group (for example Ludwig et al, 1996, PMID 8880739), where it was reported that plasma AVP is increased until 20.7 pg/ml already after 2 days of salt-loading. More, they also shown that an acute *i.p.* injection of hypertonic saline increased AVP plasma level from 11.5{plus minus}2.4 to 75.4{plus minus}15.5 pg/ml (control, $P < 0.05$), and that in salt-loaded animals the increase was much more strong: from 20.7{plus minus}5.2 to 113.7{plus minus}19.3 pg/ml (salt-loaded, $P < 0.05$). This contradict also the affirmation that "Maximal antidiuresis is achieved at relatively low plasma concentrations (~10 pg/ml)".

4. The paper that the authors are citing in the introduction (Dyball & Pountney, 1973) indicate that urethane anaesthesia seems to stimulate the liberation of neurohypophysial hormones. Moreover, it was shown that it can significantly modify the hormonal content of the neurolobe, and increase the plasma concentration of the hormones (Ginsburg & Brown, 1957; Dyball, 1971). The authors are attributing this effect to a reduce clearance by the urine, but alternatively, it can be the result of the known effect of urethane to raise in the systemic osmolality. This point should be more addressed by the authors.

5. With regards to authors conclusions that basal rates show very little increases in salt-loaded animals. This is a central point that should be clarified, since this represents a major contradiction with previous studies. For example, an early study by (Dyball & Pountney, 1973) show that salt-loading increase the firing rate of neurons in the SON and PVN. Furthermore, Jones & Pickering (1969) and (Dyball & Pountney, 1973) postulated that decreased content of the hormonal stores in the pituitaire are not due to decreased biosynthesis but due to increased release. In the (Dyball & Pountney, 1973), they

demonstrated after 3 days of salt-loading, that in phasic SON cells (putative AVP neurons), the frequency of firing in the bursts was largely increased from ~5-6Hz to ~8-9Hz. This is consistent with the more recent evidence from in vivo study by Choe et al, showing that basal firing increase of AVP neurons from ~5.5 to ~9.4Hz after 7 days of salt-loading. The authors should discuss why their results deviate so much from these previous studies?

6. For the experiment where microdialysis probe was used to apply drugs, what was the osmolality of aCSF measured and adjusted whether the recordings were made from control or salt-loaded rats? Based on the composition of the aCSF, it seems that the rats were perfused with hypertonic aCSF (314?). This suggests that before testing, the control rats received a moderate chronic hypertonic stimulus (+20, considering their baseline osmolality was 295), while the salt-loaded rats received a moderate hypotonic stimulus (-17, considering their osmolality after 7 days of salt-loading was ~331). Therefore, what is presented as the "basal firing activity" of AVP neurons was in fact, after prolonged perfusing of control rats with hypertonic and salt-loaded rats with hypotonic stimuli. This likely increased the measured "basal activity" of control AVP neurons while decrease the "basal firing" of salt-loaded rats, and eliminating the differences between these 2 conditions.

7. For hypertonic stimulations, the authors depend on theoretical calculation based on the amount 2M NaCl injected, animals weight, and the fact that urethane anesthesia attenuate urine production (and thus sodium excretion). Based on these calculations, the putative increase in osmolality is ~14 mOsm/kg, which is relatively small increased. However, given all the assumption, especially the assumed effects on sodium excretion, it is unclear what the stimuli actually were. Furthermore, the effect of urethane on urine production might differ between control salt-loaded, and hypovolemia, and thus injecting same volume of NaCl can produce different changes in plasma osmolality. The authors should measure blood osmolality in rats in these settings, to provide actual (and not putative) measurements of the hypertonic stimuli in each condition (control, salt-loaded, hypovolemic). One of the most surprising statements of the authors, that contradicts previous studies, is that neurons from salt-loaded animals are not more but less osmosensitive. This conclusion simply cannot be drawn unless it has been measured what osmolality changes were triggered by the manipulation.

8. The main conclusion of the study, claiming that AVP (and OXT) neurons are not more osmosensitive in salt-loading, should be discussed in the context of methodological limitations that are associated with in vivo recordings conducted under urethane anaesthesia. It has been well established that urethane anaesthesia in rats cause pronounced plasma hyperosmolality of body fluids without affecting plasma sodium levels. This suggests that urethane interferes with the body's ability to regulate water balance, increase in plasma osmolality, potentially impacting AVP release in response to hyperosmolality (e.g. PMID: 7022489, 20470865). These findings indicate that urethane anaesthesia modifies the body's normal response to osmotic challenges, potentially affecting AVP neuron activity. This represents a major limitation of studies on fluid homeostases conducted in vivo but using urethane anaesthesia. The authors should discuss this limitation of their study, and also discuss whether alternative methods can be implicated and beneficial (in vivo, ex vivo, or in vitro).

9. Related to the point#8, previous study focusing on the similar question, utilized an approach that circumvent using urethane anaesthesia (Levi et al 2021). Instead, they use estimated neuronal activity using c-Fos. Despite the limitation of c-Fos, that does not allow to look at the actual firing patterns, it provides a measure of overall neuronal activity in response to a stimulus. In contrast to the results described by the authors, Levi et al reported that AVP neurons from rats exposed to 7 days of salt-loading exhibit increased osmosensitivity to acute osmotic stimuli (NaCl injections). Importantly, in this case, rats were not exposed to urethane that disrupts response to hyperosmolality, and the isoflurane anaesthesia used prior to sacrificing the animals cannot trigger any effects that would fast enough to have an effect on c-Fos expression, thus, representing a more "clean" demonstration of in vivo effect of salt-loading on osmosensitivity of AVP neurons. The authors should incorporate this additional evidence and discuss their results in this context.

Minor comments:

1. For key points - the reviewer is unsure whether the key points should summarize the results of the current study or provide more general statements relevant to the study? This is since the first key point does not reflect the results of the current study, but concepts which were established by decade of previous studies, which, however, were more systematically summarized by the authors. I am not sure whether this is appropriate for the "key points" and perhaps more suitable for the abstract or discussion.

2. Moreover, this statement that depletion of the pituitary content of AVP and OXT "apparently reflecting enhanced secretion" is not supported by the authors data.

3. The key point statement "These findings contradict the prevailing understanding that salt-loading promotes a pathological hyperexcitability and exaggerated vasopressin secretion..." should be modified, unless the authors provide evidence where previous studies interpreted increased osmosensitiveness as "pathological hyperexcitability". Furthermore, Hyperexcitability would imply changes in response to (also) non-osmotic stimuli, which contradicts the results.

4. In the introduction: "in salt-loaded rats as a result of increased intracellular [Cl⁻] resulting from upregulation of the Na⁺/K⁺/Cl⁻ cotransporter1 (Balapattabi et al., 2019)" - this is a wrong reference. Kim et al showed that but they didn't use salt-loading but a different model of salt-dependent hypertension. While Balapattabi et al. used salt-loaded, they showed down-regulation of K⁺/Cl⁻ co-transporter (KCC2) with no or insignificant upregulation of Na⁺/K⁺/Cl⁻ (NKCC1).

5. Also in the discussion, the authors refer the Choe et al 2015 paper, stating that "However, Choe et al. reported that this switch was accompanied by a downregulation of NKCC1 expression in vasopressin cells with no change in NKCC2 expression, and that the NKCC1 blocker bumetanide was without effect (Choe et al., 2015). This is incorrect statement, since in this paper, Choe et al reported that "our electrophysiological analysis with blockers of KCC2 and NKCC1 indicated that this effect is due to a SL-mediated reduction in KCC2 activity, rather than an increase in NKCC1 activity". In fact, they have also show using western blot that while KCC2 levels decreased in salt-loading, while NKCC1 levels are not significantly affected.

6. In the intro, authors state "plasma vasopressin by well-validated radioimmunoassays in rats after salt-loading for {greater than or equal to}3 days; the mean vasopressin concentration in euhydrated rats was 1.8 pg ml⁻¹, rising to 7.3 pg ml⁻¹ after salt-loading" this seem inconsistent with the previous reports of the same group showing that plasma AVP is increased to 20.7 pg/ml already after 2 days of salt loading. Further, they also showed that while an acute "i.p. injection of hypertonic saline increased plasma AVP levels from 11.5{plus minus}2.4 to 75.4{plus minus}15.5 pg/ml (control, P<0.05)" the increase in SL animals was much stronger "from 20.7{plus minus}5.2 to 113.7{plus minus}19.3 pg/ml (salt-loaded, P<0.05)". This contradicts to the statement that "Maximal antidiuresis is achieved at relatively low plasma concentrations (~10 pg ml⁻¹),"

7. In tables summarizing values from previous studies reporting posterior pituitary levels of AVP and OXT. These studies used different durations of salt-loading, however the authors averaged experiments with salt-loading lasting 5, 6, and 7 days. Why different salt-loading lengths were group together? These analyses should be done for different days. Also, some experiments of 8 days and 5 days were excluded. It's unclear why.

Similar problems with the other tables summarizing values from previous studies, i.e. reporting AVP and OXT plasma concentrations. It seems like random days were picked from different studies and averaged together.

END OF COMMENTS

We thank the Reviewing Editor and the three expert referees for the scholarly reading of our manuscript, for spotting errors, for their considered and constructive suggestions for revisions, and for their interesting questions and pointed but constructive criticisms. We have undertaken a major revision, shortening the introduction and expanding and re-structuring the discussion to address all of the comments and criticisms raised. In the marked copy of the revised manuscript, we have highlighted only major changes – not minor changes and corrections. In the following we describe how we have responded to all the points raised

Reviewing Editor:

1. The introduction section was noted as overly lengthy and in need of substantial editing. The reviewer recommends condensing the background to improve clarity and focusing more directly on content that is essential to your hypotheses and findings.

We have shortened the Introduction to 595 words, focussing on essential content

2. A major concern was raised regarding the interpretation of the 7-day salt-loading protocol. The reviewer questions whether the observed effects reflect true salt loading or a dehydration state. This distinction should be clarified in the discussion, supported by additional data or literature-based justification.

We have addressed this in detail in a new section to the Discussion headed “The salt loading model” (lines 740-790, and see below)

3. Concerns were expressed about the strength of the conclusions. The reviewer calls for a more detailed and transparent methodology, a clear acknowledgment of the study's limitations, and a thorough comparison with prior studies - particularly those with conflicting results. ***We have striven to describe our methodology in great detail. We have also addressed conflicts with previous studies in the revised discussion, as we detail below.***

Referee #1:

My main concern is the experimental model (salt loading with 20% NaCl) used and what it means from a physiological viewpoint. A rat drinking a pure solution of 0.34M NaCl is unlikely to happen outside a laboratory. While the rats are probably salt loaded and are extremely hypernatremic (plasma), it is not clear whether they are dehydrated and extracellular volume reduced.

An important piece of information that is missing is a 7 day history of the rats prior to experimentation. The daily intakes of 2% NaCl, food, body weight, and excretion of Na in urine and faeces would have been very helpful in knowing how much Na had been ingested and retained during those 7 days. I suspect that food intake would have been reduced in these rats, thus if they maintained or increased body weight it would indicate they were not dehydrated.

We have added a section “The salt loading model” to the Discussion (lines 740-790) to address these and other comments. There, we explain from literature sources that rats on 2% NaCl increase their fluid intake considerably and increase their urine excretion similarly; plasma haematocrit is increased by < 5%. To consider them dehydrated seems inappropriate, they drink copious amounts of fluid and show at most a modest rise in haematocrit which does not increase progressively. The rats reduce their food intake (by ~10%) – this appears to be a general response to hypernatraemia, and it has been speculated that this is a homeostatic adaptation to reduce salt intake, including by the actions of oxytocin to suppress salt appetite (Verbalis et al (1993). Central oxytocin inhibition of food and salt ingestion: a mechanism for intake regulation of solute homeostasis. Regul Pept 29;45:149-54.)

The measurements mentioned above could still be done in a group of rats. As well, the maximum Na concentration that rats achieve in urine under this condition would also be helpful to know.

Such measurements have been made in some previous studies, and we have now quoted these in the new section on The salt loading model (lines 740-790). We do have our own data on body weight and fluid intake from a group of rats, but do not have data from metabolic cage experiments – in discussions with the local vets on ethics before beginning the study we noted that rats in metabolic cages are stressed through being kept on wire flooring, and since we would be replicating published studies and combining this with salt loading, we agreed that this should be avoided unless essential.

Stricker et al. Am J Physiol 280, R831-R842, 2001 showed evidence that rats drinking 0.3M NaCl could maintain normal plasma NaCl over 24 h, but did not study a longer period. It should be noted that dehydration from water deprivation results in a natriuresis causing an overall bodily Na deficit (Schloorlemmer & Evered, Can J Physiol Pharmacol 71, 379-386, 1993; McKinley et al. Am J Physiol 245, R287-R292, 1983). I suspect that the rats drinking 2% NaCl had reduced food intake which will also affect body weight.

0.3M NaCl is close to 2% NaCl (0.36M) but this difference may be important; Stricker et al. inferred that drinking this did not change plasma osmolality (they did not measure it); these rats drank much more fluid than rats on 2% NaCl do in the first 24 h (> 3 x control levels): They state “The more fluid they consumed, the more concentrated urine they excreted and, therefore, the less concentrated their urine had to be to conserve enough urinary water for osmoregulation. Ultimately, rats ingested so much fluid that estimated P_{osmol} was restored to normal.” However, as shown in our Table 4, after one day of drinking 2% NaCl plasma osmolality was raised (in 7 studies) by at least 6 mOsm kg⁻¹ – and (in two studies) > 20 mOsm kg⁻¹. Rats on 2% NaCl do increase their fluid intake, but not as rapidly as in Stricker et al.; they do so progressively, reaching a 3-fold increase only after several days. Schloorlemmer & Evered studied rats after 10 h of water deprivation and saw increased natriuresis but little change in plasma osmolality. Over 24 h, in McKinley et al. plasma [Na⁺] rose from 142 to 148 mM in rats (~12 mOsm kg⁻¹), despite causing an overall bodily Na deficit.

Thus we believe that these studies are consistent with the data we show here, and with our interpretation of those data. We are reluctant to extend our discussion to cover these studies as they do not seem to pose an additional challenge that we need to answer.

Would a better model of salt loading be providing rats with only isotonic saline to drink together with a high sodium diet?

This is an interesting question, but one that would take a lot of words to do it justice. No experimental model is perfect, and at the end of this long study we certainly agree that the model has limitations that we did not appreciate at the outset. Ultimately whether something is a “better model” depends on specifying “better for what?” This is a good model in that the rats appear to reach a new equilibrium within a few days with minimal stress, and a great deal is already known from a very large number of previous studies – so it is a good model for the question of how animals adapt to circumstances that require them to find a new homeostatic equilibrium. It is an interesting model in that it raises fundamental questions about the mechanisms underlying such adaptations and how they might have evolved. It is a poor model in that it probably reflects no physiological challenge that humans will experience. It seems probable that each rat achieves its own equilibrium at different levels of plasma [Na⁺]; if so, this is very interesting, but to test this would require extensive studies that I am afraid are past our capacity to perform.

The meta-analyses of literature on plasma osmolality, haematocrit and plasma vasopressin and oxytocin is "overkill" and better suited to a review article. They should be removed from methods, and results and the appropriate ranges of values given and referenced in the Discussion when comparing the current results, by citing only the most appropriate of the references.

There seems no objective way of deciding which references are 'most appropriate' given the massive unexplained variability between previous studies. Plasma hormone measurements vary by assay conditions and different authors have measured concentrations at different timepoints. Plasma osmolality measures vary enormously in salt-loaded rats (as do plasma sodium measurements, which we did not address). The source of this variability is obscure, it is not a factor of age, gender or strain of rats as far as we can see from the studies we found; the salt intake in the diet may vary but this detail is generally not given. We have previously strongly criticised studies for citation bias – for "cherry picking" references to support their claims, and here we have tried hard to avoid any suggestion of this.

In Results, the number of animals used in some experimental groups is not given. In Methods, it states that "there was a large number of animals in each experimental group". This is an open-ended statement and does not provide sufficient information on numbers which should be stated in results in each section where animal numbers are missing.

We have been clearer about animal numbers, and have inserted them in Tables, figure legends or text as seemed most appropriate. We have made all primary data and all analyses openly available in a form that makes it transparent when data (cells) come from the same rat.

In the legend Figure 1, it is stated that there was stimulation of OVLT, but the technique used is not given in Methods.

Thanks; we have added this to the Methods (lines 261—263)

In Fig. 5G, the fawn and brown colours are rather similar could be better differentiated.

We have changed this

On the third page of Discussion, line 25, I think "hypertonicity" rather than "hypotonicity" is meant. Minor comments.

1. Throughout, 2% NaCl should have a space (not 2%NaCL).
2. Where first used, abbreviations should be specified e.g. interspike interval (ISI), hyperpolarizing afterpotential (HAP) and DAP.
3. Page 6, line 20. A word is missing between "made --- an Axopatch"
4. Page 7, 3rd line from bottom, "all".
5. Page 10, 2 nd line. A reference should be cited for total body water.

We have attended to all of these – thanks for the careful attention to detail

Referee #2:

The title conveys two messages, (i) chronic salt loading is associated with reduced responsiveness of magnocellular neurons, and (ii) this reduced responsiveness can be rationalized to constitute avoidance of futile hypersecretion. The former is a clear statement open to assessment. The latter is less comprehensible. Notably, the title does not reveal that the manuscript includes a (pointed, but detailed) literature review of 36 studies in rats drinking 2% NaCl for 2 days or more.

We have replaced the title with a simpler one: we refer to the literature review in the abstract.

The introduction is extensive (1000+ words) and begins with the description of the elevation in blood pressure occurring in rats exposed with 2% saline as the only drinking water. The modest elevation in systolic blood pressure seen under these circumstances (+15 mmHg is provided as a frequent observation) probably has little to do with the normal concept of salt-sensitive hypertension.

We have trimmed the introduction to ~600 words. On the issue of hypertension, we agree with the referee, though obviously some authors have taken a different view. Our approach was not to question whether the rise in blood pressure in salt-loaded rats is analogous to salt-sensitive hypertension, but to test whether it is plausibly a consequence of hyperactivation of vasopressin cells. We have shown (from our data) that they are not hyperactivated and (from a review of the literature) that vasopressin secretion is only modestly increased, in line with the electrophysiological data.

The authors describe that (i) 2% saline is aversive to normal rats, (ii), prolonged exposure to it generates 'excessive thirst', and (iii) literature data indicate that plasma osmolality in this situation increases by some 35 mOsm/kg (which is a lot). This means that the rats for days were kept under conditions which were very unpleasant for them. In this context, a 15 mmHg increase in systolic blood pressure seems unexpectedly small and fully explainable by the stressful situation. Remarkably there is no indication in the manuscript acknowledging that the rats were kept under continuously stressful conditions for a week. The salt loading increased plasma sodium concentrations from 136 to 174 mmol/L indicating that the increase in plasma osmolality was higher than 35 mOsm/kg. Potentially, there is imbalance between the unquestionable distress of the animals and the potential advancement of science provided by the results.

This is an important issue, and we have addressed it in detail in the revised ms. We surveyed more than 80 primary research papers using this model, published over 50 years; these have come from most of the major research groups that have studied the hypothalamo-neurohypophysial system. In early studies, rats were maintained for up to 90 days on 2% NaCl (e.g., Dellmann et al. (1988) *Neuroendocrinology* 47:335-42). It is often assumed that salt loading is a chronic stress, but there

is no sign that the stress axis is activated (see lines 761-766)– this is in contrast to dehydration caused by removal of drinking water which causes a massive rise in CRH expression, as first noted by Scott Young (1986; Corticotropin-releasing factor mRNA in the hypothalamus is affected differentially by drinking saline and by dehydration. FEBS Letters 208, 158-162) (in the revised ms we have quoted a later study from Young). In most studies, rats initially reduce their food intake (by 5-10%) and lose a similar proportion of body weight but in some studies they have recovered to baseline levels within 7 days (though it seems they always lag behind the body weight of rats maintained on tap water throughout). The lower food intake may not be a symptom of stress i; a lower food intake seems to accompany chronic hypernatremia generally, and has been suggested to be a homeostatic adaptation to limit salt intake (see e.g. Verbalis et al. Regul Pept. 1993 29;45:149-54), via the central effects of oxytocin release (which suppresses salt appetite and food intake).

*Euhydrated rats find 2% NaCl aversive (by a behavioural conditioned avoidance test), but they prefer isotonic saline to water; whether they still find it unpleasant after days of drinking it is unknown. As we now state in the Discussion (lines 751-760) “Norway rats (*Rattus norvegicus*, from which laboratory rats are derived) have colonised a wide range of environments (Modlinska & Pisula, 2020), including estuaries and islands with no standing sources of fresh water. They are believed to have spread globally as stowaways on ships, with bilge water – a mix of fresh water and seawater - the only available source of fluid.”*

In the presentation and discussion of the experimental and literature parts of the present study, the experiments should be given priority. The literature survey should be appreciated in light of the new data.

We have changed the order of presentation accordingly

Numerically, the key results are remarkable. In salt loaded rats, plasma vasopressin levels were 2.3-fold higher than in euhydrated controls (2.5 vs.1.1 pg/ml). The corresponding firing rates for supraoptic neurons averaged 6.0 and 5.1 spikes/s (cf. legend to Fig. 3: "Supraoptic cells of all types fire ~1 spikes s⁻¹ faster after 7 days of salt loading and recover after rehydration."). Under the (reasonable) assumption that the total body clearance of plasma vasopressin is not markedly changed by salt loading, the discrepancy between the increases in firing frequency and plasma vasopressin is puzzling, particularly in light of the expected finding that pituitary stores of vasopressin were markedly reduced during salt loading.

We have extended the discussion to address this point in a revised section of the discussion

(lines 652-661). The relationship between firing rate and vasopressin secretion is linear (in the relevant range), but has a steep slope. We have studied the relationship between spike activity and

secretion very extensively in previous studies, as have a number of other groups, and those previous findings were aggregated in the computational model that we refer to.

The section in its entirety now reads: “Is an increased firing rate of 1 spike s⁻¹ consistent with the observed increase in vasopressin secretion?”

MacGregor & Leng (2013) fitted computational models of vasopressin cell activity and stimulus-secretion coupling to a range of published data. The combined model (in Figs 8E and F of their paper) allows the average spiking frequency of a heterogeneous population of model phasic cells to be linked to predicted plasma concentration; in the range 1-6 spikes s⁻¹ the relationship is linear, and secretion increases by ~6.4 pg ml⁻¹ for every increase of 1 spike s⁻¹. In the present study, vasopressin cells discharged 0.9 spikes s⁻¹ faster in salt-loaded rats than in euhydrated rats, and, across the published literature, plasma [vasopressin] is 6.6 pg ml⁻¹ higher, in agreement with the model.”

Discussion: Several comments are warranted: (i) ' However, the present results suggest that either the excitability of supraoptic neurones adapts during sustained hyperosmolality, or their osmoresponsiveness is reduced.' If it is assumed, the supraoptic neurons themselves are not the osmosensitive units, what is the differences between excitability and osmoresponsiveness?

This assumption is false. Supraoptic neurons receive both direct and indirect inputs from osmosensitive neurons in forebrain regions, but they are also intrinsically osmosensitive; the principal underlying mechanism (involving stretch-sensitive Trp channels that signal osmotically-induced cell volume changes) is now well established (see Prager-Khoutorsky M & Bourque CW. (2015). J Neuroendocrinol 27, 507-515 for a review). Accordingly, there are two ways to explain the unexpectedly modest elevation of firing rate in salt-loaded rats: either the intrinsic excitability of supraoptic neurons is reduced (for example by an increase in membrane K⁺ conductance), or mechanisms of osmoresponsiveness are attenuated (e.g. by down regulation of Trp channels or adaptation of mechanical coupling to the channels). See lines 591-604)

(ii) Vasopressin numerics: 'It seems impossible to obtain interpretable measures of plasma vasopressin in the same experimental conditions as the present study, because urethane anaesthesia impairs vasopressin and oxytocin clearance.' Is this correct? It is reasonable to assume a priori that metabolic clearance of vasopressin is much greater than the renal clearance, and numbers could be discussed.

We have added the following to the text (lines 135-141): “i.p. urethane reduces blood flow through the kidneys and blocks urine production (Severs et al., 1981). As between one third and one half of the metabolic clearance of vasopressin occurs via the kidneys (Rabkin et al., 1979), the elevated [vasopressin] in urethane-anaesthetised rats probably reflects at least in part) reduced metabolic clearance.”

(iii) The authors handle oxytocin neurons and oxytocin secretion as if it is self-evident that oxytocin should be secreted in response to salt loading. Not so to the present reviewer.

Thanks for pointing this out; we should have realised that these important facts might not be generally known. We have included a few lines to the introduction (lines 87-91): “In the rat, oxytocin neurones are activated by systemic osmotic challenge to a similar extent as vasopressin cells (Leng & Russell, 2019); in rodents, oxytocin is a potent natriuretic factor at physiological concentrations (Verbalis et al., 1991), so we expected that their activity would also be elevated in salt-loaded rats.”

(iv) several new paragraphs are not immediately appearing as self-contained trains of thoughts/arguments (do not use 'this' in first sentence of new paragraph).

Thanks for the advice, which we have tried to follow.

(v) 'hypotonicity': rather 'hypertonicity'.

Thank you for spotting this typo

(vi) is it possible that neurohypophysial tissue normally provides both a passing way and a parking lot for vasopressin, and that under chronic salt loading just the parking lot is empty?

Yes, indeed that is what we believe: it is currently understood that newly synthesised neurosecretory granules, on arriving at the secretory organelles in the neurohypophysis (the nerve terminals and axonal swellings) go first to the juxta-membrane sites that form a readily-releasable pool, and if not released, are transferred to the central reserve pools (from which they can be trafficked back into the readily-releasable pool). Eventually, if they are not released, they will be moved from the reserve pool into a non-releasable pool thought to occur in the largest axonal swellings (the Herring bodies), where they are dismantled for recycling. Thus in the salt-loaded state, it seems that most secretion is of newly synthesised granules with little reserve capacity for responding to acute challenges, because the reserve secretory pool is drastically depleted. However we have no direct evidence for this hypothesis so thought it beyond the scope of this paper

The attitude of the editor to the inclusion of a review type section in an original, hypothesis driven, project-based manuscript could be either 'this does not belong here' or 'this is a model manuscript'. Considering the present, highly questionable biomedical publishing environment, this reviewer favours the second option. Addition of the review section adds valuable perspectives to the study.

Thank you for these comments!

Minor issues

Introduction: 'futile waste': a pleonasm, waste is futile

We have removed this

Acute hyperosmotic stimulation with 2M NaCl: '0.91 mg of NaCl': please use mmol

Discussion: 2 x 'per 100 mg infused': please use mmol

We have changed to mmol

Referee #3:

1. In summary, the authors write: "We tested the hypothesis that salt loading, induced by replacing drinking water with 2% NaCl, hyperactivates these neurons leading to hypersecretion and salt-dependent hypertension." However, this statement should be reconsidered, since the authors did not measure neither hormonal secretion nor hypertension.

We have removed this phrasing

2. In the introduction, it is mentioned: "In humans, as summarised by Verbalis, plasma osmolality is normally between 280 and 295 mOsm/kg; at ~295 mOsm/kg the urine is maximally concentrated, and 'the entire physiologic range of urine osmolality is accomplished by relatively small changes in plasma vasopressin of 0 to 5 pg/ml' (Verbalis, 2018)." This affirmation is supposed to apply in the same way for the rats. But on what the authors did based themselves for to say this? Why this should apply also to the rat? Because rats dispose of a more large capacity for concentrate the urine compared to the human (2-3 folds), it seems very probably that the physiological range of osmolality which can be regulated by vasopressin is much more large in rat. So, this affirmation, and also the calculation of the "putative" maximal efficient concentration of AVP, are not really supported and should be modified. Notably, the estimation of what is the maximal "efficient" concentration of AVP is presented like the main rationale of the study and of the conclusions, and it is actually misleading.

We have addressed these comments in the revised discussion (lines 740-750), citing classic studies from Heinz Valtin and colleagues and from Gary Robertson and colleagues.

"In humans, plasma osmolality is normally between 280 and 295 mOsm kg⁻¹; at ~295 mOsm kg⁻¹ the urine is maximally concentrated, and "the entire physiologic range of urine osmolality is accomplished by relatively small changes in plasma vasopressin of 0 to 5 pg ml⁻¹" (Verbalis, 2018). In these respects, rats do not appear to be very different. For example, Gellai et al. (1984) studied the renal effects of infusing vasopressin to Brattleboro rats deficient in endogenous vasopressin, finding that a plasma concentration of 8 pg ml⁻¹ "may be approaching the level where vasopressin induces maximal water permeability of the late distal tubules and collecting ducts". In their classic study, Dunn et al. (1973) reported that plasma [vasopressin] in conscious rats reaches ~10 pg ml⁻¹ when plasma osmolality is raised by ~10 mOsm kg⁻¹"

3- Further, the authors indicate that "plasma vasopressin by well-validated radioimmunoassays in rats after salt-loading for {greater than or equal to}3 days; the mean vasopressin concentration in

euhydrated rats was 1.8 pg/ml, rising to 7.3 pg/ml after salt-loading." These values appear not coherent with the previous publications of the same group (for example Ludwig et al, 1996), where it was reported that plasma AVP is increased until 20.7 pg/ml already after 2 days of salt-loading.

We did not include these data because they were obtained in urethane-anesthetised rats but we have now addressed them in the revised discussion as follows (lines 623-631):

“Choe et al. (2015) asserted that the elevation of firing rate that they observed was “consistent with the increased vasopressin secretion observed after [salt loading]”, quoting (Ludwig et al., 1996), who had reported that, under urethane-anaesthesia, plasma [vasopressin] was 11.5 pg ml⁻¹ euhydrated rats, and 20.7 pg ml⁻¹ in rats that had drunk 2% NaCl for 2 days, by which time plasma [Na⁺] had increased by 14 mM (28 mOsm kg⁻¹). These data are not included in Table 7 because of the impact of urethane on plasma [vasopressin] (see Methods); the 14 studies in Table 7 all measured plasma [vasopressin] by radioimmunoassays in blood samples taken without confounding anaesthetic effects, with a study mean [vasopressin] in salt-loaded rats of 8.7 pg ml⁻¹”

More, they also shown that an acute i.p. injection of hypertonic saline increased AVP plasma level from 11.5{plus minus}2.4 to 75.4{plus minus}15.5 pg/ml (control, P<0.05), and that in salt-loaded animals the increase was much more strong: from 20.7{plus minus}5.2 to 113.7{plus minus}19.3 pg/ml (salt-loaded, P<0.05). This contradict also the affirmation that "Maximal antidiuresis is achieved at relatively low plasma concentrations (~10 pg/ml)".

These data we also address in the revised discussion as follows (lines 679-684):

Ludwig et al. (1996) reported that (under urethane anaesthesia) an i.p. injection of 3.5 M NaCl (0.6ml/ 100g bw) raised plasma [vasopressin] by 64 pg ml⁻¹ in euhydrated rats and 93 pg ml⁻¹ in 2-day salt-loaded rats – a mean difference of 29 pg ml⁻¹, with a standard error (calculated from the data provided) of 25 pg ml⁻¹. Thus this study (which used an unusually large dose of hypertonic saline) provides no clear evidence of a change in osmoresponsiveness at this early stage of salt loading.”

4. The paper that the authors are citing in the introduction (Dyball & Pountney, 1973) indicate that urethane anaesthesia seems to stimulate the liberation of neurohypophysial hormones. Moreover, it was shown that it can significantly modify the hormonal content of the neural lobe, and increase the plasma concentration of the hormones (Ginsburg & Brown, 1957; Dyball, 1971).

Dyball & Pountney are neutral on this matter: they stated, in full, “Anaesthesia with urethane appears to stimulate the release of the neurohypophysial hormones, since it both alters the ratio of oxytocin to vasopressin in the neural lobe (Ginsburg & Brown, 1957) and is associated with increased plasma concentrations of the hormones (Ginsburg & Brown, 1957; Dyball, 1971). On the other hand, Cross & Dyer (1971) found that the firing rate of hypothalamic units in diencephalic

islands was not affected by urethane and they concluded that the anaesthetic had no direct action on the discharge of hypothalamic neurones.”

...The authors are attributing this effect to a reduce clearance by the urine, but alternatively, it can be the result of the known effect of urethane to raise in the systemic osmolality. This point should be more addressed by the authors.

We have addressed this carefully in the Methods section, lines 129-139 as follows:

“Anaesthetic doses of urethane raise systemic osmolality by 10-15 mOsmol kg⁻¹ and it is commonly assumed that this explains the increased hormone levels. However, transduction of osmotic pressure changes by osmoreceptors depends on stretch-sensitive membrane receptors that signal volume changes (Prager-Khoutorsky & Bourque, 2015); as urethane is readily soluble in water and lipids (Field & Lang, 1988) it will cross cell-membranes freely, and therefore should not produce a sustained osmotic pressure gradient across cell membranes. However, i.p. urethane reduces blood flow through the kidneys and blocks urine production (Severs et al., 1981). As between one third and one half of the metabolic clearance of vasopressin occurs via the kidneys (Rabkin et al., 1979), the elevated [vasopressin] in urethane-anaesthetised rats probably reflects at least in part) reduced metabolic clearance”

5. With regards to authors conclusions that basal rates show very little increases in salt-loaded animals. This is a central point that should be clarified, since this represents a major contradiction with previous studies. For example, an early study by (Dyball & Pountney 1973) show that salt-loading increase the firing rate of neurons in the SON and PVN. ... In the (Dyball & Pountney 1973), they demonstrated after 3 days of salt-loading, that in phasic SON cells (putative AVP neurons), the frequency of firing in the bursts was largely increased from ~5-6Hz to ~8-9Hz.

Dyball and Pountney’s results were remarkably similar to ours, given that, at that time, cell firing rates could be estimated only very approximately from polygraph records. We now spell their findings out in the discussion as follows (lines 612-618):

“Dyball & Pountney (1973) reported that the median firing rate of supraoptic neurones increased from between 2 and 4 spikes s⁻¹ in euhydrated rats (n=69 cells) to > 4 spikes s⁻¹ in 3-day salt-loaded rats (again 69 cells), while the intraburst firing rate of phasic cells increased from ~5 to ~8 spikes s⁻¹. In the present study the corresponding levels were similar: the median firing rate was 4.8 spikes s⁻¹ in euhydrated rats and 5.6 spikes s⁻¹ in salt-loaded rats, while the mean intraburst firing rate in phasic cells increased from 5.5 to 8.6 spikes s⁻¹”

This is consistent with the more recent evidence from in vivo study by Choe et al, showing that basal firing increase of AVP neurons from ~5.5 to ~9.4Hz after 7 days of salt-loading. The authors should discuss why their results deviate so much from these previous studies?

The data from Choe et al are different from both ours (as we had made clear) and Dyball and Pountney. It was a smaller study than either of these (Choe et al.: 13 cells in euhydrated rats, 21 in salt-loaded rats), but even so the discrepancy is surprising. We draw attention to the discrepancy in lines 618-621, stating “Choe et al. (2015), in a relatively small study, gave no data for oxytocin cells but reported a larger increase in mean firing rate of 3.9 spikes s⁻¹ in vasopressin cells after 7 days of salt loading. This seems inconsistent with the outcomes of the present study.”

6. For the experiment where microdialysis probe was used to apply drugs, what was the osmolality of aCSF measured and adjusted whether the recordings were made from control or salt-loaded rats?

The microdialysis experiments were performed only in euhydrated rats. The osmolality of the aCSF can be calculated from the recipe; we did not measure it. The recipe used is as used previously; it is modestly hypertonic, but it should be noted that this was used as a microdialysate, not for icv injection; very little penetrates the extracellular fluid, and in past experiments we have never noticed a change in the firing rate of supraoptic neurones when the infusion pump is switched on.

7. For hypertonic stimulations, the authors expend on theoretical calculation based on the amount 2M NaCl injected, animals weight, and the fact that urethane anesthesia attenuate urine production (and thus sodium excretion). Based on these calculations, the putative increase in osmolality is ~14 mOsm/kg, which is relatively small increased. However, given all the assumption, especially the assumed effects on sodium excretion, it is unclear what the stimuli actually were. Furthermore, the effect of urethane on urine production might differ between control salt-loaded, and hypovolemia, and thus injecting same volume of NaCl can produce different changes in plasma osmolality. The authors should measure blood osmolality in rats in these settings, to provide actual (and not putative) measurements of the hypertonic stimuli in each condition (control, salt-loaded, hypovolemic). One of the most surprising statements of the authors, that contradicts previous studies, is that neurons from salt-loaded animals are not more but less osmosensitive. This conclusion simply cannot be drawn unless it has been measured what osmolality changes were triggered by the manipulation.

These measurements have been reported by us in a previous study using exactly this protocol and were referred to alongside the calculation – our point being that theoretical calculation and experimental findings were in very close agreement. See lines 255-276 containing the sentence “Leng et al. (2001) measured plasma [Na⁺] while infusing 2M NaCl at various rates and concluded that plasma [Na⁺] increased by 6.1 mM per mmol infused, close to the level expected from these theoretical considerations.”

8. The main conclusion of the study, claiming that AVP (and OXT) neurons are not more osmosensitive in salt-loading, should be discussed in the context of methodological limitations that are associated with in vivo recordings conducted under urethane anaesthesia. It has been well

established that urethane anaesthesia in rats cause pronounced plasma hyperosmolality of body fluids without affecting plasma sodium levels. This suggests that urethane interferes with the body's ability to regulate water balance, increase in plasma osmolality, potentially impacting AVP release in response to hyperosmolality (e.g. PMID: 7022489, 20470865). These findings indicate that urethane anaesthesia modifies the body's normal response to osmotic challenges, potentially affecting AVP neuron activity. This represents a major limitation of studies on fluid homeostasis conducted in vivo but using urethane anaesthesia.

See above. Urethane is lipid permeable as well as water permeable, so it will not exert osmotic pressure on cells (urea also exerts no osmotic pressure as it too crosses cell membranes, and hence does not activate osmoreceptors, as is well established). Certainly anaesthesia blocks urine production and impairs vasopressin clearance, but there is no evidence that this affects vasopressin cell activity: vasopressin does not cross the blood-brain barrier so elevated plasma levels do not affect vasopressin cell activity.

The authors should discuss this limitation of their study, and also discuss whether alternative methods can be implicated and beneficial (in vivo, ex vivo, or in vitro).

The limitations of the salt-loading model are we think, implicit in the discussion of this model in the new section (lines 740-790). It would certainly be preferable to have conducted this study in conscious rats, but that, at present at least, is in practice beyond present technology. The advantages and limitations of urethane are discussed in lines 128-140. The uncertainties about cell identification are discussed in lines 570-579. The limitations of *in vitro* studies for this purpose are apparent in our discussion of the internally inconsistent in vitro evidence on GABA (lines 715-735). We believe we have been open about discrepancies with other findings (e.g. lines 636-641 where we discuss discrepant vasopressin measurements) and discrepancies described above.

9. Related to the point#8, previous study focusing on the similar question, utilized an approach that circumvent using urethane anaesthesia (Levi et al 2021). Instead, they use estimated neuronal activity using c-Fos... The authors should incorporate this additional evidence and discuss their results in this context.

Fos expression is not a linear reporter of cell activity and is subject to subjective determination of a threshold. There are also issues in statistical analysis – Fos counts in the SON are not normally distributed; at low levels of population activation they show a log-normal distribution of counts, while at high levels there is a clear ceiling effect as activation extends to the whole population. Thus it is excellent for categorical conclusions – is the system activated or not – but very weak at distinguishing between levels of activation. We have discussed this as follows (lines 687-703)

“Levi et al. (2021) avoided these confounds by injecting conscious rats s.c. with 2M NaCl at 1.43 ml kg⁻¹ and measured Fos expression in rats killed 1 h later. This induced expression of Fos

immunoreactivity in ~90% of vasopressin cells in salt-loaded rats compared to ~65% in euhydrated rats. However, Fos immunoreactivity is a non-linear reporter of neuronal activation; for any single cell, Fos is detected or not, but how many cells express detectable Fos depends on the tissue processing – especially the antibody concentration, the time and temperature of incubation, and the observer’s criteria for ‘clear’ Fos expression. Studies of Fos expression thus process control and experimental tissues together, stop the incubation of tissue at a point governed by the observed expression, and conduct analyses blind. In Levi et al. (2021) there was little difference in Fos expression between euhydrated and salt-loaded rats given a control injection. By contrast, Miyata et al. (2001) reported a high level of Fos expression in the supraoptic nucleus after 2 and 5 days of salt loading compared with euhydrated rats. Thus, it appears possible that, in Levi et al. (2021), there was an increase in Fos expression in vasopressin cells in salt-loaded rats that was below the detection threshold– but an acute stimulus, by inducing an increment in Fos expression above this elevated baseline, produced an exaggerated increase in detected Fos immunoreactivity.

Minor comments:

1. For key points - the reviewer is unsure whether the key points should summarize the results of the current study or provide more general statements relevant to the study? This is since the first key point does not reflect the results of the current study, but concepts which were established by decade of previous studies, which, however, were more systematically summarized by the authors. I am not sure whether this is appropriate for the "key points" and perhaps more suitable for the abstract or discussion.

We felt that it was necessary to provide context for the subsequent key points, but will follow the editor’s advice

2. Moreover, this statement that depletion of the pituitaire content of AVP and OXT "apparently reflecting enhanced secretion" is not supported by the authors data.

We show that the change in firing rate that we see, though small, is consistent with the observed increase in plasma vasopressin and that his increase is consistent with the observed depletion of pituitary content. See the sections “Is an increased firing rate of 1 spike s⁻¹ consistent with the observed increase in vasopressin secretion?” lines 652-661

“Are the reported levels of vasopressin and oxytocin consistent with the observed depletion of pituitary content?” lines 662-677

3. The key point statement "These findings contradict the prevailing understanding that salt-loading promotes a pathological hyperexcitability and exaggerated vasopressin secretion..." should be modified, unless the authors provide evidence where previous studies interpreted increased osmoresponsiveness as "pathological hyperexcitability".

We have changed this

4. In the introduction: "in salt-loaded rats as a result of increased intracellular [Cl⁻] resulting from upregulation of the Na⁺/K⁺/Cl⁻ cotransporter1 (Balapattabi et al., 2019)" - this is a wrong reference. Kim et al showed that but they didn't use salt-loading but a different model of salt-dependent hypertension.

Thanks – we have amended this: Balapattabi et al. (2019) concluded from their study that “Salt-loading increases [Cl⁻]_i in SON AVP neurones via a TrKB-KCC2-NKCC1-dependent mechanism.” but they did not directly show upregulation. Kim et al. did use the salt loading model to show this - they state “the chronic hyperosmotic stress produced by 7 d of high Na⁺ drinking water produced a profound increase in the level of NKCC1 protein (137.0±23.1%, n= 8) in the SON.” Accordingly, our revised ms (lines 719-721) now reads: “Kim et al. (2011) reported that the switch was accompanied by an increase in NKCC1 protein expression in the supraoptic nucleus and a relatively small increase in NKCC2 expression, and that bumetanide reversed the switch.”

5. Also in the discussion, the authors refer the Choe et al 2015 paper, stating that "However, Choe et al. reported that this switch was accompanied by a downregulation of NKCC1 expression in vasopressin cells with no change in NKCC2 expression, and that the NKCC1 blocker bumetanide was without effect (Choe et al., 2015). This is incorrect statement, since in this paper, Choe et al reported that "our electrophysiological analysis with blockers of KCC2 and NKCC1 indicated that this effect is due to a SL-mediated reduction in KCC2 activity, rather than an increase in NKCC1 activity". In fact, they have also show using western blot that while KCC2 levels decreased in salt-loading, while NKCC1 levels are not significantly affected.

Thanks. The relevant section now reads (lines 716-719) “Choe et al. (2015) reported that this switch was accompanied by a downregulation of NKCC2 expression in vasopressin cells with no significant change in NKCC1 expression, and that the NKCC1 blocker bumetanide was without effect”

6. In the intro, authors state "plasma vasopressin by well-validated radioimmunoassays in rats after salt-loading for {greater than or equal to}3 days; the mean vasopressin concentration in euhydrated rats was 1.8 pg ml⁻¹, rising to 7.3 pg ml⁻¹ after salt-loading" this seem inconsistent with the previous reports of the same group showing that plasma AVP is increased to 20.7 pg/ml already after 2 days of salt loading.

We have addressed this in the revised ms as follows (lines 623-631): “Choe et al. (2015) asserted that the elevation of firing rate that they observed was “consistent with the increased vasopressin secretion observed after [salt loading]”, quoting Ludwig et al. (1996), who had reported that, under urethane-anaesthesia, plasma [vasopressin] was 11.5 pg ml⁻¹ euhydrated rats, and 20.7

pg ml⁻¹ in rats that had drunk 2% NaCl for 2 days, by which time plasma [Na⁺] had increased by 14 mM (28 mOsm kg⁻¹). These data are not included in Table 7 because of the impact of urethane on plasma [vasopressin] (see Methods); the 14 studies in Table 7 all measured plasma [vasopressin] by radioimmunoassays in blood samples taken without confounding anaesthetic effects, with a study mean [vasopressin] in salt-loaded rats of 8.7 pg ml⁻¹.”

Further, they also showed that while an acute "i.p. injection of hypertonic saline increased plasma AVP levels from 11.5{plus minus}2.4 to 75.4{plus minus}15.5 pg/ml (control, P<0.05)" the increase in SL animals was much stronger "from 20.7{plus minus}5.2 to 113.7{plus minus}19.3 pg/ml (salt-loaded, P<0.05)".

We have addressed this in the revised ms as follows (lines 679-684): “Ludwig et al. (1996) reported that (under urethane anaesthesia) an i.p. injection of 3.5 M NaCl (0.6ml/ 100g bw) raised plasma [vasopressin] by 64 pg ml⁻¹ in euhydrated rats and 93 pg ml⁻¹ in 2-day salt-loaded rats – a mean difference of 29 pg ml⁻¹, with a standard error (calculated from the data provided) of 25 pg ml⁻¹. Thus this study (which used an unusually large dose of hypertonic saline) provides no clear evidence of a change in osmoresponsiveness at this early stage of salt loading.”

This contradicts to the statement that "Maximal antidiuresis is achieved at relatively low plasma concentrations (~10 pg ml⁻¹),"

Antidiuresis wasn't measured in these studies, so there is no contradiction. It was measured in the study of Gellai et al. who infused vasopressin at different concentrations in conscious rats, concluding that a plasma concentration of 8 mM was “close to the level where vasopressin induces maximal water permeability at the kidney” (quoted in lines 781-782)

7. In tables summarizing values from previous studies reporting posterior pituitaire levels of AVP and OXT. These studies used different durations of salt-loading, however the authors averaged experiments with salt-loading lasting 5, 6, and 7 days. Why different salt-loading lengths were group together? These analyses should be done for different days. Also, some experiments of 8 days and 5 days were excluded. It's unclear why.

There wasn't much literature data from day 7 for oxytocin so we broadened the window for this. We show all the data in the new Figure 7C, and in Table 6 we calculate averages of all data on rats salt-loaded for 5-8 days (there are no studies at 9 days).

Similar problems with the other tables summarizing values from previous studies, i.e. reporting AVP and OXT plasma concentrations. It seems like random days were picked from different studies and averaged together.

Thanks; it's important to exclude any suggestion of any cherry-picking of the data. All tables

showed (and still show) all published data from every day measured; we quoted averaged data as illustrative of the trends, but now we display the data as panels in Fig.7. We think that this shows clearly that there is no evidence of progressively increasing plasma concentrations of vasopressin and oxytocin, and no evidence of a progressive increase in haematocrit, while displaying the variability between studies (which is much greater than the variability within studies).

Professor Mike Ludwig

Professor of Neurophysiology
Centre for Discovery Brain Sciences
University of Edinburgh
George Square
Edinburgh EH8 9XD, UK

Tel: -44 (0) 131 650 3275

Fax: -44 (0) 131 650 6527

email: mike.ludwig@ed.ac.uk

28/03/2025

Dear Editor,

Apart from the responses to the referees, we have also revised the figures to meet the request that we show all data as means (SD) not means (SEM) and to show data as points where $n < 30$.

The last issue has presented some difficulties of practicality as follows:

Fig 6 shows waveforms as means with SD and in each case the n values are between 17 and 21. However each waveform is generated at a resolution of 0.1 ms so comprises 100 values; hence to include the data as points would involve, for each of the 6 panels, plotting at least 1700 points most of which would fall within the ranges shown by the SD; they would be effectively indistinguishable. Moreover, even if resolvable, these points would be incompletely meaningful as they are not independent - but to show their interdependence would require plotting not 1700-2100 points but overlaying 17-21 waveforms that overlap extensively. Please note that on our open access site we are providing all the raw waveform data for every cell; I don't really see how we could do better.

Fig 4 E, F and G has similar issues – here the n values are 20, 21 and 23; we have plotted means with SD for 11 time points - to show the individual data points would involve plotting 220 or more points in each panel, but, as above this, because the points are not independent it would be important to connect the points with lines – rendering the figure unintelligible. Again, we have given all the raw data (including all the underlying raw spike data at a time resolution of 0.1 ms for the 35 min durations of recordings) on our open data site.

We hope that these changes are to your satisfaction.

With best wishes

Mike Ludwig
Professor of Neurophysiology

Dear Dr Ludwig,

Re: JP-RP-2025-288860R1 "**Reduced osmoresponsiveness in magnocellular neuroendocrine neurones during chronic salt-loading**" by Maja Lozic, Roongrit Klinjampa, Nancy Sabatier, Duncan J MacGregor, Gareth Leng, and Mike Ludwig

Thank you for submitting your manuscript to The Journal of Physiology. It has been assessed by a Reviewing Editor and by 3 expert referees and we are pleased to tell you that it is acceptable for publication following satisfactory revision.

REVISION CHECKLIST:

We look forward to receiving your revised submission.

Yours sincerely,

Vaughan Macefield
Senior Editor
The Journal of Physiology

EDITOR COMMENTS

Reviewing Editor:

Thank you for your revised manuscript and thoughtful responses to the reviewers' comments. We appreciate the substantial revisions made to improve the clarity and overall quality of your work.

The reviewers have largely acknowledged the originality of your study and the relevance of the findings. Reviewer 1 considers the manuscript to be in good shape overall and leaves the decision about retaining the meta-analyses to editorial discretion. The Reviewing Editor is favourable to maintaining the meta-analytical data, as long as its limitations are clearly acknowledged.

However, several important concerns remain, particularly those raised by Reviewer 3 regarding methodological limitations. In particular, it is essential that the final version of the manuscript explicitly addresses the caveats associated with the use of urethane anesthesia, which significantly affects plasma AVP levels and potentially confounds interpretation of physiological relevance. This issue must be acknowledged in the discussion, and conclusions should be adjusted accordingly to reflect these limitations.

In addition, please incorporate a more balanced and constructive discussion of how your findings align or contrast with existing literature, rather than broadly dismissing previous studies. It is also necessary to address the absence of key physiological measurements such as plasma osmolality, hematocrit, and drinking behaviour, at least by discussing how this impacts interpretation and generalisability.

Reviewer 2 has also raised thoughtful points about the applicability of your findings to other species, the influence of anaesthesia, and the osmoregulatory dynamics. We ask that you respond to these by strengthening the discussion with appropriate caveats and justifications, as applicable.

We invite you to submit a further revised version of the manuscript that addresses these methodological concerns and clarifies the interpretation of the data in light of the limitations described above.

Senior Editor:

Thank you for submitting your revised manuscript to the Journal of Physiology. As you will see from the comments of the Reviewing Editor and the three independent reviewers, all experts in the field, while it is acknowledged that you attended to many of the concerns already raised, there are significant concerns that you will need to address before we can proceed. In particular, the methodological limitations in the use of urethane, which - as pointed out by Reviewer 2 - significantly elevates plasma AVP to non-physiological levels (~20.7 pg/ml), far above what is observed with salt loading alone (~8.7 pg/ml). Despite this, the authors interpret their findings as if they reflect normal physiological conditions, which cannot be the case. Please discuss the limitations of the use of urethane and other issues summarised by the Reviewing Editor. I look forward to receiving a revised version of the manuscript in due course.

REFEREE COMMENTS

Referee #1:

This study has been extensively revised largely in response to the reviewers comments. It is an original study that will contribute significantly to the field. The methodology is adequate and state of the art. The conclusions drawn are warranted. The revision appears to be adequate, and while I still am not convinced that a meta-analysis of the literature is necessary here (and sets an unwieldy precedent), another reviewer (rev 2) does not favour my view so I will leave it for the Editor's judgement. Otherwise the manuscript is in good shape.

Referee #2:

The authors have responded appropriately to most of my comments. The new discussion, however, seems almost entirely focused on surgically prepared rats and perhaps a few caveats with regard to general applicability of the results are warranted.

Firstly, the influence of the (inevitable) use of anaesthesia could be an issue as anaesthesia augments vasopressin secretion; in this context, it could be argued that the single digit plasma levels demonstrate that such increase is relatively modest under the present conditions. However, the issue is pertinent.

Secondly, it is argued that rats respond to osmotic stimuli by secretion of vasopressin as well as oxytocin. It is not discussed to what extent (it is known that) this applies to other species for which the rat serves as a convenient model.

Thirdly, the information about the activity of the supraoptic neurons being governed by input from separate osmosensitive units as well as by an intrinsic osmosensitive mechanism may deserve some reservation. As discussed, under physiological conditions the osmoreceptor function is exquisitely sensitive, the data presented here indicate that a full physiological regulatory range (delta plasma osmolality 10 mOsm/kg) is covered by small changes in neuronal activity (delta spike activity 2 spikes/sec. From a system dynamics point of view, it seems unlikely that several osmoregulatory inputs cooperate to provide this steep relationship, but it is not unlikely that several inputs participate during more robust challenges. Arguments in favor of the double input control under physiological conditions would improve the discussion of the present (otherwise impressive) model results.

Referee #3:

The authors did not adequately address my previous criticism, and most of the points which I have raised still remain without being addressed.

The authors have realised their experiments in rats are under urethane anaesthesia, but they interpret their data with values taken from "blood samples without confounding anaesthetic effects." This is putting in evidence a very important methodological problem: the utilisation of urethane anaesthesia, which by itself is significantly modifying the physiological parameters. According to their own anterior works, the level of plasma AVP under urethane can attain 20.7 pg/ml, whereas in animals without anaesthesia it is only 8.4 pg/ml. This is showing that urethane is provoking a very strong, non-physiological liberation of AVP.

Moreover, in physiological conditions, salt loading increases the AVP only to ~8.7 pg/ml. In comparison, their experiments with salt loading and urethane-give values which are around 2.5 times more. Because of that, their interpretations and conclusions must consider that the data are coming from a condition which is not physiological.

It is equally important to take in account that the observed effects like the decrease of osmosensitivity, the depletion of AVP from the pituitaire, and the modifications of firing of AVP and OT neurons, can not be interpreted like normal physiological response to salt loading. These could be the result of artificial conditions where salt loading is added on a major osmotic stimulus - the urethane. To reach 20.7 pg/ml AVP in plasma is indicating an over-activation of the system, which could provoke a depletion of hormonal stock or a diminution of reactivity of magnocellular neurons. Such phenomena would probably not occur in physiological context, even with 7 days of salt loading (which gives ~8.7 pg/ml).

In consequence, the authors must restrict their conclusion to the specific case tested - salt loading in presence of urethane - and must not generalise to physiological conditions. Because urethane has an important effect on AVP in plasma, it cannot be considered as a small confounding factor. It is a principal variable which totally changes the interpretation of the data.

Although the authors say that urethane is not an osmolyte because it is crossing membranes and "should not" modify blood osmolality, their data prove the contrary. Urethane is augmenting osmolality and is inducing a big and probably non-physiological elevation of AVP. The authors admit this fact in the rebuttal, but they do not recognise how this is in contradiction with the validity of their interpretations.

END OF COMMENTS

We thank the editors and reviewers for their careful consideration,

Reviewing Editor:

It is also necessary to address the absence of key physiological measurements such as plasma osmolality, hematocrit, and drinking behaviour, at least by discussing how this impacts interpretation and generalisability.

We have added our own data on body weight and fluid intake to the manuscript (**lines 304-307 and 571-575**) and Fig. 8; these data are similar to published data from others, and were part of an ongoing study, but we include them here in response to the editor's comment above. There is a very large amount of data on osmolality in the literature (Table 7), and a fair amount on haematocrit (Table 6). There is less on $[Na^+]$, but what there is shows the expected relationship to osmolality; we include our own measures of $[Na^+]$.

We have revised the discussion to systematically consider our data in the context of previous studies. This has involved some re-ordering of the previous discussion, and the moving of a paragraph on urethane from the methods where perhaps it had been overlooked into the discussion. We have added detail of literature data where pertinent, and omitted some minor points in the previous discussion that merely reiterated uncontroversial findings in the results.

In the reconstructed discussion we (i) summarise the key outcomes (**lines 578-597**); (ii) compare our data with other electrophysiological data (**lines 598-611**); (iii) report what is known from the literature on hormone levels in salt loaded rats (**lines 612-636**); (iv) show that the firing rate that we recorded is consistent with these values for plasma vasopressin in conscious rats given what we know of stimulus secretion coupling and vasopressin clearance in normal rats (**lines 637-648**); (v) show how that the published values for plasma concentration are consistent with the observed depletion of pituitary content (**lines 649-665**). We then discuss the diverse evidence for changes in GABA actions (**lines 666-700**), the nature of the osmoreceptive mechanism (**lines 701-721**) and the evidence for changes in osmoresponsiveness in vitro vs evidence in vivo, with possible explanations (**lines 722-753**). We then discuss at length the effects of urethane (**lines 754-788**). We end with a section on the physiological implications (lines 789-830).

Beyond this, we have gone through the manuscript to try to ensure that it reads as clearly and concisely as we can make it – most of these changes are minor and we don't detail them. We had inadvertently omitted 3 studies from Tables 5-7 -we have added these and corrected the data in the text; they make no noticeable difference. We have moved the account of hormone measurements made by ELISA from the discussion to the results section, where data from measurements made by RIA are reported (**lines 545-561**).

We address the points raised by the referees in detail below.

Referee #2:

Firstly, the influence of the (inevitable) use of anaesthesia could be an issue as anaesthesia augments vasopressin secretion...

Please see below for a detailed response to this point, which was also raised by the other referee

Secondly, it is argued that rats respond to osmotic stimuli by secretion of vasopressin as well as oxytocin. It is not discussed to what extent (it is known that) this applies to other species for which the rat serves as a convenient model.

This is the case in rats, and (with much sparser evidence) in mice (from mRNA measurements), hamsters, cats and dogs. There is apparently no osmotic stimulation of oxytocin release in goats, pigs or humans. We have now referenced the human data in the introduction (**see lines 84-93, highlighted**), but for other species we don't think there is enough data to be confident.

Thirdly, the information about the activity of the supraoptic neurons being governed by input from separate osmosensitive units as well as by an intrinsic osmosensitive mechanism may deserve some reservation...

This is a very interesting question, but we think one that would be a tangent from the present paper. There is a large literature on what came to be called "the osmoreceptor complex" (see Leng & Russell (2019) The osmosensitiveness of oxytocin and vasopressin neurones: Mechanisms, allostasis and evolution *J Neuroendocrinol.* 31(3):e12662.), Briefly while the supraoptic neurones are depolarised linearly in response to cell shrinkage, they depend on voltage fluctuations produced by synaptic input to translate this into spike activity. Some of these inputs are themselves osmosensitive (and while some are excitatory, others are inhibitory). The osmosensitive inputs have been traced back to the OVLT and subfornical organ, where osmoreceptor cells appear to use the same mechanism as is found in the magnocellular neurones. We have added a brief account of this to the discussion (**lines 714-721**) but have not gone into the implications for system dynamics here, but we addressed them in the cited papers by MacGregor & Leng (**lines 639-648**).

Referee #3:

The authors have realised their experiments in rats are under urethane anaesthesia, but they interpret their data with values taken from "blood samples without confounding anaesthetic effects." This is putting in evidence a very important methodological problem: the utilisation of urethane anaesthesia, which by itself is significantly modifying the physiological parameters. According to their own anterior works, the level of plasma AVP under urethane can attain 20.7 pg/ml, whereas in animals without anaesthesia it is only 8.4 pg/ml. This is showing that urethane is provoking a very strong, non-physiological liberation of AVP.

We have discussed the limitations of urethane in a new section of the discussion (**lines 754-788**, into which we have moved a section on this that was previously in the Methods and perhaps overlooked by the referee. As urethane crosses cell membranes freely, the rise in osmolality that it produces should not activate osmoreceptor mechanisms (as explained in **lines 702-713** by reference to Verney's work). Indeed, if urethane did not freely enter cells it would produce hyponatraemia by withdrawing water from cells, as occurs, for example, after i.p injection of mannitol (Oernbo et al. (2018) *Fluids Barriers CNS.* 15:27).

Certainly urethane augments plasma vasopressin (and oxytocin) concentrations. This might arise either through stimulating the electrical activity of magnocellular neurones, or through reducing the rate of clearance from the blood. As we have detailed, there is no direct evidence of any increase in electrical activity of the neurones after urethane – the only direct comparisons show no difference in activity levels under urethane (**lines 759-763**), and activity levels are the same as under isoflurane, an inhalational anaesthetic which does not raise osmolality: the milk-ejection reflex under isoflurane is indistinguishable from that in conscious animals, and the electrical properties and responses of

supraoptic neurones are the same as under urethane (**lines 764-771**). Urethane appears to raise plasma vasopressin levels by proportionately increasing levels as would be expected of a reduced clearance rate, rather than by raising them by a fixed pedestal, as would be expected of an additional osmotic load.

However, urethane blocks blood flow through the kidneys – a major site of vasopressin clearance. Its effects on the liver, the other major site of clearance, are not known (**lines 779-786**). Blocking blood flow through the kidneys markedly extends the half life of vasopressin and (to a lesser extent) oxytocin in plasma as first shown in 1953 in rats by Heller and Ginsburg who selectively occluding different blood vessels (J Endocr 9:283-291.- “the kidneys accounted for about 50% of the vasopressin cleared and the splanchnic vascular area for at least 40%). Their work was subsequently confirmed by other groups including on isolated perfused kidneys by which the mechanisms were established. Similar experiments to those of Heller and Ginsburg were used by Fabian et al (J Endocr 1969 43: 173-189) to show that the kidneys are responsible for ~30% of oxytocin clearance in rats. Under urethane vasopressin concentrations are typically enhanced by 2-3 fold above those found in conscious rats – but this would be expected from a 50% reduction in the clearance rate by impairing blood flow through the kidneys and liver.

It is equally important to take in account that the observed effects like the decrease of osmosensitivity, the depletion of AVP from the pituitaire, and the modifications of firing of AVP and OT neurons, can not be interpreted like normal physiological response to salt loading. These could be the result of artificial conditions where salt loading is added on a major osmotic stimulus - the urethane. To reach 20.7 pg/ml AVP in plasma is indicating an over-activation of the system, which could provoke a depletion of hormonal stock or a diminution of reactivity of magnocellular neurons.

We think that much of this is covered by the above, but taking specific points :

(a) *“the depletion of AVP from the pituitary...cannot be the physiological response to salt loading”*

The data on depletion of AVP from the pituitary in Table 6 all come from rats decapitated without urethane anaesthesia. The only study of content in urethane anaesthetised rats is the present study – our data are not included in Table 6 for this reason. We found a depletion of vasopressin to 14% of initial content – the data in Table 6 show a mean of 17.6% (95% CI 11-24.2%) at 7 days, not significantly different.

(b) *“To reach 20.7 pg/ml AVP in plasma is indicating an over-activation of the system”*

Urethane raises osmolality by 10-15 mOsm/kg – we have explained that the direct effect of this is minimal because urethane crosses cell membranes freely. In 7-day salt-loaded rats, plasma osmolality is elevated much more than this - by (on average) by ~39 mOsm/kg, accounted for by the increase in plasma [NaCl]. As detailed in **lines 617-624**, hypertonic infusions in conscious rats that raise osmolality to this degree raise plasma concentrations of vasopressin and oxytocin to over 100 pg/ml, and infusions in urethane-anaesthetised rats raise oxytocin levels by a similar amount – the osmotic regulation of vasopressin is linear over a wide range – and in the concurrent presence of hypovolaemia the secretion is further exaggerated (**line 530**). In conscious rats the vasopressin system can secrete vasopressin at a very high rate – and these high concentrations are indeed necessary for it to exert its pressor activity on V1 receptors.

(c) *“an over-activation of the system, which could provoke a depletion of hormonal stock”*

Assuming a normal clearance rate of vasopressin, to maintain a level of 20.7 pg/ml requires a secretion rate of ~40 ng/h, and this will indeed cause a marked depletion of content within a few days, depending on the rate of replenishment, but not on the time scale of electrical experiments. There is no evidence that the depletion of pituitary content affects neuronal excitability: for example, responses of oxytocin cells to CCK in severely depleted salt loaded rats are normal.

(d). .. an over-activation of the system, which could provoke a ... diminution of reactivity of magnocellular neurons.

Vasopressin and oxytocin do not cross the blood-brain barrier and studies involving infusions of oxytocin and vasopressin show no evidence that high plasma levels have any effect on neuronal activity (except in pregnant rats where oxytocin cell activity is stimulated by evoked uterine contractions, and transient effects on vasopressin cells arising from pressor effects).

Although the authors say that urethane is not an osmolyte because it is crossing membranes and "should not" modify blood osmolality, their data prove the contrary. Urethane is augmenting osmolality and is inducing a big and probably non-physiological elevation of AVP. The authors admit this fact in the rebuttal, but they do not recognise how this is in contradiction with the validity of their interpretations.

This is certainly not what we said. Of course, urethane augments osmolality, but because it crosses cell membranes it does not activate osmoreceptors. The mechanism of osmoreception involves cell shrinkage, and only solutes that do not cross cell membranes have this effect – as we now explain by close reference to Verney's classic work. Yes, urethane elevates plasma AVP, and hence we collated published data with values taken from "blood samples without confounding anaesthetic effects." - because the effects of urethane on the kidney impair vasopressin clearance and hence elevate plasma AVP levels. There is however no direct evidence that urethane stimulates increased secretion or that it directly affects AVP cell properties.

Dear Dr Ludwig,

Re: JP-RP-2025-288860R2 "**Reduced osmoresponsiveness in magnocellular neuroendocrine neurones during chronic salt-loading**" by Maja Lozic, Roongrit Klinjampa, Nancy Sabatier, Duncan J MacGregor, Gareth Leng, and Mike Ludwig

Thank you for submitting your manuscript to The Journal of Physiology. It has been assessed by a Reviewing Editor and by 3 expert referees and we are pleased to tell you that it is acceptable for publication following satisfactory revision.

REVISION CHECKLIST:

Please upload two versions of your manuscript text: one with all relevant changes highlighted and one clean version with no changes tracked. The manuscript file should include all tables and figure legends, but each figure/graph should be uploaded as separate, high-resolution files. The journal is now integrated with Wiley's Image Checking service. For further details, see: <https://www.wiley.com/en-us/network/publishing/research-publishing/trending-stories/upholding-image-integrity-wileys->

image-screening-service

We look forward to receiving your revised submission.

Yours sincerely,

Vaughan Macefield
Senior Editor
The Journal of Physiology

EDITOR COMMENTS

Reviewing Editor:

Dear Author,

In the second round of review, two referees provided positive recommendations. A third referee expressed that some concerns were not fully addressed, noting that further debate over these interpretations would not be productive. However, it was emphasised that ensuring full transparency is essential so that readers can form their own judgments.

The referee offered the following specific suggestions, which I agree:

- The limitation related to the urethane-induced increase in blood osmolality, which may affect the interpretation of neuronal activity, should be clearly acknowledged not only in the Discussion but also in the Abstract.
- The Key Points Summary should explicitly state that the conclusions are based on experiments conducted in urethane-anesthetized animals, to prevent overinterpretation by readers.

Please revise the manuscript accordingly so that the paper may proceed toward final acceptance.

Senior Editor:

Thank you for attending to the reviewers' comments. As you will see from the Reviewing Editor's summary, reviewers 1 and 2 are satisfied but reviewer 3 requires acknowledgement of the use of urethane anaesthesia, and this should be acknowledged in the Key Points, Abstract and in the Discussion. Once you have done this your manuscript will be accepted. I look forward to receiving your amendments shortly.

REFEREE COMMENTS

Referee #1:

The authors have considered the reviewer's comments and largely dealt with them in a thoughtful and reasonable way in the revised manuscript. I have no further issues with the manuscript.

Referee #2:

The text revisions are appreciated. They place the experimental findings in a broader perspective thereby increase the impact of the study.

END OF COMMENTS

26/08/2025

Professor Mike Ludwig

Professor of Neurophysiology
Centre for Discovery Brain Sciences
University of Edinburgh
George Square
Edinburgh EH8 9XD, UK

Tel: -44 (0) 131 650 3275
Fax: -44 (0) 131 650 6527
email: mike.ludwig@ed.ac.uk

Dear Dr Macefield,

We have now acknowledged the use of urethane anaesthesia in the Key Points, Abstract and in the Discussion as requested. All changes in text have been marked in yellow.

We hope that these changes are to your satisfaction.

With best wishes

Mike Ludwig
Professor of Neurophysiology

Dear Professor Ludwig,

Re: JP-RP-2025-288860R3 "**Reduced osmoresponsiveness in magnocellular neuroendocrine neurones during chronic salt-loading**" by Maja Lozic, Roongrit Klinjampa, Nancy Sabatier, Duncan J MacGregor, Gareth Leng, and Mike Ludwig

We are pleased to tell you that your paper has been accepted for publication in The Journal of Physiology.

Yours sincerely,

Vaughan Macefield
Senior Editor
The Journal of Physiology

If you would like to receive our 'Research Roundup', a monthly newsletter highlighting the cutting-edge research published in The Physiological Society's family of journals (The Journal of Physiology, Experimental Physiology, Physiological Reports, The Journal of Nutritional Physiology and The Journal of Precision Medicine: Health and Disease), please click this link, fill in your name and email address and select 'Research Roundup':
<https://www.physoc.org/journals-and-media/membernews>

- You can help your research get the attention it deserves! Check out Wiley's free Promotion Guide for best-practice recommendations for promoting your work at: www.wileyauthors.com/eeo/guide. You can learn more about Wiley Editing Services which offers professional video, design, and writing services to create shareable video abstracts, infographics, conference posters, lay summaries, and research news stories for your research at: www.wileyauthors.com/eeo/promotion.

EDITOR COMMENTS

Reviewing Editor:

The authors have submitted a revised version of the manuscript and addressed the referees' concerns satisfactorily, and the manuscript has been improved accordingly. In my assessment, the article is now suitable for publication in the Journal of

Physiology.

Senior Editor:

Thank you for attending to these remaining issues. I am pleased to report that your manuscript is now considered acceptable for publication in The Journal of Physiology.